# An Uncertainty Principle is a Price of Privacy-Preserving Microdata

**John Abowd**
U.S. Census Bureau
and Cornell University

**Robert Ashmead**
U.S. Census Bureau

**Ryan Cumings-Menon**
U.S. Census Bureau

**Simson Garfinkel**
(formerly) U.S. Census Bureau
U.S. Department of Homeland Security
and George Washington University

**Daniel Kifer**
U.S. Census Bureau
and Penn State University

**Philip Leclerc**
U.S. Census Bureau

**William Sexton**
(formerly) U.S. Census Bureau
and Tumult Labs

**Ashley Simpson**
Knexus

**Christine Task**
Knexus

**Pavel Zhuravlev**
U.S. Census Bureau

## Abstract

Privacy-protected microdata are often the desired output of a differentially private algorithm since microdata is familiar and convenient for downstream users. However, there is a statistical price for this kind of convenience. We show that an uncertainty principle governs the trade-off between accuracy for a population of interest ("sum query") vs. accuracy for its component sub-populations ("point queries"). Compared to differentially private query answering systems that are not required to produce microdata, accuracy can degrade by a logarithmic factor. For example, in the case of pure differential privacy, without the microdata requirement, one can provide noisy answers to the sum query and all point queries while guaranteeing that each answer has squared error $O(1/\epsilon^2)$. With the microdata requirement, one must choose between allowing an additional $\log^2(d)$ factor ($d$ is the number of point queries) for some point queries or allowing an extra $O(d^2)$ factor for the sum query. We present lower bounds for pure, approximate, and concentrated differential privacy. We propose mitigation strategies and create a collection of benchmark datasets that can be used for public study of this problem.

## 1  Introduction

Differential Privacy [16] is a mathematical theory of information leakage that allows organizations to publish noisy statistics about their datasets while protecting the confidentiality of user information. Its state-of-the-art guarantees have resulted in adoption by data collectors such as the U.S. Census Bureau [31, 10, 23, 1], Google [19, 6], Apple [37], Microsoft [13], Uber [26], and Facebook [33].

In many cases, downstream users want the output of disclosure avoidance systems in the form of microdata (a set of records about individuals). For example, this has historically been the case for tabulations of Census Bureau data, and is currently a requirement for most 2020 Census of Population and Housing tabulations[20]. However, an end-user study of demonstration data products released by an early prototype of the Census Bureau's disclosure avoidance system showed significant

anomalies in the privacy-protected microdata [34].[1] They noted the following: the system first produced differentially private noisy query answers, called *measurements*, and then synthesized privacy-protected microdata so that query answers computed from the privacy-protected microdata matched the noisy measurements as closely as possible (based on some objective function). However, after the privacy-protected microdata were created, they compared (1) the original measurement query noisy answers and (2) the values of the same queries computed from the privacy-protected microdata. They noted that in some cases, the query error from the privacy-protected microdata was "much larger" than the measurement query error [34].

In this paper, we show that such anomalies are an inherent and unavoidable consequence of privacy-protected microdata (they affect all differentially private algorithms that must output microdata). We further show that the additional errors caused by privacy-protected microdata also satisfy a *new* uncertainty principle that trades off error between accuracy on populations and accuracy on sub-populations. We next explain this principle.

First, our criterion is *per-query* expected squared error. That is, if $Q$ is a collection of queries, $\mathfrak{D}$ is the true data, and $\widetilde{\mathfrak{D}}$ is the privacy-protected microdata, we are interested in the left side of Equation 1 (below), where the expectation is taken over the randomness of the algorithm that ingests $\mathfrak{D}$ and outputs privacy protected $\widetilde{\mathfrak{D}}$.

$$\underbrace{\max_{q \in Q} E_{\widetilde{\mathfrak{D}}}[(q(\mathfrak{D}) - q(\widetilde{\mathfrak{D}}))^2]}_{\text{Our focus: per-query error}} \leq \underbrace{E_{\widetilde{\mathfrak{D}}}[\max_{q \in Q}(q(\mathfrak{D}) - q(\widetilde{\mathfrak{D}}))^2]}_{\text{Most other papers: simultaneous/outlier error.}} \quad . \tag{1}$$

This metric measures whether there exist "bad" queries that have systematically large errors *on average*. It is *not* to be confused with simultaneous/outlier noise error (right side of Equation 1) that is the focus of most theoretical papers on differential privacy, such as [7]. The reason is that simultaneous error cannot distinguish between systematic error in specific queries vs. outliers that result by chance when dealing with many random variables. On the other hand per query-error can make this distinction because it considers the average behavior of each query separately.

Next, consider a collection of $d$ disjoint[2] counting queries $q_1, \ldots, q_d$ and a special query $q_*$ that is equal to their sum ($q_*(\mathfrak{D}) = \sum_i q_i(\mathfrak{D})$). We call $q_1, \ldots, q_d$ the *point queries* and $q_*$ the *sum* query. Examples include (1) $q_*(\mathfrak{D}) = $ "# of Black or African Americans in the data living in California" and $q_i(\mathfrak{D}) = $ "# of Black or African Americans in the data living in county $i$ in California" and (2) $q_*(\mathfrak{D}) = $ "population of a given county" (which can be used in federal and state-level funding allocations) and $q_i(\mathfrak{D}) = $ "population in census block $i$ in that county" (useful for redistricting). Thus, for different use-cases, accuracies at these local and aggregate scales are important.

It is well-known that queries $q_1, \ldots, q_d, q_*$ can be answered using $\epsilon$-differential privacy by adding Laplace($2/\epsilon$) noise to each query [18], thus guaranteeing that each query answer has expected squared error $8/\epsilon^2$. However, in this paper, we show that it is not possible to guarantee this kind of error if one is required to produce differentially private microdata $\widetilde{\mathfrak{D}}$ and answer queries using it (i.e., computing $q_1(\widetilde{\mathfrak{D}}), \ldots, q_d(\widetilde{\mathfrak{D}}), q_*(\widetilde{\mathfrak{D}})$). Specifically, suppose an $\epsilon$-differentially private microdata-producing algorithm can guarantee that, for all datasets $\mathfrak{D}$, $E_{\widetilde{\mathfrak{D}}}[(q_*(\mathfrak{D}) - q_*(\widetilde{\mathfrak{D}}))^2] \leq D^2$ and $\max_i E_{\widetilde{\mathfrak{D}}}[(q_i(\mathfrak{D}) - q_i(\widetilde{\mathfrak{D}}))^2] \leq C^2$ for some constants $C$ and $D$. Then one has to choose:

- If $D^2 \in O(1/\epsilon^2)$ then $C^2 \in \Omega(\frac{1}{\epsilon^2} \log^2(d))$. That is, making the sum query accurate may force us to take a $\log^2(d)$ penalty in the expected squared error some of the point queries, or

- If $C^2 \in O(1/\epsilon^2)$ then $D^2 \in \Omega(\frac{d^2}{\epsilon^2})$. That is, a low per-query error guarantee for point queries may increase expected squared error of the sum query by a factor of $d^2$.

We present such lower bound results for pure differential privacy [16], approximate differential privacy [15], and concentrated differential privacy [8], with nearly matching upper bounds.

---

[1]Throughout this paper we use *privacy-protected* and *privacy-preserving* synonymously. The Census Bureau prefers "privacy-protected," whereas the scientific literature has more often used "privacy-preserving." Both terms mean that the confidentiality of individual responses has been protected using differentially private algorithms.

[2]That is, adding/removing a record into the data can only affect the answer to **one** of the queries.

We note that this uncertainty principle affects some, but not all, possible datasets. That is, there are datasets for which the error penalties do not exist. Thus, the goal in practical privacy-protected microdata generation should be to minimize the occurrence of this uncertainty principle (since eliminating it entirely is impossible). To this end, we propose a benchmark suite of real and synthetic datasets that can be used by the wider community for further study of this problem. We also propose some algorithms, inspired by our lower and upper bound proofs, for mitigating the effects of this uncertainty principle. Limitations: empirically, these algorithms perform well on the benchmarks but we do not have theoretical proofs of performance.

## 2  Preliminaries

Let $\mathfrak{D}$ denote a dataset, $M$ a differentially private algorithm, and let $\widetilde{\mathfrak{D}}$ be a privacy-preserving dataset (e.g., $M(\mathfrak{D}) = \widetilde{\mathfrak{D}}$). A counting query $q$ is associated with a predicate $\psi$, and the query answer $q(\mathfrak{D})$ is the number of records in $\mathfrak{D}$ that satisfy $\psi$. We let $q_1, \ldots, q_d$ represent a set of $d$ counting queries whose corresponding predicates $\psi_1, \ldots, \psi_d$ are **disjoint** (no record can satisfy more than one of the predicates). We also let $q_*$ denote their sum: $q_*(\mathfrak{D}) = \sum_{i=1}^{d} q_i(\mathfrak{D})$.

### 2.1  Differential Privacy

Differential privacy is currently considered the gold standard in privacy protections. It relies on the concept of neighboring datasets, defined as follows.

**Definition 1** (Neighbors). *Two datasets $\mathfrak{D}_1$ and $\mathfrak{D}_2$ are neighbors, denoted by $\mathfrak{D}_1 \sim \mathfrak{D}_2$, if $\mathfrak{D}_1$ can be obtained from $\mathfrak{D}_2$ by adding or removing one record.*

Using this concept of neighbors, differential privacy ensures that adding or removing one record from a dataset has little effect on the probabilistic outcomes of an algorithm:

**Definition 2** (Differential Privacy [16]). *Given privacy parameters $\epsilon > 0$ and $\delta \geq 0$, a randomized algorithm M satisfies $(\epsilon, \delta)$-DP if for all pairs of datasets $\mathfrak{D}_1, \mathfrak{D}_2$ that are neighbors of each other, and for all $S \subseteq \text{range}(M)$, the following equation holds:*

$$P(M(\mathfrak{D}_1) \in S) \leq e^{\epsilon} P(M(\mathfrak{D}_2) \in S) + \delta,$$

*where the probability is only over the randomness in $M$ (not the randomness in the data). When $\delta = 0$, we say that $M$ satisfies* pure differential privacy *(also known as $\epsilon$-differential privacy or $\epsilon$-DP) and when $\delta > 0$ we say that $M$ satisfies* approximate differential privacy.

Another important version of differential privacy, is $\rho$-zCDP (concentrated differential privacy):

**Definition 3** (zCDP [8]). *Given a privacy parameter $\rho$, a randomized algorithm $M$ satisfies $\rho$-zCDP if for all pairs of datasets $\mathfrak{D}_1, \mathfrak{D}_2$ that are neighbors of each other and all numbers $\alpha > 1$,*

$$\mathcal{D}_{\alpha}(M(\mathfrak{D}_1) || M(\mathfrak{D}_2)) \leq \rho \alpha$$

*where $\mathcal{D}_{\alpha}(P||Q) \equiv \frac{1}{\alpha-1} \log \left( E_{x \sim P} \left[ \frac{P(x)^{\alpha-1}}{Q(x)^{\alpha-1}} \right] \right)$ is the Renyi divergence of order $\alpha$ between probability distributions $P$ and $Q$.*

Although zCDP is difficult to interpret, there are useful results that help provide intuition. First, any $M$ that satisfies $\epsilon$-differential privacy also satisfies $\rho$-zCDP with $\rho = \frac{\epsilon^2}{2}$ [8]. In general a $\rho$-zCDP algorithm does not satisfy pure differential privacy but does satisfy $(\epsilon, \delta)$-DP for infinitely many pairs of $\epsilon$ and $\delta$ that lie along a curve (see [9] and [2] for conversions between $\rho$-zCDP and $(\epsilon, \delta)$-DP).

### 2.2  Algorithm Design with Differential Privacy

A few basic principles underlie the construction of many algorithms for differential privacy. The first is sensitivity, which measures the maximum impact that one record can have on a set of queries (regardless of input data):

**Definition 4** (Sensitivity [16]). *The $L_p$ global sensitivity of a set $Q$ of queries, denoted by $\Delta_p(Q)$, is defined as $\sup_{\mathfrak{D}_1 \sim \mathfrak{D}_2} \left( \sum_{q \in Q} |q(\mathfrak{D}_1) - q(\mathfrak{D}_2)|^p \right)^{1/p}$.*

Global sensitivity can be used with the Laplace and Gaussian distributions to form basic mechanisms. Let $Lap(\alpha)$ represent a draw from the Laplace distribution with density $f(x) = \frac{1}{2\alpha}e^{-|x|/\alpha}$ and $N(0, \sigma^2)$ represent the zero-mean Gaussian distribution with variance $\sigma^2$. Each appearance of $Lap(\alpha)$ or $N(0, \sigma^2)$ represents an independent sample from the corresponding distribution.

**Theorem 1** (Laplace Mechanism [16]). *Given a privacy parameter $\epsilon > 0$, a set $Q$ of queries, and an input dataset $\mathfrak{D}$, the mechanism $M$ that returns the set of noisy answers $\{q(\mathfrak{D}) + Lap(\Delta_1(Q)/\epsilon)\}_{q \in Q}$ satisfies $\epsilon$-differential privacy.*

**Theorem 2** (Gaussian Mechanism [8]). *Given a privacy parameter $\epsilon > 0$, a set $Q$ of queries, and an input dataset $\mathfrak{D}$, the mechanism $M$ that returns the set of noisy answers $\{q(\mathfrak{D}) + N(0, \Delta_2(Q)^2/(2\rho))\}_{q \in Q}$ satisfies $\rho$-zCDP.*

All of these privacy definitions are postprocessing invariant [18]. That is, let $A$ be an arbitrary algorithm. Then $A \circ M$ (i.e., the algorithm that outputs $A(M(\mathfrak{D}))$) satisfies $(\epsilon, \delta)$-DP (resp., $\rho$-zCDP) if $M$ satisfies $(\epsilon, \delta)$-DP (resp., $\rho$-zCDP); in other words, the privacy parameters do not degrade.

They also have useful sequential composition properties. Let $M_1, \ldots, M_k$ be algorithms that satisfy pure differential privacy with corresponding parameters $\epsilon_1, \ldots, \epsilon_k$ (resp., zCDP with corresponding privacy parameters $\rho_1, \ldots, \rho_k$), then the algorithm $M$ that releases all of their outputs (i.e., releases $M_1(\mathfrak{D}), \ldots, M_k(\mathfrak{D})$) satisfies $\sum_i \epsilon_i-$differential privacy [18] (resp., $\sum_i \rho_i$-zCDP [8]).

# 3 The Uncertainty Principle

The setting of $d$ disjoint queries $q_1, \ldots, q_d$ and their sum $q_*$ are some of the most important types of query sets. As discussed earlier, population counts in small geographic regions such as census blocks (examples of $q_i$) are important for redistricting while population counts in larger regions such as counties (examples of $q_*$) are used for federal and state funding formulas. Thus any tension between the $q_i$ and $q_*$ can have significant impact on the entire U. S. population. While this is just one example of a query set, almost every table produced in previous censuses is a query set with disjoint queries and their sums [40]. Thus this is an important collection of queries to study.

## 3.1 Lower Bounds

We first remove some restrictions on $M$. While its input is a dataset, its output can be a positively weighted dataset – a collection of records in which each record $r$ has a nonnegative weight $w$. A query $q$ with predicate $\psi$ can be evaluated over a weighted dataset by summing the weights of the records that satisfy $\psi$. This simplifies our proofs and slightly increases generality, since normal microdata is a special case of positively weighted data in which all weights are 1 (hence lower bounds for positively weighted data are also lower bounds for normal microdata). It also emphasizes the fact that these lower bounds arise specifically because negative query answers are disallowed. The lower bound is the following (see supplementary material for proofs).

**Theorem 3.** *Let $q_1 \ldots, q_d$ be a set collection of disjoint queries and let $q_*$ be their sum. Let $M$ be a randomized algorithm whose input is a dataset and whose output is a positively weighted dataset. Suppose $M$ guarantees that for each query $q_i$ and dataset $\mathfrak{D}$, $E[(q_i(\mathfrak{D}) - q_i(M(\mathfrak{D})))^2] \le C^2$ and $E[(q_*(\mathfrak{D}) - q_*(M(\mathfrak{D})))^2] \le D^2$ for some values $C$ and $D$, where the expectation is **only** over the randomness in $M$.*

- *If $M$ satisfies $\epsilon$-differential privacy then for any $k > 0$, we have $e^{2\epsilon(2C+k)} \ge \frac{k(d-1)}{16C+8D+4k}$ which implies **(a)** if $D^2 \le \lambda/\epsilon^2$ for some constant $\lambda$, then $C^2 \in \Omega(\frac{1}{\epsilon^2}\log^2(d))$, and **(b)** if $C \le \lambda/\epsilon^2$ then $D \in \Omega(d^2/\epsilon^2)$.*

- *If $M$ satisfies $(\epsilon, \delta)$-DP then for any $k > 0$, we have $\left(\frac{\delta}{\epsilon} + \frac{4C+2D+k}{k(d-1)}\right)e^{4\epsilon C + 2k\epsilon} \ge 1/4$, which implies **(a)** if $D^2 \le \lambda/\epsilon^2$ for some constant $\lambda$, then $C^2 \in \Omega\left(\min(\frac{1}{\epsilon^2}\log^2(d), \frac{1}{\epsilon^2}\log^2\frac{\epsilon}{\delta})\right)$; **(b)** if $C \le \lambda/\epsilon^2$ then either $\epsilon \in O(\delta)$ or $D^2 \in \Omega(d^2/\epsilon^2)$.*

- *If $M$ satisfies $\rho$-zCDP, then the tradeoff function between $C$ and $D$ (which is more complex and omitted due to space constraints) implies: **(a)** if $D^2 \le \lambda/\rho$ for some $\lambda$, then $C^2 \in \Omega\left(\log(d)/\rho\right)$, and **(b)** if $C^2 \le \lambda/\rho$, then for any $\gamma \in (0, 1)$, we must have $D^2 \in \Omega(d^{2\gamma}/\rho)$.*

Balcer and Vadhan [3] recently showed a statistical price of privacy-preserving release of the top-k counts in a histogram. They proved an analogous $O(\log^2(d/k))$ penalty for point queries under $\epsilon$-DP (and also results for approximate DP). Interestingly, although they did not consider tradeoffs with the sum query (since its value was assumed to be public in their work), the results in our Theorem 3 (for $\epsilon$-DP and approximate DP, but not zCDP) can be proved using the result of their Theorem 7.2.

We also note that the tradeoff functions between $C$ and $D$ in Theorem 3 show a much stronger result than items (a) and (b) in Theorem 3. For example, they rule out the possibility that both $C^2$ and $D^2$ can simultaneously be just slightly larger than $O(1/\epsilon^2)$. To understand and interpret Theorem 3, let us compare to the Laplace and Gaussian mechanisms, which can produce negative query answers, hence are not equivalent to producing positively weighted datasets (hence not covered by Theorem 3).

It is easy to see that $\Delta_1(q_1, \ldots, q_d, q_*) = 2$ and $\Delta_2(q_1, \ldots, q_d, q_*) = \sqrt{2}$. Hence, an algorithm $M'_\epsilon$ can add independent $\mathrm{Lap}(2/\epsilon)$ noise to each query to satisfy $\epsilon$-DP, and an algorithm $M'_\rho$ can add independent $N(0, 1/\rho)$ noise to each query to satisfy $\rho$-zCDP. Thus $M'_\epsilon$ achieves expected squared error of $8/\epsilon^2$ for $q_*$ and each $q_i$ (i.e., $C^2 = D^2 = 8/\epsilon^2$). Meanwhile $M'_\rho$ achieves $1/\rho$ expected squared error ($C^2 = D^2 = 1/\rho$). These expected error guarantees hold for all datasets $\mathfrak{D}$.

Theorem 3 says that privacy-preserving algorithms $M$ that are required to produce positively weighted datasets cannot guarantee the same low error – there are input datasets $\mathfrak{D}$ for which the expected errors can be significantly larger. In the case of $M$ that satisfy $\epsilon$-DP, if we want low error for the sum query (e.g., $D^2 = O(1/\epsilon^2)$, matching the Laplace mechanism), on some datasets we may need to pay a $\log^2(d)$ penalty for some point queries (i.e., there will be specific point queries with consistently large error). On the other hand, if we want low error for the point queries (e.g., $C^2 = O(1/\epsilon^2)$) then on some datasets we will pay a $d^2$ penalty on the sum query.

In the case of $\rho$-zCDP, the penalties are smaller. If we want to match the error of the Gaussian mechanism on the sum query, we may need to pay a penalty of $\log(d)$ on point queries; if we want $O(1/\rho)$ expected squared error on each point query, we may need to pay a penalty of nearly $d^2$ on $q_*$.

For approximate DP, the weakest privacy definition here, the degradation factor can be roughly $\log^2(\epsilon/\delta)$ no matter how large $d$ is.

**Remark 1.** The lower bounds in Theorem 3 imply that if privacy-preserving microdata is generated by obtaining noisy measurement query answers (e.g., with the Laplace or Gaussian mechanisms) and then postprocessing the noisy answers (e.g., [28, 24]), some of the measurement queries computed directly from the privacy-preserving microdata will have errors that are larger than their original noisy answers.

**Remark 2.** All is not lost, however, as the proofs are based on packing arguments that show that these errors are unavoidable for some difficult datasets (but not all datasets are difficult). An example of a difficult dataset $\mathfrak{D}^*$ under pure differential privacy is one for which exactly one of the query answers $q_1(\mathfrak{D}^*), \ldots, q_d(\mathfrak{D}^*)$ equals $\log(d)/\epsilon$ while the other $d-1$ queries equal 0 (clearly, $q_*(\mathfrak{D}^*) = \log(d)/\epsilon$). As mentioned earlier, the Laplace mechanism [18], which does not produce microdata, can achieve $8/\epsilon^2$ per query error although many of the noisy query answers will be negative. However, the proof of Theorem 3 implies that no algorithm that produces privacy-protected microdata (and hence nonnegative query answers) can do as well on such a dataset. In fact, for this specific difficult dataset $\mathfrak{D}^*$, the large error described by Theorem 3 will either occur for $q_*$ or for that $q_i$ whose answer on $\mathfrak{D}^*$ is $\log(d)/\epsilon$. On the other hand, an easy dataset is one for which $q_1(\mathfrak{D}), \ldots, q_d(\mathfrak{D})$ are all large, since almost no effort is needed in ensuring that the privacy-protected query answers are nonnegative.

## 3.2 Upper Bounds

These lower bounds are nearly tight, as shown by the upper bounds in Theorem 4. The proofs construct postprocessing algorithms that first obtain noisy answers $a_1, \ldots, a_d, a_*$ to the queries $q_1, \ldots, q_d, q_*$. A postprocessing step converts the $a_i$ and $a_*$ into consistent noisy answers $a'_1, \ldots, a'_d, a'_*$ (i.e., they are nonnegative and $\sum_i a'_i = a'_*$). Weighted datasets are constructed from the latter quantities. To get weighted datasets with higher accuracy on point queries, the postprocessing ignores $a_*$ and sets $a'_i = \max\{0, a_i\}$. To obtain synthetic data with higher accuracy on the sum query, $a'_*$ is set to $a_*$ and

the $a_i'$ are obtained by minimizing squared distance to the $a_i$ subject to the $a_i'$ being nonnegative and adding up to $a_*$. The full proofs are in the supplementary material.

**Theorem 4** (Upper bound for pure DP and zCDP). *Let $q_1, \ldots, q_d$ be a set of disjoint queries and let $q_*$ be their sum. Given privacy parameters $\epsilon > 0$ and $\rho > 0$, there exist algorithms $M_\epsilon, M_\rho, M_\epsilon', M_\rho',$ $M_{\epsilon,\delta}'$ that output a positively weighted dataset and have the following properties:*

1. *$M_\epsilon$ satisfies $\epsilon$-DP, and for all $\mathfrak{D}$ and $i$, $E\left[(q_i(M_\epsilon(\mathfrak{D})) - q_i(\mathfrak{D}))^2\right] \leq 2/\epsilon^2$ and $E\left[(q_*(M_\epsilon(\mathfrak{D})) - q_*(\mathfrak{D}))^2\right] \leq 2d^2/\epsilon^2$.*

2. *$M_\rho$ satisfies $\rho$-zCDP, and for all $\mathfrak{D}$ and $i$, $E\left[(q_i(M_\rho(\mathfrak{D})) - q_i(\mathfrak{D}))^2\right] \leq 1/(2\rho)$ and $E\left[(q_*(M_\rho(\mathfrak{D})) - q_*(\mathfrak{D}))^2\right] \leq d^2/(2\rho)$.*

3. *$M_\epsilon'$ satisfies $\epsilon$-DP, and for all $\mathfrak{D}$ and $i$, $E\left[(q_i(M_\epsilon'(\mathfrak{D})) - q_i(\mathfrak{D}))^2\right] \in O(\log^2(d)/\epsilon^2)$ and $E\left[(q_*(M_\epsilon'(\mathfrak{D})) - q_*(\mathfrak{D}))^2\right] \in O(1/\epsilon^2)$*

4. *$M_\rho'$ satisfies $\rho$-zCDP, and for all $\mathfrak{D}$ and $i$, $E\left[(q_i(M_\rho'(\mathfrak{D})) - q_i(\mathfrak{D}))^2\right] \in O(\log(d)/\rho)$ and $E\left[(q_*(M_\rho'(\mathfrak{D})) - q_*(\mathfrak{D}))^2\right] \in O(1/\rho)$*

5. *$M_{\epsilon,\delta}'$ satisfies $(\epsilon, \delta)$-DP and for all $\mathfrak{D}$ and $i$, $E\left[(q_i(M_{\epsilon,\delta}'(\mathfrak{D})) - q_i(\mathfrak{D}))^2\right] \in O(\log^2(1/\delta)/\epsilon^2 + 1)$ and $E\left[(q_*(M_{\epsilon,\delta}'(\mathfrak{D})) - q_*(\mathfrak{D}))^2\right] \in O(1/\epsilon^2)$. Also note $M_\epsilon$ and $M_\epsilon'$ satisfy $\epsilon, \delta$-DP.*

Note that Theorem 4 matches the lower bounds in Theorem 3 except for a slight difference for zCDP, where Item 2 of Theorem 4 has a $d^2$ while the lower bound in Theorem 3 has in its place a $d^{2\gamma}$ for any $\gamma$ arbitrarily close to 1.

## 4 Algorithms

For tabular data, typically end-users are interested in multiple marginals of the data. Examples include the gender by age marginals at the national, state, and county levels (for constructing age pyramids); the marginal on race at the national, state, county, tract, and block levels both for demographic research and for enforcement of voting rights; total populations in each state, county, etc. (for various funding formulas). Thus these query sets have many different point query/sum query collections embedded in them. Examples include: female population in a county (sum query) and number of females of each age in the county (point queries); or total Asian population (sum query) and Asian population in each county (point queries). Thus algorithms designed to minimize the appearance of the uncertainty principle should not be designed for a *single* collection of sum/point queries; instead, they should support *many* counting queries.

To describe algorithms, it is helpful to view the dataset $\mathfrak{D}$ as a vector $\mathbf{x}$, where each element $i$ corresponds to a possible record $r_i$. Then $\mathbf{x}[i]$ is the number of times $r_i$ appears in $\mathfrak{D}$. The goal is to produce a privacy-protected version $\widetilde{\mathbf{x}}$ whose entries are nonnegative real numbers, which can be converted to a positively weighted dataset $\widetilde{\mathfrak{D}}$ ($\widetilde{\mathbf{x}}[i]$ is the weight of record $r_i$ in $\widetilde{\mathfrak{D}}$). In this setting, a counting query $q$ is just a vector of 1s and 0s with the same dimensionality as $\mathbf{x}$, and the query answer is computed as the dot product $q \cdot \mathbf{x}$.

The algorithms we present here (2 baselines and 2 proposed algorithms) are all based on the idea of first computing noisy query answers and then postprocessing them to obtain $\widetilde{\mathbf{x}}$. This setup allows an organization to release both $\widetilde{\mathbf{x}}$ and the noisy answers (for more statistically-oriented end-users). Thus, given a set $Q$ of counting queries, for each $q \in Q$, the data collector computes a noisy answer $a_q$ by adding noise with distribution $F_q$ to the true answer and then must postprocess them to create microdata.[3] We assume the data collector chooses the noise distributions to achieve their desired privacy definition (e.g., $\epsilon$-DP, $\rho$-zCDP).

---

[3] Although a data collector could add noise to a different set of queries and use them to infer the answers to $q \in Q$ [43, 29, 42], it is the subsequent postprocessing step that would be more important in mitigating the uncertainty principle.

**Baseline: NNLS Postprocessing.** The first baseline we consider is the commonly used nonnegative least squares (NNLS), in which $\widetilde{\mathbf{x}}$ is produced as the solution to the following optimization problem:

$$\widetilde{\mathbf{x}} \leftarrow \arg\min_{\widetilde{\mathbf{x}}} \sum_{q \in Q} \frac{(a_q - q \cdot \widetilde{\mathbf{x}})^2}{variance(F_q)} \text{ s.t. } \widetilde{\mathbf{x}}[i] \geq 0 \text{ for all } i$$

**Baseline: Max Fitting Postprocessing.** The next baseline is an adaptation of a bilevel optimization approach [32] that was originally used for optimization problems whose parameters are sensitive. The idea here is to find the positively weighted datasets whose query answers minimize the $L_\infty$ distance to the noisy query answers, breaking ties using least squares error:

$$dist \leftarrow \min_{\widetilde{\mathbf{x}}} \max_{q \in Q} \frac{|a_q - q \cdot \widetilde{\mathbf{x}}|}{std(F_q)} \text{ s.t. } \widetilde{\mathbf{x}}[i] \geq 0 \text{ for all } i$$

$$\widetilde{\mathbf{x}} \leftarrow \arg\min_{\widetilde{\mathbf{x}}} \sum_{q \in Q} \frac{(a_q - q \cdot \widetilde{\mathbf{x}})^2}{variance(F_q)} \text{ s.t. } \max_{q \in Q} \frac{|a_q - q \cdot \widetilde{\mathbf{x}}|}{std(F_q)} \leq dist \text{ and } \widetilde{\mathbf{x}}[i] \geq 0 \text{ for all } i$$

**Sequential Fitting Postprocessing.** Since it is provably not always possible to output microdata that fits the noisy answers well, we propose an approach that prioritize queries. Thus the query set $Q$ is partitioned by the user into query sets $Q_1, \ldots, Q_k$. We use the above NNLS approach to fit a vector $\widetilde{\mathbf{x}}_1$ to the noisy answers of queries in $Q_1$ (highest priority). We then fit $\widetilde{\mathbf{x}}_2$ to the noisy answers for queries in $Q_2$ (next highest priority) subject to the constraints that $\widetilde{\mathbf{x}}_2$ matches $\widetilde{\mathbf{x}}_1$ on queries in $Q_1$. Then we fit $\widetilde{\mathbf{x}}_3$ using noisy answers to queries in $Q_3$ while forcing $\widetilde{\mathbf{x}}_3$ to match $\widetilde{\mathbf{x}}_2$ on queries in $Q_1$ and $Q_2$, and so on and return the final $\widetilde{\mathbf{x}}_k$ at the end. The pseudocode is shown in Algorithm 1. This algorithm is the one that matches the upper bounds in Theorem 4 (referred to as $M'_\epsilon$ when the noisy answers $a_q$ use Laplace noise, and $M'_\rho$ for Gaussian noise).

---

**Algorithm 1:** Sequential Fitting (Postprocessing)

---

**1 Input:** Query set $Q$, noisy answers $a_q$ for $q \in Q$ and noise distributions $F_q$ for $q \in Q$.
**2 Input:** $Q_1, \ldots, Q_k$: partition of $Q$ based on query priority.
**3** $\widetilde{\mathbf{x}}_1 \leftarrow \arg\min_{\widetilde{\mathbf{x}}} \sum_{q \in Q_1} \frac{(a_q - q \cdot \widetilde{\mathbf{x}})^2}{variance(F_q)}$ s.t. $\widetilde{\mathbf{x}}[i] \geq 0$ for all $i$
**4** Fit $\leftarrow Q_1$
**5 for** $\ell = 2, \ldots, k$ **do**
**6** $\quad$ $\widetilde{\mathbf{x}}_\ell \leftarrow \arg\min_{\widetilde{\mathbf{x}}} \sum_{q \in Q_\ell} \frac{(a_q - q \cdot \widetilde{\mathbf{x}})^2}{variance(F_q)}$ s.t. $\widetilde{\mathbf{x}}[i] \geq 0$ for all $i$ and $q \cdot \widetilde{\mathbf{x}} = q \cdot \widetilde{\mathbf{x}}_{\ell-1}$ for all $q \in$ Fit
**7** $\quad$ Fit $\leftarrow$ Fit $\cup Q_\ell$
**8 Return:** $\widetilde{\mathbf{x}}_k$

---

**Remark.** The constrained optimizations in max fitting and sequential fitting are difficult for quadratic program optimizers, often resulting in numerical errors, slow convergence, and infeasibility errors (due to occasional insufficient solution quality in earlier stages of the multistage optimization). They require significant engineering effort, tuning of slack parameters (slightly relaxing equality and inequality constraints) and optimizer-specific parameters. So, an ideal solution would also avoid constraints other than nonnegativity for point queries. This is a rationale for our next method.

**ReWeighted Fitting Postprocessing.** This method (shown in Algorithm 2) avoids constraints as much as possible in an eventual NNLS solve (Line 14) but is limited to query sets of the form $Q = \bigcup_{i=1}^{k} Q_i$, where the queries inside each $Q_i$ are disjoint and have the same noise distribution. One example is when $Q$ is a collection of marginal queries (e.g., $Q_1 =$ marginal on age, $Q_2 =$ marginal on age by race, $Q_3 =$ marginal on gender by race), which are arguably the most important types of queries. Within each $Q_i$, the algorithm tries to find a cutoff value so that queries with noisy answers above it are likely to have true value that is non-zero (Lines 5-6). The idea is that if $n_\dagger$ is the number of queries below the threshold, and if they truly had value 0, then their largest noisy value (i.e., the max of $n_\dagger$ 0-mean Laplace or Gaussian random variables) should not be near the cutoff with high probability (controlled by the confidence parameter $\gamma$). The "low" queries are the ones with noisy answers below the cutoff. The algorithm uses the existing noisy answers to estimate the sum of these "low" queries (Lines 11-12) and adds that "low query sum" (Line 13) to the nonnegative

least squares optimization while downweighting the individual low queries (Line 9, the downweight depends on the extreme value distribution of the max of $n_\dagger$ 0-mean Laplace or Gaussian random variables, Line 7). To avoid double counting, both places where a "low" query is used (individually and as part of a sum) have their weights cut in half. Note the algorithm only uses existing noisy answers and has no access to the true data.

---

**Algorithm 2:** ReWeighted Fitting (Postprocessing)

---

1 **Input:** Query set $Q = \bigcup_{i=1}^{k} Q_i$; Within a $Q_i$, the queries are disjoint. $F_i$ is the noise distribution of each query in $Q_i$. Given noisy answers $a_q$ for $q \in Q$ that satisfy the chosen privacy definition.

2 **Input:** Confidence parameter $\gamma$ close to 1 (e.g., 0.99, the setting used in experiments)

3 $S \leftarrow \emptyset$ **for** $i = 1, \ldots, k$ **do**

4      $a_{(1)}, a_{(2)}, \ldots$ are the given noisy answers (to queries in $Q_i$) arranged in sorted order

5      $j^* \leftarrow$ smallest $j$ s.t. $P(\max(j$ fresh random variable with distribution $F_i) \geq a_{(j)}) \leq 1 - \gamma$

6      $cutoff \leftarrow a_{(j^*)}$.

7      $downweight \leftarrow$ median of distribution of max of $j$ random variables sampled from $F_i$

8      For each query $q \in Q_i$ whose noisy answer $a_q$ is $\geq cutoff$, add $(q, a_q, 1/var(F_i))$ to $S$.

9      For each query $q \in Q_i$ whose $a_q$ is $< cutoff$, add $(q, a_q, \frac{1}{2*var(F_i)*downweight^2})$ to $S$.

10      $n_i^\dagger \leftarrow$ number of queries selected in Line 9 (i.e., their noisy answers were $< cutoff$)

11      $q_\dagger \leftarrow$ sum of queries selected in Line 9

12      $a_\dagger \leftarrow$ sum of their existing noisy answers

13      Add $(q_\dagger, a_\dagger, \frac{1}{2*n_i^\dagger var(F_i)})$ to $S$

14 $\widetilde{\mathbf{x}} \leftarrow \arg\min_{\widetilde{\mathbf{x}}} \sum_{(q', a', w') \in S} w'(q'(\widetilde{\mathbf{x}}) - a')^2$ s.t., $\widetilde{\mathbf{x}}[i] \geq 0$ for all $i$.

15 **Return:** $\widetilde{\mathbf{x}}$

---

## 5 Experiments

To make our code fully open source, we wrote it in Julia [5] and after trying several open-source optimizers, we settled on COSMO [21]. We created a collection of benchmark datasets that were small enough to permit running the postprocessing algorithms thousands of times on each dataset (to estimate expected errors) but large enough to demonstrate the uncertainty principle. The full benchmark of 15 real datasets and 16 synthetic datasets is described in the supplementary material.[4] Here we present results for an interesting subset. The only synthetic dataset discussed here, called Level00-2d, is a $10 \times 10$ histogram where one element is large (i.e., 10,000) and the others are 0. The other 15 datasets we discuss here were taken from the 2016 ACS Public-Use Microdata Sample [39]. Each represents a $9 \times 24$ "race by Hispanic origin" histogram from Public-Use Microdata Areas that were considered outliers in their states in terms of racial composition.

For these datasets, we applied the Laplace mechanism with $\epsilon = 0.5$ to answer the sum query, both 1-way marginal queries, and identity queries (for each cell, how many people are in it). This is also the priority order used by Sequential Fitting. Error results for the marginals, other privacy parameters and zCDP results can be found in the supplementary material. We ran the Laplace mechanism using different postprocessing strategies (described in Section 4) 1,000 times for each dataset to estimate expected squared error of each query. We added an ordinary least squares (OLS) optimization for comparison purposes (OLS is NNLS without nonnegativity constraints). OLS is free from the uncertainty principle because it does not produce positively weighted microdata. Thus, to minimize the effect of the uncertainty principle, the other postprocessing methods should try to achieve errors that are not much worse than OLS. We note that the multi-stage optimization in Max and Sequential fitting are generally very difficult for optimization software, so we only kept those runs in which the optimizer succeeded (thus results for Max and Sequential Fitting are slightly optimistically biased).

In Table 1, we show the squared error of these postprocessing methods for the sum query. The NNLS and MaxFitting baselines perform poorly for this query, with errors typically 4-5x those of the OLS method (which is close to the variance of the original noisy answer to the sum query). Meanwhile

---

[4]See `https://github.com/uscensusbureau/CostOfMicrodataNeurIPS2021` for the code and data.

| Dataset Nickname | Dataset | OLS | NNLS | MaxFit | Seq | ReWeight |
|---|---|---|---|---|---|---|
| $\mathfrak{D}01$ | Level00-2d | 101.3 | 461.9 | 533.9 | 149.2 | 108.5 |
| $\mathfrak{D}02$ | PUMA0101301 | 107.2 | 547.2 | 500.3 | 106.7 | 112.5 |
| $\mathfrak{D}03$ | PUMA0800803 | 107.2 | 446.1 | 571.7 | 120.3 | 107.2 |
| $\mathfrak{D}04$ | PUMA1304600 | 107.2 | 408.1 | 426.3 | 120.8 | 109.8 |
| $\mathfrak{D}05$ | PUMA1703529 | 107.2 | 435.3 | 426.3 | 134.9 | 110.9 |
| $\mathfrak{D}06$ | PUMA1703531 | 107.2 | 584.0 | 677.4 | 111.4 | 108.1 |
| $\mathfrak{D}07$ | PUMA1901700 | 107.2 | 395.1 | 443.6 | 119.1 | 110.4 |
| $\mathfrak{D}08$ | PUMA2401004 | 107.2 | 369.3 | 329.0 | 109.6 | 107.5 |
| $\mathfrak{D}09$ | PUMA2602702 | 107.2 | 467.8 | 472.0 | 146.0 | 109.2 |
| $\mathfrak{D}10$ | PUMA2801100 | 107.2 | 543.7 | 558.2 | 117.8 | 110.8 |
| $\mathfrak{D}11$ | PUMA2901901 | 107.2 | 485.2 | 464.4 | 126.5 | 110.8 |
| $\mathfrak{D}12$ | PUMA3200405 | 107.2 | 329.1 | 301.0 | 122.9 | 108.4 |
| $\mathfrak{D}13$ | PUMA3603710 | 107.2 | 300.3 | 293.3 | 85.7 | 108.8 |
| $\mathfrak{D}14$ | PUMA3604010 | 107.2 | 399.9 | 386.5 | 129.8 | 111.3 |
| $\mathfrak{D}15$ | PUMA5101301 | 107.2 | 396.1 | 369.5 | 139.2 | 107.2 |
| $\mathfrak{D}16$ | PUMA5151255 | 107.2 | 330.7 | 280.3 | 139.1 | 107.8 |

Table 1: Squared Error for Sum Query (overall $\epsilon = 0.5$))

| | OLS | | NNLS | | MaxFit | | Seq | | ReWeight | |
|---|---|---|---|---|---|---|---|---|---|---|
| Data | Total | Max | Total | Max | Total | Max | Total | Max | Total | Max |
| $\mathfrak{D}01$ | 10516.5 | 124.0 | 344.2 | 147.4 | 443.7 | 173.1 | 437.3 | 283.6 | 159.2 | 78.4 |
| $\mathfrak{D}02$ | 23906.2 | 142.9 | 809.0 | 135.6 | 910.7 | 144.6 | 782.9 | 179.7 | 731.3 | 209.8 |
| $\mathfrak{D}03$ | 23906.2 | 142.9 | 1179.8 | 107.5 | 1235.3 | 125.4 | 1171.7 | 189.8 | 1123.8 | 141.9 |
| $\mathfrak{D}04$ | 23906.2 | 142.9 | 1313.0 | 111.9 | 1385.5 | 142.3 | 1049.1 | 126.3 | 1264.4 | 136.0 |
| $\mathfrak{D}05$ | 23906.2 | 142.9 | 1243.8 | 105.3 | 1257.2 | 96.7 | 1019.1 | 114.3 | 1285.7 | 160.5 |
| $\mathfrak{D}06$ | 23906.2 | 142.9 | 562.2 | 94.9 | 599.0 | 72.1 | 429.9 | 112.9 | 409.8 | 78.8 |
| $\mathfrak{D}07$ | 23906.2 | 142.9 | 1516.1 | 115.9 | 1665.9 | 129.7 | 1312.1 | 156.9 | 1617.1 | 205.0 |
| $\mathfrak{D}08$ | 23906.2 | 142.9 | 1954.4 | 130.0 | 1971.8 | 147.8 | 1983.4 | 311.3 | 1760.1 | 168.9 |
| $\mathfrak{D}09$ | 23906.2 | 142.9 | 977.2 | 100.0 | 956.4 | 109.4 | 843.4 | 121.7 | 930.1 | 156.2 |
| $\mathfrak{D}10$ | 23906.2 | 142.9 | 686.9 | 97.5 | 705.7 | 79.0 | 534.2 | 92.7 | 516.0 | 78.7 |
| $\mathfrak{D}11$ | 23906.2 | 142.9 | 944.4 | 100.4 | 919.2 | 103.2 | 809.4 | 131.6 | 888.2 | 138.2 |
| $\mathfrak{D}12$ | 23906.2 | 142.9 | 2189.2 | 119.6 | 2191.5 | 134.7 | 1918.5 | 142.3 | 2336.1 | 259.1 |
| $\mathfrak{D}13$ | 23906.2 | 142.9 | 2884.1 | 119.2 | 3088.6 | 149.1 | 2484.2 | 140.7 | 2870.4 | 166.1 |
| $\mathfrak{D}14$ | 23906.2 | 142.9 | 1432.5 | 105.9 | 1442.3 | 120.6 | 1262.1 | 122.7 | 1448.6 | 194.0 |
| $\mathfrak{D}15$ | 23906.2 | 142.9 | 1474.7 | 108.3 | 1498.6 | 101.8 | 1394.5 | 203.4 | 1392.9 | 153.2 |
| $\mathfrak{D}16$ | 23906.2 | 142.9 | 2239.7 | 130.3 | 2274.1 | 124.3 | 2079.0 | 178.5 | 2123.0 | 172.8 |

Table 2: Squared Errors Id Query (overall $\epsilon = 0.5$).

Sequential and ReWeighted fitting perform much better. Standard errors were roughly 2-6% of the reported metrics (omitted for space, but shown in the supplementary materials).

For Table 2 we examine the expected errors of each cell query (i.e., $q_i$ is the number of people in cell $i$). We find the cell with the largest expected error and report it (the "Max" column). We also find the total squared error of the cell queries and report them in the "Total" column. Again, the standard errors are roughly 2-6% of the reported metrics, except that they are sometimes higher for Max and Sequential fitting since averages were only computing on the subset of runs for which the optimizer did not fail.

Generally, NNLS performed slightly better in terms of the maximum expected error compared to ReWeight, although their total errors are comparable and ReWeight significantly outperforms NNLS on the sum query.

Overall, these experiments and our supplementary material show that both ReWeight and Sequential fitting (though not perfect) avoid incidents where there are extremely high errors (unlike NNLS and Max Fitting for sum queries), and this is important in practice. ReWeight and Sequential fitting have similar performance. ReWeight is faster while Sequential needs significant tuning of optimizers in order to succeed. However, one advantage of Sequential is its algorithmic transparency – it can

directly prioritize queries for the tradeoffs caused by the uncertainty principle (in our experiments, the sum query had highest priority for Sequential Fitting).

# 6    Related Work

The requirement to produce microdata is an example of consistency in privacy-preserving query answering. A variety of work [4, 25, 35, 28, 11, 14, 30, 27, 24] has shown that creation of a privacy-preserving data synopsis from which all queries are answered can improve query accuracy under a variety of metrics such as maximum simultaneous error and total error. However, it is known that the production of privacy-preserving microdata comes at the expense of increased computational cost [41, 17, 38]. For example, under standard complexity assumptions [38], there is no polynomial-time algorithm for generating privacy-protected synthetic data whose two-way marginals are all accurate.

Aside from the computational price, Balcer and Vadhan [3] also recently showed a statistical price of privacy-protected synthetic data. They considered releasing different kinds of privacy-protected representations of nonnegative noisy histograms (for example, releasing the top-k noisy cells under $\epsilon$-DP had a $\log^2(d/k)$ penalty term for squared error), but assumed the value of the sum query was publicly known in their work. Our constructions are based on their proof techniques (see discussion after Theorem 3).

# 7    Conclusions, Future Work, and Broader Impact

Public-use data have many different end-users, so a single aggregated performance measure, such as total error across all queries, is not a reliable measure of data quality. The accuracy of each query is important, which implies multiple conflicting quality criteria for public-use data. Thus an important direction for future work is to identify all tradeoffs in privacy-preserving microdata as well as algorithms with provable guarantees on instance-optimality (i.e., improve performance on datasets that do not trigger the uncertainty principles).

### Broader Impact

The uncertainty principle presented in this paper (as well as the cost of microdata results in [3]) along with the known computational price of generating microdata suggests that organizations should also consider alternative formats for their privacy-protected data products. The uncertainty principle can be avoided by releasing noisy query answers that are allowed to be negative or by producing weighted datasets that can feature negative weights (however, adding a sparsity requirement could re-introduce systematic errors [3]). Such alternative formats may also require educating and providing training materials to end-users. If an organization nevertheless decides to produce privacy-protected microdata, then microdata-generating algorithms should be designed as postprocessing algorithms that convert unbiased noisy measurements into microdata (so that the "statistics-friendly" noisy measurements can also be released and studied by data scientists). Further research into such postprocessing algorithms is needed to mitigate the effects of the uncertainty principle.

### Acknowledgments and Disclosure of Funding

We thank Salil Vadhan for helpful discussions that allowed us to sharpen the lower bound results. Affiliations are provided solely for the purpose of identification. All work was performed under the supervision of the U.S. Census Bureau as part of the authors' employment or contractual work product. The views and opinions in this article are those of the authors and do not represent the policy or official position of the U.S. Government, the U.S. Department of Commerce, the U.S. Census Bureau, the U.S. Department of Homeland Security, Knexus, or Tumult Labs.

**Competing interests:** None.

**Additional revenues related to this work:** None.

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
