# A Appendix (Supplementary Material)

## A.1 Proofs Lower Bound Results

**Theorem 3.** *Let $q_1 \ldots, q_d$ be a set collection of disjoint queries and let $q_*$ be their sum. Let $M$ be a randomized algorithm whose input is a dataset and whose output is a positively weighted dataset. Suppose $M$ guarantees that for each query $q_i$ and dataset $\mathfrak{D}$, $E[(q_i(\mathfrak{D}) - q_i(M(\mathfrak{D})))^2] \le C^2$ and $E[(q_*(\mathfrak{D}) - q_*(M(\mathfrak{D})))^2] \le D^2$ for some values $C$ and $D$, where the expectation is **only** over the randomness in $M$.*

- *If $M$ satisfies $\epsilon$-differential privacy then for any $k > 0$, we have $e^{2\epsilon(2C+k)} \ge \frac{k(d-1)}{16C+8D+4k}$ which implies **(a)** if $D^2 \le \lambda/\epsilon^2$ for some constant $\lambda$, then $C^2 \in \Omega(\frac{1}{\epsilon^2}\log^2(d))$, and **(b)** if $C \le \lambda/\epsilon^2$ then $D \in \Omega(d^2/\epsilon^2)$.*

- *If $M$ satisfies $(\epsilon, \delta)$-DP then for any $k > 0$, we have $\left(\frac{\delta}{\epsilon} + \frac{4C+2D+k}{k(d-1)}\right)e^{4\epsilon C+2k\epsilon} \ge 1/4$, which implies **(a)** if $D^2 \le \lambda/\epsilon^2$ for some constant $\lambda$, then $C^2 \in \Omega\left(\min(\frac{1}{\epsilon^2}\log^2(d), \frac{1}{\epsilon^2}\log^2\frac{\epsilon}{\delta})\right)$; **(b)** if $C \le \lambda/\epsilon^2$ then either $\epsilon \in O(\delta)$ or $D^2 \in \Omega(d^2/\epsilon^2)$.*

- *If $M$ satisfies $\rho$-zCDP, then the tradeoff function between $C$ and $D$ (which is more complex and omitted due to space constraints) implies: **(a)** if $D^2 \le \lambda/\rho$ for some $\lambda$, then $C^2 \in \Omega(\log(d)/\rho)$, and **(b)** if $C^2 \le \lambda/\rho$, then for any $\gamma \in (0,1)$, we must have $D^2 \in \Omega(d^{2\gamma}/\rho)$.*

*Proof.* The lower bounds for pure and approximate DP (but not zCDP) can be proved as consequences of the work of Balcer and Vadhan [3]. To make this material more self-contained, we write out a direct proof of the lower bounds by borrowing their proof technique.

For notational convenience, we will let $\mathbf{x}[i]$ denote $q_i(\mathfrak{D})$ (so $\sum_i \mathbf{x}[i] = \sum_i q_i(\mathfrak{D}) = q_*(\mathfrak{D})$). Similarly, we let $\widetilde{\mathbf{x}}[i]$ denote $q_i(\widetilde{\mathfrak{D}})$ (so $\sum_i \widetilde{\mathbf{x}}[i] = \sum_i q_i(\widetilde{\mathfrak{D}}) = q_*(\widetilde{\mathfrak{D}})$). Thus the vector $\mathbf{x}$ represents the true point query answers and $\widetilde{\mathbf{x}}$ represents the privacy protected point query answers. In particular, $\mathbf{x}$ is a vector of nonnegative integers and $\widetilde{\mathbf{x}}$ is a vector of nonnegative real numbers.

All probabilities are taken with respect to only the randomness in $M$.

In this proof, $\alpha$, $\beta$, and $k$ are constants that we will set later. For any fixed $j$, by Markov's inequality,

$$P(|\widetilde{\mathbf{x}}[j] - \mathbf{x}[j]| \ge \alpha C) \le \frac{E\left[(\mathbf{x}[j] - \widetilde{\mathbf{x}}[j])^2\right]}{C^2\alpha^2} \le \frac{1}{\alpha^2} \tag{2}$$

$$P\left(\left|\sum_{i=1}^d \widetilde{\mathbf{x}}[i] - \sum_{i=1}^d \mathbf{x}[i]\right| \ge \beta D\right) \le \frac{1}{\beta^2} \tag{3}$$

For each positive integer $n$, positive number $k$, and $i = 2, \ldots, d$, define the set

$$G_{i,n,k} = \left\{ \widetilde{\mathbf{x}} \; : \; \begin{matrix} \widetilde{\mathbf{x}}[i] \in [k, k+2\alpha C], \\ \widetilde{\mathbf{x}}[1] \in [n-2\alpha C-k, n-k], \\ \sum_{j=1}^d \widetilde{\mathbf{x}}[j] \in [n-\beta D, n+\beta D] \end{matrix} \right\}$$

The intuition behind the meaning of $G_{i,n,k}$ is that suppose we had a dataset $\mathfrak{D}_i$ with vector $\mathbf{x}_i$ of point query answers where the $i^{\text{th}}$ point query satisfied $\mathbf{x}_i[i] = k + \alpha C$ and $\mathbf{x}_i[1] = n - k - \alpha C$ (all other entries are 0) then $G_{i,n,k}$ is the set of all possible outputs $M(\mathfrak{D}_i)$ where the $1^{\text{st}}$ and $i^{\text{th}}$ entries are within $\alpha C$ of their true value and the sum is within $\beta D$ of its true value.

For any fixed $n$ and $k$, we next examine how many $G_{i,n,k}$ a vector $\widetilde{\mathbf{x}}$ can belong to (i.e., an overlap condition). A necessary condition for $\widetilde{\mathbf{x}}$ to belong to some $G_{i,n,k}$ is that $\widetilde{\mathbf{x}}[i] \ge k$ and $\widetilde{\mathbf{x}}[1] \ge n - 2\alpha C - k$. This means that after assigning the minimal necessary mass to the $1^{\text{st}}$ element, there is at most $k + 2\alpha C + \beta D$ mass to assign to the other elements (without exceeding the upper limit of $n + \beta D$ on the sum of all the cells). Since at least $k$ units of this mass must be assigned to the $i^{\text{th}}$ element in order for $\widetilde{\mathbf{x}}$ to belong to $G_{i,n,k}$, this means that $\widetilde{\mathbf{x}}$ can belong to $G_{i,n,k}$ for at most $\frac{2\alpha C+\beta D+k}{k}$ different choices of $i$.

Now define $\mathbf{x}_1, \ldots, \mathbf{x}_d$ as follows. $\mathbf{x}_1[1] = n$ with all other entries being 0. Next for $i = 2, \ldots, d$ we set $\mathbf{x}_i[1] = n - \alpha C - k$ and $\mathbf{x}_i[i] = \alpha C + k$ and all other entries of $\mathbf{x}_i$ are 0. For each $i$, Let $\mathfrak{D}_i$ be

a database whose point query answers are $\mathbf{x}_i$, which is possible since the point queries are disjoint (and this means that $\mathfrak{D}_1$ differs from all of the others by the addition/removal of at least $2(\alpha C + k)$ records).

**For pure differential privacy**, we have:

$$1 \geq P\left(M(\mathfrak{D}_1) \in \bigcup_{i=2}^{d} G_{i,n,k}\right) \geq \frac{k}{2\alpha C + \beta D + k} \sum_{i=2}^{d} P\left(M(\mathfrak{D}_1) \in G_{i,n,k}\right) \text{ by overlap condition}$$

$$\geq e^{-\epsilon 2(\alpha C + k)} \frac{k}{2\alpha C + \beta D + k} \sum_{i=2}^{d} P\left(M(\mathfrak{D}_i) \in G_{i,n,k}\right) \text{ by group privacy property of } \epsilon\text{-DP [18]}$$

$$\geq e^{-\epsilon 2(\alpha C + k)} \frac{k}{2\alpha C + \beta D + k} \sum_{i=2}^{d} \left(1 - \frac{2}{\alpha^2} - \frac{1}{\beta^2}\right) \text{ by the Markov inequality and union bound}$$

$$= e^{-\epsilon 2(\alpha C + k)} \frac{k(d-1)}{2\alpha C + \beta D + k} \left(1 - \frac{2}{\alpha^2} - \frac{1}{\beta^2}\right)$$

Now we set $\alpha = 2$ and $\beta = 2$ to get

$$e^{2\epsilon(2C+k)} \geq \frac{k(d-1)}{16C + 8D + 4k}$$

If $D$ is allowed to be $\leq C$, then we set $k = C$ and get

$$e^{6\epsilon C} \geq \frac{d-1}{28} \qquad \Rightarrow \qquad C \geq \frac{1}{6\epsilon} \log \frac{d-1}{28}$$

In general, if $D \in O(C)$ (i.e., $D$ is allowed to be at most some constant times $C$) then similar arguments show $C \in \Omega\left(\frac{1}{\epsilon} \log(d)\right)$.

If $D$ is allowed to be $> C$ then we set $k = 1/\epsilon$ and get

$$e^{4\epsilon C + 2} \geq \frac{(d-1)}{24\epsilon D + 4} \qquad \Rightarrow \qquad C \geq \frac{1}{4\epsilon}\left(\log\left(\frac{(d-1)}{24\epsilon D + 4}\right) - 2\right)$$

In general, if $D \in \Omega(C)$ (i.e., $D$ is required to be at least some constant times $C$) then similar arguments show that $C \in \Omega\left(\frac{1}{\epsilon} \log(\frac{d}{\epsilon D})\right)$.

Putting these facts together, we see that if $D \in O(1/\epsilon)$ then $C \in \Omega(\frac{1}{\epsilon}\log(d))$. Meanwhile, if $C \in O(1/\epsilon)$ then we must have $D \in \Omega(d/\epsilon)$.

**For approximate, differential privacy**, using the group privacy property of approximate differential privacy [3],

$$1 \geq P\left(M(\mathfrak{D}_1) \in \bigcup_{i=2}^{d} G_{i,n,k}\right) \geq \frac{k}{2\alpha C + \beta D + k} \sum_{i=2}^{d} P\left(M(\mathfrak{D}_1) \in G_{i,n,k}\right)$$

$$\geq \frac{k}{2\alpha C + \beta D + k} \sum_{i=2}^{d} \left(e^{-\epsilon 2(\alpha C + k)} P\left(M(\mathfrak{D}_i) \in G_{i,n,k}\right) - \delta/\epsilon\right) \text{ by group privacy}$$

$$\geq \frac{k}{2\alpha C + \beta D + k} \sum_{i=2}^{d} \left(e^{-\epsilon 2(\alpha C + k)} \left(1 - \frac{2}{\alpha^2} - \frac{1}{\beta^2}\right) - \delta/\epsilon\right) \text{ Markov inequality, union bound}$$

$$= \frac{k(d-1)}{2\alpha C + \beta D + k} \left(e^{-\epsilon 2(\alpha C + k)} \left(1 - \frac{2}{\alpha^2} - \frac{1}{\beta^2}\right) - \delta/\epsilon\right)$$

Setting $\alpha = 2$ and $\beta = 2$ gives

$$1 \geq \frac{k(d-1)}{4C + 2D + k} \left(\frac{1}{4} e^{-\epsilon 2(2C+k)} - \delta/\epsilon\right) \text{ and so } \left(1 + \frac{\delta}{\epsilon} \frac{k(d-1)}{4C + 2D + k}\right) e^{4\epsilon C + 2k\epsilon} \geq \frac{1}{4} \frac{k(d-1)}{4C + 2D + k}$$

and this is the same as

$$\left(\frac{\delta}{\epsilon} + \frac{4C + 2D + k}{k(d-1)}\right) e^{4\epsilon C + 2k\epsilon} \geq 1/4$$

Noting that for any $z$, $1 + z \leq 2\max(1, z)$ and so

$$e^{4\epsilon C + 2k\epsilon} \geq \frac{1}{8} \min\left(\frac{k(d-1)}{4C + 2D + k}, \frac{\epsilon}{\delta}\right)$$

Proceeding as we did for pure differential privacy, if $D$ is allowed to be $O(C)$, then $C \in \Omega\left(\min\left(\frac{1}{\epsilon}\log(d), \frac{1}{\epsilon}\log\frac{\epsilon}{\delta}\right)\right)$; if $D$ is allowed to be $\Omega(C)$ then $C \in \Omega\left(\min(\frac{1}{\epsilon}\log\frac{d}{\epsilon D}, \frac{1}{\epsilon}\log\frac{\epsilon}{\delta})\right)$.

Putting this together, if $D \in O(1/\epsilon)$ then $C \in \Omega\left(\min\left(\frac{1}{\epsilon}\log(d), \frac{1}{\epsilon}\log\frac{\epsilon}{\delta}\right)\right)$ and if $C \in O(1/\epsilon)$ then either $\epsilon \in O(\delta)$ or $D \in \Omega(d/\epsilon)$.

**For $\rho$-zCDP**, consider a random variable $X$ that is uniform over $\mathfrak{D}_2, \ldots, \mathfrak{D}_d$ (i.e., with probability $1/(d-1)$, $X$ is the dataset $\mathfrak{D}_i$). Note that the $\mathfrak{D}_i$ we have been using can be constructed so that $i \neq j$, $\mathfrak{D}_i$ and $\mathfrak{D}_j$ differ on the addition/removal of $2aC + 2k$ people. Let $I(\cdot; \cdot)$ denote mutual information and $H(\cdot)$ denote entropy. By the group privacy property of zCDP [8] we have two facts relating group privacy to mutual information: (1) $\rho(2\alpha C + 2k)^2 \geq I(M(\mathfrak{D}_i); M(\mathfrak{D}_j))$ for all $i$ and $j$ (from Proposition 5.3 proof in [8]) and (2) the corollary that $\rho(2\alpha C + 2k)^2 \geq I(X, M(X))$ (from Proposition 6.1 proof in [8]). Then

$$\rho(2\alpha C + 2k)^2 \geq I(X; M(X)) = H(X) - H(X \mid M(X))$$
$$= \log_2(d-1) - H(X \mid M(X)) \tag{4}$$

and now we need to upper bound $H(X \mid M(X))$. Define $G$ to be the event that $M(X)$ is in the $G_{i,n,k}$ that corresponds to the realized value of $X$ (i.e., the event $X = \mathfrak{D}_j \Rightarrow M(X) \in G_{j,n,k}$ for $j = 2, \ldots, d$). Then we obtain a Fano-like inequality (following the proof structure in [12]) as follows:

$$H(X \mid M(X)) = H(X \mid M(X)) + H(G \mid X, M(X))$$

(the last entropy is 0 since $G$ is a deterministic function of $X$ and $M(X)$)

$$= H(G, X \mid M(X)) \quad \text{by the chain rule for conditional entropy}$$
$$= H(G \mid M(X)) + H(X \mid G, M(X)) \quad \text{by chain rule for conditional entropy}$$
$$\leq 1 + H(X \mid G, M(X)) \quad \text{since G is binary, its entropy is } \leq 1$$
$$= 1 + P(G = 0)H(X \mid M(X), G = 0) + P(G = 1)H(X \mid M(X), G = 1)$$
$$\leq 1 + P(G = 0)\log_2(d-1) + P(G = 1)H(X \mid M(X), G = 1)$$

(since the entropy of $X$ is $\leq \log_2(d-1)$ )

$$\leq 1 + P(G = 0)\log_2(d-1) + P(G = 1)\log_2\left(\frac{2\alpha C + \beta D + k}{k}\right)$$

(This follows from $G = 1$, by the overlap condition, since then $M(X)$ can

belong to at most $\dfrac{2\alpha C + \beta D + k}{k}$ of the $G_{i,n,k}$ so conditioned on knowing $M(X)$

there are at most $\dfrac{2\alpha C + \beta D + k}{k}$ possible choices for $X$

and hence $\log_2$ of this quantity upper bounds the conditional entropy)

$$\leq 1 + P(G = 0)\log_2(d-1) + \log_2\left(\frac{2\alpha C + \beta D + k}{k}\right)$$
$$\leq 1 + \left(\frac{2}{\alpha^2} + \frac{1}{\beta^2}\right)\log_2(d-1) + \log_2\left(\frac{2\alpha C + \beta D + k}{k}\right) \tag{5}$$

Where the last inequality follows from the Markov inequality and union bound on $P(G = 0)$. Now, setting $\alpha = \beta = 2$ and combining Equations 4 and 5, we have:

$$\rho(4C + 2k)^2 \geq \frac{1}{4} \log_2(d-1) - \log_2\left(\frac{4C + 2D + k}{k}\right) - 1$$

If $D$ is allowed to be $\leq C$, we set $k = C$ to get

$$\rho(6C)^2 \geq \frac{1}{4} \log_2(d-1) - \log_2\left(\frac{4C + 2D + C}{C}\right) - 1 \geq \frac{1}{4} \log_2(d-1) - \log_2\left(\frac{7C}{C}\right) - 1$$

$$\geq \frac{1}{4} \log_2(d-1) - 4 \Rightarrow C \geq \frac{1}{6\sqrt{\rho}} \sqrt{\frac{\log_2(d-1)}{4} - 4}$$

so in general, if $D \in O(C)$ then similar arguments show that $C \in \Omega\left(\sqrt{\frac{1}{\rho} \log(d)}\right)$.

If $D$ is allowed to be $> C$, then let $\gamma$ be any number strictly between 0 and 1. Then we set $k = 1/\sqrt{\rho}$, and $\alpha = \beta = \sqrt{\frac{3}{(1-\gamma)}}$. Combining Equations 4 and 5

$$(2\alpha\sqrt{\rho}C + 2)^2$$
$$\geq \left(1 - \frac{2}{\alpha^2} - \frac{1}{\beta^2}\right) \log_2(d-1) - \log_2\left(2\alpha\sqrt{\rho}C + \beta\sqrt{\rho}D + 1\right) - 1$$
$$= \gamma \log_2(d-1) - \log_2\left(2\alpha\sqrt{\rho}C + \beta\sqrt{\rho}D + 1\right) - 1$$
$$\geq \gamma \log_2(d-1) - \log_2\left(3\sqrt{\frac{3}{1-\gamma}}\sqrt{\rho}D + 1\right) - 1$$

which implies

$$C \geq \sqrt{\frac{1-\gamma}{12}} \sqrt{\frac{\gamma \log_2(d-1) - \log_2(3\sqrt{\frac{3}{1-\gamma}}\sqrt{\rho}D + 1) - 1}{\rho}} - \frac{1}{\sqrt{\rho}}\sqrt{\frac{1-\gamma}{3}}$$

In general, if $D$ is allowed to be $\Omega(C)$ then similar arguments show that $C \in \Omega\left(\sqrt{1-\gamma}\sqrt{\frac{\gamma \log_2(d-1) - \log_2(\sqrt{\frac{3}{1-\gamma}}\sqrt{\rho}D+1) - 1}{\rho}} - \frac{1}{\sqrt{\rho}}\sqrt{\frac{1-\gamma}{3}}\right)$.

Putting all of this together, if $D \in O(1/\sqrt{\rho})$ then $C \in \Omega\left(\sqrt{\log(d)/\rho}\right)$. But in order to get $C = O(1/\sqrt{\rho})$, we must have $D \in \Omega(d^\gamma/\sqrt{\rho})$.

$\square$

## A.2 Proof of Upper Bound Results

We first need some facts about Gaussian and Laplace random variables.

**Lemma 1.** *Let $z_1, \ldots, z_d$ be i.i.d. random variables from a distribution $F$.*

- *If $F$ is $N(0, \sigma^2)$ then*

  - $E[z_i^2] = \sigma^2$ *for all $i$*
  - $E[|z_i|] \leq \sigma$ *for all $i$*
  - $E[\max_i |z_i|] \in O(\sigma \sqrt{\log(d)})$
  - $E[\max_i z_i^2] \in O(\sigma^2 \log(d))$

- *If $F$ is $Lap(1/\epsilon)$ then*

  - $E[z_i^2] = 2/\epsilon^2$ *for all $i$*
  - $E[|z_i|] = 1/\epsilon$ *for all $i$*
  - $E[\max_i |z_i|] \leq \frac{1}{\epsilon}(\ln(d) + 1)$
  - $E[\max_i z_i^2] \leq \frac{1}{\epsilon^2}(\ln^2(d) + 2\ln(d) + 2)$

*Proof.* The variance of a Gaussian is known to be $\sigma^2$ and that of the Laplace distribution is known to be $2/\epsilon^2$.

The absolute value of a Laplace is an Exponential random variable with rate $\epsilon$ and so the expectation is $1/\epsilon$. Next, by Jensen's inequality $(E[|z_i|])^2 \leq E[z_i^2]$ and so $E[|z_i|] \leq \sqrt{(E[z_i^2])}$. Thus, in the case of a Gaussian, this is upper bounded by $\sigma$.

**To compute the expectation of the maxes**, we note that if $z_i'$ follows the $\text{Lap}(1)$ distribution, then $z'/\epsilon$ follows the $\text{Lap}(1/\epsilon)$ and if $z'$ follows $N(0,1)$ then $\sigma z$ follows the $N(0, \sigma^2)$ distribution. Thus we compute the expectations under the assumption that the scale variables are 1 and then we multiply by $1/\epsilon$ or $\sigma$ for the first moment, and $1/\epsilon^2$ or $\sigma^2$ for the second moment to get the results for $z_i$ from the results for $z_i'$.

Next we let $G$ be the cdf of a continuous nonnegative random variable and $g$ the corresponding pdf. Then for any $p \geq 1$,

$$E_{X \sim G}[X^p] = \int_0^\infty x^p g(x) \, dx = \int_0^\infty g(x) \left( \int_0^\infty p t^{p-1} 1_{\{t \leq x\}} \, dt \right) dx$$

$$= \int_0^\infty p t^{p-1} \left( \int_0^\infty g(x) 1_{\{t \leq x\}} \, dx \right) dt = \int_0^\infty p t^{p-1} (1 - G(t)) \, dt$$

Now we let $F_+$ be the cdf of $|z_1'|$ (the random variables with location parameter 1), and let $G$ be the distribution of $\max_i |z_i'|$. Then for all $t$, $G(t) = F_+(t)^d$ and also by the union bound, $1 - F_+(t)^d = 1 - G(t) \leq d(1 - F_+(t))$. For any $\gamma > 0$,

$$E\left[ \max_i |z_i'|^p \right] = \int_0^\infty p t^{p-1} (1 - F_+(t)^d) \, dt$$

$$= \int_0^\gamma p t^{p-1} (1 - F_+(t)^d) \, dt + \int_\gamma^\infty p t^{p-1} (1 - F_+(t)^d) \, dt$$

$$\leq \int_0^\gamma p t^{p-1} \, dt + \int_\gamma^\infty p t^{p-1} d(1 - F_+(t)) \, dt$$

$$= \gamma^p + \int_\gamma^\infty p t^{p-1} d(1 - F_+(t)) \, dt$$

**For the Laplace distribution**, $F_+(t) = 1 - e^{-t}$ thus, for any $\gamma > 0$

$$E\left[\max_i |z_i'|\right] \le \gamma + \int_\gamma^\infty de^{-t}\, dt = \gamma + de^{-\gamma}$$

$$E\left[\max_i |z_i'|^2\right] \le \gamma^2 + \int_\gamma^\infty 2tde^{-t}\, dt = \gamma^2 + 2\gamma de^{-\gamma} + 2de^{-\gamma}$$

Setting $\gamma = \ln(d)$ and converting from $z_i'$ to $z_i$, we get $E[\max_i |z_i|] \le \frac{1}{\epsilon}(\ln(d) + 1)$ and $E[\max_i |z_i|^2] \le \frac{1}{\epsilon^2}(\ln^2(d) + 2\ln(d) + 2)$.

**For the Gaussian distribution**, a well-known tail bound on the Gaussian is that $1 - F_+(t) \le \frac{2}{t}\frac{1}{\sqrt{2\pi}}e^{-t^2/2}$. Thus we get

$$E\left[\max_i |z_i'|\right] \le \gamma + \int_\gamma^\infty d\frac{2}{t}\frac{1}{\sqrt{2\pi}}e^{-t^2/2}\, dt \le \gamma + \frac{2d}{\gamma}\int_\gamma^\infty \frac{1}{\sqrt{2\pi}}e^{-t^2/2}\, dt$$

$$\le \gamma + 2d\frac{2}{\gamma^2}\frac{1}{\sqrt{2\pi}}e^{-\gamma^2/2}$$

$$E\left[\max_i |z_i'|^2\right] \le \gamma^2 + \int_\gamma^\infty 2td\frac{2}{t}\frac{1}{\sqrt{2\pi}}e^{-t^2/2}\, dt = \gamma^2 + 4d\int_\gamma^\infty \frac{1}{\sqrt{2\pi}}e^{-t^2/2}\, dt$$

$$= \gamma^2 + 4d\frac{1}{\gamma}\frac{1}{\sqrt{2\pi}}e^{-\gamma^2/2}$$

Setting $\gamma = \sqrt{2\ln(d)}$ and converting from $z_i'$ to $z_i$, we get $E[\max_i |z_i|] \le \sigma(\sqrt{2\ln(d)} + \frac{4}{\sqrt{2\pi}}\frac{1}{2\ln(d)})$ and $E[\max_i |z_i|^2] \le \sigma^2(2\ln(d) + \frac{4}{\sqrt{2\pi}}\frac{1}{\sqrt{2\ln(d)}})$. $\qquad\square$

We next need a technical lemma about the solution to a constrained nonnegative least squares problem.

**Lemma 2.** *Let $a_1, \ldots, a_d$ be real numbers and let $a_* \ge 0$. The solution to the optimization problem*

$$\arg\min_{x_1,\ldots,x_d} \frac{1}{2}\sum_{i=1}^d (x_i - a_i)^2$$

$$\text{s.t. } \sum_{i=1}^d x_i = a_*$$

$$x_i \ge 0, \text{ for } i = 1, \ldots, d$$

*is $x_i = \max\{a_i - \gamma, 0\}$ (for all $i$) where $\gamma$ is chosen so that $\sum_{i=1}^d \max\{0, a_i - \gamma\} = a_*$.*

*Proof.* Let us use the shorthand $(a - \gamma)_+$ to mean $\max\{0, a - \gamma\}$.

First, it is easy to see that by continuity, there exists a $\gamma$ such that $\sum_i (a_i - \gamma)_+ = a_*$.

The gradient of the objective function with respect to the $x_i$ is:

$$\frac{\partial \text{obj}}{\partial x_i} = (x_i - a_i) = (a_i - \gamma)_+ - a_i$$

and if this choice of $x_i$ is not optimal, then any descent direction $(y_1, \ldots, y_n)$ (i.e., for which $x_1 + y_1, \ldots, x_1 + y_2$ is feasible and reduces the objective function) must satisfy (1) $\sum_{i=1}^d y_i = 0$ to maintain feasibility of the equality constraint, (2) $\sum_{i=1}^d y_i((a_i - \gamma)_+ - a_i) < 0$ to be a descent direction, (3) $y_i \ge 0$ when $a_i \le \gamma$ and $y_i \ge \gamma - a_i$ when $a_i > \gamma$ to maintain nonnegativity of $x_i + y_i \equiv (a_i - \gamma)_+ + y_i$.

Now,

$$\sum_{i=1}^{d} y_i((a_i - \gamma)_+ - a_i) \sum_{i:a_i > \gamma} y_i((a_i - \gamma)_+ - a_i) + \sum_{i:a_i \le \gamma} y_i((a_i - \gamma)_+ - a_i)$$

$$= -\gamma \sum_{i:a_i > \gamma} y_i - \sum_{i:a_i \le \gamma} y_i a_i$$

$$= -\gamma \sum_{i:a_i > \gamma} y_i - \sum_{i:a_i \le \gamma} y_i \gamma$$

since feasibility of $x_i + y_i$ requires $y_i \ge 0$ when $a_i \le \gamma$

$$= -\gamma \sum_{i=1}^{d} y_i = 0 \quad \text{since feasibility requires } \sum_i y_i = 0$$

contradicting that $y_1, \ldots, y_d$ is a descent direction.

$\square$

**Theorem 4** (Upper bound for pure DP and zCDP). *Let $q_1, \ldots, q_d$ be a set of disjoint queries and let $q_*$ be their sum. Given privacy parameters $\epsilon > 0$ and $\rho > 0$, there exist algorithms $M_\epsilon, M_\rho, M'_\epsilon, M'_\rho, M'_{\epsilon,\delta}$ that output a positively weighted dataset and have the following properties:*

1. *$M_\epsilon$ satisfies $\epsilon$-DP, and for all $\mathfrak{D}$ and $i$, $E\left[(q_i(M_\epsilon(\mathfrak{D})) - q_i(\mathfrak{D}))^2\right] \le 2/\epsilon^2$ and $E\left[(q_*(M_\epsilon(\mathfrak{D})) - q_*(\mathfrak{D}))^2\right] \le 2d^2/\epsilon^2$.*

2. *$M_\rho$ satisfies $\rho$-zCDP, and for all $\mathfrak{D}$ and $i$, $E\left[(q_i(M_\rho(\mathfrak{D})) - q_i(\mathfrak{D}))^2\right] \le 1/(2\rho)$ and $E\left[(q_*(M_\rho(\mathfrak{D})) - q_*(\mathfrak{D}))^2\right] \le d^2/(2\rho)$.*

3. *$M'_\epsilon$ satisfies $\epsilon$-DP, and for all $\mathfrak{D}$ and $i$, $E\left[(q_i(M'_\epsilon(\mathfrak{D})) - q_i(\mathfrak{D}))^2\right] \in O(\log^2(d)/\epsilon^2)$ and $E\left[(q_*(M'_\epsilon(\mathfrak{D})) - q_*(\mathfrak{D}))^2\right] \in O(1/\epsilon^2)$*

4. *$M'_\rho$ satisfies $\rho$-zCDP, and for all $\mathfrak{D}$ and $i$, $E\left[(q_i(M'_\rho(\mathfrak{D})) - q_i(\mathfrak{D}))^2\right] \in O(\log(d)/\rho)$ and $E\left[(q_*(M'_\rho(\mathfrak{D})) - q_*(\mathfrak{D}))^2\right] \in O(1/\rho)$*

5. *$M'_{\epsilon,\delta}$ satisfies $(\epsilon, \delta)$-DP and for all $\mathfrak{D}$ and $i$, $E\left[(q_i(M'_{\epsilon,\delta}(\mathfrak{D})) - q_i(\mathfrak{D}))^2\right] \in O(\log^2(1/\delta)/\epsilon^2 + 1)$ and $E\left[(q_*(M'_{\epsilon,\delta}(\mathfrak{D})) - q_*(\mathfrak{D}))^2\right] \in O(1/\epsilon^2)$. Also note $M_\epsilon$ and $M'_\epsilon$ satisfy $\epsilon, \delta$-DP.*

*Proof.* The double-sided geometric mechanism $DGeo(\epsilon)$ is a discrete version of the Laplace distribution, supported over integers, with probability mass function $p(k) = \frac{1-e^{-\epsilon}}{1+e^{-\epsilon}} e^{-\epsilon|k|}$ [22]. It has several useful properties: (a) its mean is 0, (b) its variance is $2\frac{e^{-\epsilon}}{(1-e^{-\epsilon})^2} \le 2/\epsilon^2$, (c) given an integer-valued query $q$, adding $DGeo(\epsilon/\Delta_1(q))$ to its answer satisfies $\epsilon$-differential privacy.

Similarly, the discrete Gaussian $DGauss(0, 1/(2\rho))$ is a discrete version of the Gaussian distribution [9] with several useful properties: (a) its mean is 0, (b) its variance is less than that of $N(0, 1/(2\rho))$, (c) given an integer-valued query $q$, adding $DGauss(0, \Delta_2(q)^2/(2\rho))$ to its answer satisfies $\rho$-zcdp.

**To prove Item 1,** let $r_1, \ldots, r_d$ be records satisfying the predicates for point queries $q_1, \ldots, q_d$, respectively. Let $M_\epsilon$ be the algorithm that first computes nonnegative noisy query answers $a_i = \max\{0, q_i(\mathfrak{D}) + DGeo(1/\epsilon)\}$ for $i = 1, \ldots, d$ and then outputs the synthetic dataset $\widetilde{\mathfrak{D}}$ that has $a_i$ copies of record $r_i$ for each $i$. Note that $M_\epsilon$ does not obtain a noisy answer to $q_*$, and so it satisfies $\epsilon$-differential privacy since $\Delta_1(q_1, \ldots, q_d) = 1$ Since $q_i(\mathfrak{D}) \ge 0$ for all $i$, we have:

$$E\left[(q_i(\mathfrak{D}) - q_i(M_\epsilon(\mathfrak{D})))^2\right] = E\left[(q_i(\mathfrak{D}) - \max\{0, q_i(\mathfrak{D}) + DGeo(\epsilon)\})^2\right]$$

$$\le E\left[(q_i(\mathfrak{D}) - (q_i(\mathfrak{D}) + DGeo(\epsilon)))^2\right] \le 2/\epsilon^2$$

Furthermore

$$E\left[(q_*(\mathfrak{D}) - q_*(M_\epsilon(\mathfrak{D})))^2\right] = E\left[(\sum_i q_i(\mathfrak{D}) - \sum_i q_i(M_\epsilon(\mathfrak{D})))^2\right]$$

$$= \sum_i E\left[(q_i(\mathfrak{D}) - \max\{0, q_i(\mathfrak{D}) + DGeo(\epsilon)\})^2\right]$$

$$+ 2\sum_{i,j:i<j} E\left[\left(q_i(\mathfrak{D}) - \max\{0, q_i(\mathfrak{D}) + DGeo(\epsilon)\}\right)\right] E\left[\left(q_j(\mathfrak{D}) - \max\{0, q_j(\mathfrak{D}) + DGeo(\epsilon)\}\right)\right]$$

$$\leq d\frac{2}{\epsilon^2} + d(d-1)\frac{2}{\epsilon^2} = d^2\frac{2}{\epsilon^2}.$$

**To prove Item 2**, we use the same proofs as before, except that $M_\rho$ synthesizes $\widetilde{\mathfrak{D}}$ using the noisy answers $a_i = q_i(\mathfrak{D}) + \max\{0, DGauss(0, 1/(2\rho))\}$. Following essentially the same calculations, we see that the expected squared error of each point query $q_i$ is at most $1/(2\rho)$ and for the sum query $q_*$ it is at most $d^2/(2\rho)$.

**To prove Item 3**, let $r_1, \ldots, r_d$ be records satisfying the predicates for point queries $q_1, \ldots, q_d$, respectively. Let $M'_\epsilon$ be the algorithm that does the following. First, it obtains noisy answers for each query: $a_i = q_i(\mathfrak{D}) + Lap(2/\epsilon)$ for $i = 1, \ldots, d$ and $a_* = q_*(\mathfrak{D}) + Lap(2/\epsilon)$. (Since $\Delta_1(q_1, \ldots, q_d, q_*) = 2$, this clear satisfies $\epsilon$-differential privacy). Next, $M$ solves the following optimization problem:

$$\arg\min_{x_1,\ldots,x_d} \frac{1}{2}\sum_{i=1}^d (x_i - a_i)^2$$

$$\text{s.t. } \sum_{i=1}^d x_i = \max\{0, a_*\}$$

$$x_i \geq 0, \text{ for } i = 1, \ldots, d$$

and creates a privacy protected microdata $\widetilde{\mathfrak{D}}$ that consists of the records $r_1, \ldots, r_d$ with respective weights $x_1, \ldots, x_d$.

Since the sum query is nonnegative and the problem is constrained so that $\sum_i x_i$ is equal to $\max\{0, a_*\}$, clearly $E\left[(q_*(M'_\epsilon(\mathfrak{D})) - q_*(\mathfrak{D}))^2\right] \leq 2/\epsilon^2$.

Now let us derive an upper bound on $E\left[(q_i(M'_\epsilon(\mathfrak{D})) - q_i(\mathfrak{D}))^2\right]$ for a point query $q_i$.

For each $i$, let $z_i = a_i - q_i(\mathfrak{D})$ and $z_* = a_* - q_*(\mathfrak{D})$ be the actual noises that are added (they are all i.i.d. Laplace$(2/\epsilon)$).

We know from Lemma 2 that the solution $x_i$ have the form $\max\{a_i - \gamma, 0\}$ (which is $\max\{0, q_i(\mathfrak{D}) + z_i - \gamma\}$) for some $\gamma$ such that $\sum_i \max\{a_i - \gamma, 0\} = \max\{0, a*\}$ and note that the left hand side is monotonic in $\gamma$.

We first find a suitable upper and lower bound on $\gamma$. Define $L = -|z_*| + \min_i z_i$ and $U = |z_*| + \max_i z_i$. Then we have:

$$\sum_i \max\{0, a_i - U\} = \sum_i \max\{0, q_i(D) + z_i - U\} \leq \sum_i \max\{0, q_i(\mathfrak{D}) - |z_*|\}$$

$$\leq \max\{0, \left(\sum_i q_i(\mathfrak{D})\right) - |z_*|\}$$

$$\text{since the } q_i(\mathfrak{D}) \text{ are nonnegative}$$

$$= \max\{0, q_*(\mathfrak{D}) - |z_*|\}$$

$$\leq \max\{0, a_*\}$$

and so $\gamma \leq U$.

Next,

$$\sum_i \max\{0, a_i - L\} = \sum_i \max\{0, q_i(D) + z_i - L\} \geq \sum_i \max\{0, q_i(\mathfrak{D}) + |z_*|\}$$

$$= \sum_i (q_i(\mathfrak{D}) + |z_*|)$$

since the $q_i(\mathfrak{D})$ are nonnegative

$$\geq \left(\sum_i q_i(\mathfrak{D})\right) + |z_*|$$

$$= q_*(\mathfrak{D}) + |z_*| \geq \max\{0, a_*\}$$

and so $\gamma \leq L$.

We next find a bound on $E\left[(q_i(M'_\epsilon(\mathfrak{D})) - q_i(\mathfrak{D}))^2\right]$ in terms of $\gamma$.

$$E\left[(q_i(M'_\epsilon(\mathfrak{D})) - q_i(\mathfrak{D}))^2\right] = E\left[(\max\{0, q_i(\mathfrak{D}) + z_i - \gamma\} - q_i(\mathfrak{D}))^2\right]$$

note the random variable here are $z_i$ and $\gamma$

$$\leq E\left[((q_i(\mathfrak{D}) + z_i - \gamma) - q_i(\mathfrak{D}))^2\right]$$

since $q_i(\mathfrak{D})$ is nonnegative and removing the max moves the left part further away from $q_i(\mathfrak{D})$

$$= E\left[(z_i - \gamma)^2\right] \leq E\left[(|z_i| + \max\{|L|, |U|\})^2\right]$$

$$\leq E\left[\left(|z_i| + |z|_* + \max_j |z_j|\right)^2\right]$$

since the noises $z_j$ are symmetric around 0

$$\leq E\left[\left(|z|_* + 2\max_j |z_j|\right)^2\right]$$

$$= E\left[z_*^2\right] + 4E\left[|z_*|\right]E\left[\max_j |z_j|\right] + 4E\left[(\max_j |z_j|)^2\right] \tag{6}$$

$$\in O(\frac{1}{\epsilon^2}\log^2(d)) \quad \text{by Lemma 1 for Laplace noise}$$

**To prove Item 4** we follow the same steps as before, but using $N(0, 1/(\rho))$ noise instead of $\text{Lap}(2/\epsilon)$ (noting that $\Delta_2(q_1, \ldots, q_d, q_*) = \sqrt{2}$) and again see that the variance of the sum query is at most $1/\rho$, while for the point queries, the only thing that changes are the calculations after Equation 6, where we use the Lemma 1 results for Gaussian noise, to conclude that $E[(q_i(\mathfrak{D}) - q_i(M(\mathfrak{D})))^2] \in O(\frac{1}{\rho}\log(d))$ for each $i$.

**To prove Item 5**, we again follow the same steps as before but with a different noise distribution. Recall that the double geometric distribution $DGeo(\epsilon)$ is supported over the integers. If $z \sim DGeo(\epsilon)$ then $P(z = k) = \frac{1 - e^{-\epsilon}}{1 + e^{-\epsilon}}e^{-\epsilon|k|}$. Furthermore, if $k \geq 0$, $P(z \geq k) = P(z \leq -k) = \frac{1}{1 + e^{-\epsilon}}e^{-\epsilon k}$.

For any integer $B > 0$, the truncated double geometric distribution $TDGeo(\epsilon, B)$ is obtained by clipping a $DGeo(\epsilon)$ at $B$ and $-B$. Specifically, if $z' \sim TDGeo(\epsilon, B)$ then

$$P(z' = k) = \begin{cases} \frac{1}{1+e^{-\epsilon}}e^{-\epsilon B} & \text{if } k = B \\ \frac{1-e^{-\epsilon}}{1+e^{-\epsilon}}e^{-\epsilon|k|} & \text{for } k = -B+1, \ldots, B-1 \\ \frac{1}{1+e^{-\epsilon}}e^{-\epsilon B} & \text{if } k = -B \end{cases}$$

So, we follow the same approach as in the proof of Item 3 but we use $TDGeo(\epsilon/2, B)$ noise to answer each query (detail queries and sum query). We first determine the value of $B$ needed to satisfy $(\epsilon/2, \delta/2)$-DP.

First note that for any integer $v$, and integer $k \in [v - B + 1, v + B - 2]$

$$e^{-\epsilon/2} \leq \frac{P(v + TDGeo(\epsilon/2, B) = k)}{P(v - 1 + TDGeo(\epsilon/2, B) = k)} \leq e^{\epsilon/2}$$

(the significance of these points are that they are not in the boundary of $v + TDGeo$ or $v - 1 + TDGeo$).

Meanwhile $P(v + TDGeo(\epsilon/2) \in \{v - B, v + B - 1, v + B\}) = P(DGeo(\epsilon/2) \geq B - 1) + P(DGeo(\epsilon/2) \leq -B) = \frac{1}{1+e^{-\epsilon/2}}e^{-\epsilon B/2} + \frac{1}{1+e^{-\epsilon/2}}e^{-\epsilon(B-1)/2} \leq 2e^{-\epsilon(B-1)/2}$. These are the boundary points where the probability ratios may be large.

Setting this equal to $\delta/2$ (and then performing similar calculations when the $v - 1$ term is in the numerator), we see that adding $TDGeo(\epsilon/2, B)$ noise satisfies $\epsilon, \delta$-DP if $B \geq \frac{2}{\epsilon} \log(4/\delta) + 1$.

Thus using a naive composition result of approximate differential privacy [18], we can add $TGeo(\epsilon/2, B)$ noise to each point query and the sum query to satisfy $(\epsilon, \delta)$-DP.

Using the same postprocessing as in the proof of Item 3, we see that the expected squared error of the sum query (when computed from the postprocessed privacy-protected data) is at most variance$(TDGeo(\epsilon/2, B)) \leq$ variance$(DGeo(\epsilon/2)) \leq 8/\epsilon^2$.

For the point queries, the only thing that changes are the calculations after Equation 6. Since the absolute value of the noises is bounded by $B$, we get that the expected squared error of the point queries is $\in O(B^2) = O(\frac{1}{\epsilon^2} \log^2(1/\delta) + 1)$.

$\square$

## B  Full Data Benchmark Description

Our benchmarks contain 15 real datasets and 16 synthetic datasets. The datasets are designed to be small enough to enable thousands of runs (in order to compute expected squared errors) but large enough to clearly illustrate postprocessing errors and present a challenge to many open-source optimizers.

### B.1  Real Datasets

The real datasets are drawn from the 2016 American Community Survey Public Use Microdata Sample (PUMS) [39], which provides records for geographies known as Public Use Microdata Areas (PUMA).

To create a benchmark data set that adequately captured the diversity of real world demographic data, we drew from outlier geographies in the 2016 ACS PUMS. We chose 15 Public Use Microdata Areas whose data distributions had been identified as conflicting significantly with the majority distributions in their states, according to the k-marginal metric used by NIST in their Differential Privacy challenge [36]. The data spanned historically redlined areas, a variety of immigrant communities, wealthy and diverse urban neighborhoods, rural agricultural communities, and included every major region in the United States.

For each of the 15 regions, we created a $9 \times 24$ Race by Hispanic Origin histogram. These were two separate questions in the ACS questionnaire. Although the questionnaire allowed respondents to select multiple races (from a list of 15 categories and 3 fill-in text boxes), most individuals belong to three or fewer races, and the 2016 ACS PUMS did not include detailed racial breakdowns for individuals with more than 3 races. To mimic the extreme sparsity and geographically diverse correlation patterns in the multi-racial checkbox variable, we selected two variables (called RAC1P and HISP; full definitions below): a smaller race variable with 9 possible values which primarily records single races, and a detailed Hispanic origin variable with 24 possible values. Any of the 216 possible combinations of race and Hispanic origin is valid; individuals of all races have origins from all across Latin America. However, in any given community the vast majority of these counts will be zero, resulting in sparse distributions. At the same time, communities with different immigration histories will differ significantly with respect to which counts are nonzero and in the size of the other counts. Algorithms which performed well across all cases in the PUMS benchmark data set should be expected perform well on the edge case complexities of national data.

```
RAC1P
    Recoded detailed race code
1. White alone
2. Black or African American alone
3. American Indian alone
4. Alaska Native alone
5. American Indian and Alaska Native tribes specified; or American
 Indian or Alaska Native, not specified and no other races
6. Asian alone
7. Native Hawaiian and Other Pacific Islander alone
8. Some Other Race alone
9. Two or More Races

HISP
Detailed Hispanic origin
01. Not Spanish/Hispanic/Latino
02. Mexican
03. Puerto Rican
04. Cuban
05. Dominican
06. Costa Rican
07. Guatemalan
08. Honduran
09. Nicaraguan
10. Panamanian
11. Salvadoran
12. Other Central American
13. Argentinean
14. Bolivian
15. Chilean
16. Colombian
17. Ecuadorian
18. Paraguayan
19. Peruvian
20. Uruguayan
21. Venezuelan
22. Other South American
23. Spaniard
24. All Other Spanish/Hispanic/Latino
```

## B.2   Synthetic Data

The synthetic data are modeled after the proofs of our lower bound results. The main idea is that suppose noise from a distribution $F$ is added to a histogram, and that there are $k$ zero cells and one cell with a count of $C$ in that histogram. Based on the noisy cell values, it is difficult to guess which cell had value $C$ when $C$ is smaller than the median of the distribution of $\max\{X_1, \ldots, X_k\}$ (whose CDF is $F^k(t)$), where each $X_i \sim F$. Thus we created datasets with sparsity patterns.

Each histogram had 100 elements, from which we created a 1-dimensional version (a 100-element vector) and a 2-dimensional version (reshaping it to a $10 \times 10$ histogram). In all of the datasets, the first histogram cell is relatively large (10,000) and should be easy to distinguish from 0 based on the noisy counts (although ordinary nonnegative least squares fails to do so).

The synthetic histograms come from 4 categories, defined as follows:

- **Level**. In the Level **k** histograms, all cells have the same value $k$ (except the first, which has value 10,000). The benchmarks include 1- and 2-dimensional versions of Level0 (i.e., only the first element is nonzero), Level1, Level16, and Level32. The Level1 dataset presents a tricky case where each cell (other than the first), based on its noisy value, may look similar to 0, but the overall sum of these small cells is clearly distinguishable from 0. The Level16

and Level32 datasets are designed to force algorithms to try to estimate the number of cells that are likely to have true value of 0. Note that 16 is roughly the 40th percentile of the distribution of $\max X_1, \ldots, X_{100}$ when each $X_i$ has the Laplace$(1/\epsilon)$ distribution with $\epsilon = 0.25$. So having a few cell noisy cell counts near 16 is possible when a histogram is mostly 0, but having many noisy counts near 16 is a sign that the histogram is not sparse.

- **Stair**. The Stair data is a histogram that looks like this: $[10000, 1, 2, 3, 4, \ldots]$ in one dimension (and is reshaped into a $10 \times 10$ matrix in 2 dimensions. It is designed to simulate a dataset with small, medium, and large values.

- **Step**. The Step k dataset is a step function. The first element is 10000, the next 49 are 0 and the last 50 are $k$. This is an interpolation between the sparse dataset synthetic dataset Level0 and Level $k$. For our benchmark, we use Step16 (i.e., $k = 16$) as a dataset of medium difficulty and Step50 as an easy dataset.

- **SplitStairs**. The SplitStairs dataset is an interpolation between Stair and a very sparse dataset. The first half looks like the Stair dataset but cells 50 until the end all have value 0. This ensures that all true cell counts that can be dominated by 50 random zero-mean Laplace random variables are represented in the dataset.

Combined, these synthetic datasets give 8 1-dimensional histograms (4 Level, 1 Stair, 2 Step, 1 SplitStairs) and 8 2-dimensional histograms.


# Complete experimental results.

In this document, we present our full experimental results. The datasets used are the PUMS datasets (2-dimensional), the 1-dimensional synthetic data, and the 2-dimensional synthetic data. These datasets are described in the appendix of the full version of the paper, which appears in the supplementary material file.

For the one-dimensional datasets, we use either the Laplace mechanism (for pure differential privacy) or the Gaussian mechanism (for zCDP) to obtain noisy answers to:

- The sum query (the sum of the histogram cells)
- The identity queries (the count in each cell).

For the two-dimensional datasets, we use either the Laplace or Gaussian mechanisms to obtain noisy answers to:

- The sum query (the sum of the histogram cells)
- The identity queries (the count in each cell).
- The marginal on the first dimension.
- The marginal on the second dimension.

We use the NNLS (referred to as nnlsalg in the tables), Max fitting, Sequential Fitting, and Weighted Fitting (with confidence parameter 0.99) postprocessing methods to obtain the privacy preserving positively weighted data $\widetilde{D}$. Sequential Fitting prioritizes queries in the order listed above. We also use OLS fitting (NNLS fitting without the nonnegativity constraints), which is referred to as olsalg in the experiments. The OLS fitting method is known to improve the squared error of the queries compared to the original noisy answers (this is a consequence of the Gauss-Markov theorem) but does not result in a positively weighted dataset. Hence the goal of the methods is not to do much worse than the OLS fitting method.

The code was written in Julia. In order to make the code fully open source, we experimented with several open source solvers compatible with Julia's JuMP framework. Out of these, the COSMO solver performed the best. However, the relatively complex multi-stage optimizations in Max fitting and Sequential fitting caused problems. In some cases the solver claimed infeasibility for problems in latter stages of the optimization (likely due to poor quality solutions in earlier stages), numerical errors, or slow convergence (hitting the iteration limit). To reduce the chance of poor solutions in earlier stages of an optimization, we set the absolute and relative tolerances to 1e-7 and an iteration limit of 20,000, which is 4 times the default. We also converted equality constraints of the form $x = constant$ to $x \leq constant + 0.001$ and $x \geq constant - 0.001$. For the Max Fitting solve, after it gets an $L_\infty$ distance estimate in the first stage of the solve, we added a slack of 0.01 to this distance to prevent it from failing in the second stage.

Despite tuning parameter and setting slack tolerances to equalities and inequalities, not all runs were successful, so we only kept the ones where all stages of the optimization were optimal. This likely optimistically biased the results of Max fitting and Sequential fitting and increased their estimated standard errors.

These optimization problems did not affect OLS, NNLS, or the Weighted Fitting approaches.

Each experiment is an average over 1000 runs (thus the expected error of a query is estimated the average of its errors across 1000 runs). However, for more complex constrained methods, the average was among fewer runs if some stage of the multi-stage optimization failed to find an optimal solution.

In each table, we evaluate the error of different queries.

- For the Sum query (as in Table 2), we display its expected error along with estimated standard deviation.
- For the Identity queries (as in Table 1), each cell $i$ in the histogram corresponds to a query $q_i$ (the count in that cell). For each cell $i$, we estimate its expected squared error $e_i = E[((q_i \mathfrak{D}) - q_i(\widetilde{\mathfrak{D}}))^2]$ by averaging the error across trials. Then we report $\max_i e_i$ and $\sum_i e_i$ along with standard errors. Again, we emphasize that our Max metric is $\max_i E[((q_i \mathfrak{D}) - q_i(\widetilde{\mathfrak{D}}))^2]$ and not outlier error $E[\max_i(((q_i \mathfrak{D}) - q_i(\widetilde{\mathfrak{D}})))^2]$.
- For the two dimensional datasets, we also have tables for each marginal and report the max and total squared errors as for the identity queries.

Note that the goal is to avoid extreme errors that are much larger than the OLS error.

The experiments are organized first by privacy definition (pure DP and zCDP). Within each privacy definition, we first present results for the 1-dimensional synthetic data (for 3 privacy parameters) followed

by the 2-dimensional synthetic data (for 3 privacy parameters) followed by the PUMS data (for 3 privacy parameters).

## 1. Pure Differential Privacy

| Dataset | olsalg | | nnlsalg | | maxalg | | seqalg | | weightalg | |
|---|---|---|---|---|---|---|---|---|---|---|
| | Total | Max | Total | Max | Total | Max | Total | Max | Total | Max |
| Level00-1d | 789.4 | 9.2 | 61.4 | 28.8 | 64.5 | 30.0 | 57.8 | 39.5 | 11.4 | 6.2 |
| | ±5.5 | ±0.6 | ±1.4 | ±0.9 | ±1.9 | ±1.2 | ±1.3 | ±1.1 | ±0.7 | ±0.5 |
| Level01-1d | 789.4 | 9.2 | 300.9 | 8.8 | 303.5 | 9.0 | 298.8 | 8.9 | 296.4 | 8.0 |
| | ±5.5 | ±0.6 | ±3.2 | ±0.6 | ±4.2 | ±0.8 | ±3.2 | ±0.6 | ±3.2 | ±0.6 |
| Level16-1d | 789.4 | 9.2 | 788.0 | 9.2 | 796.9 | 10.4 | 788.0 | 9.2 | 783.2 | 9.2 |
| | ±5.5 | ±0.6 | ±5.5 | ±0.6 | ±10.1 | ±1.5 | ±5.5 | ±0.6 | ±5.4 | ±0.6 |
| Level32-1d | 789.4 | 9.2 | 789.4 | 9.2 | 791.9 | 11.7 | 789.4 | 9.2 | 789.4 | 9.2 |
| | ±5.5 | ±0.6 | ±5.5 | ±0.6 | ±11.9 | ±2.7 | ±5.5 | ±0.6 | ±5.5 | ±0.6 |
| SplitStairs-1d | 789.4 | 9.2 | 535.1 | 9.5 | 528.9 | 9.5 | 535.0 | 9.5 | 519.8 | 13.1 |
| | ±5.5 | ±0.6 | ±4.5 | ±0.7 | ±5.8 | ±0.8 | ±4.5 | ±0.7 | ±4.4 | ±0.6 |
| Stair-1d | 789.4 | 9.2 | 779.2 | 9.1 | 778.7 | 11.0 | 779.2 | 9.1 | 781.3 | 9.1 |
| | ±5.5 | ±0.6 | ±5.5 | ±0.7 | ±10.2 | ±1.8 | ±5.5 | ±0.7 | ±5.5 | ±0.7 |
| Step16-1d | 789.4 | 9.2 | 560.2 | 9.6 | 555.4 | 10.1 | 560.2 | 9.6 | 644.4 | 11.8 |
| | ±5.5 | ±0.6 | ±4.6 | ±0.7 | ±6.2 | ±0.9 | ±4.6 | ±0.7 | ±4.9 | ±0.7 |
| Step50-1d | 789.4 | 9.2 | 561.5 | 9.6 | 563.9 | 10.1 | 561.5 | 9.6 | 427.2 | 9.1 |
| | ±5.5 | ±0.6 | ±4.7 | ±0.7 | ±6.4 | ±1.0 | ±4.7 | ±0.7 | ±4.2 | ±0.7 |

Table 1. Squared Errors (with standard deviations). Id Query. 1-d datasets. Lap Mechanism ($\epsilon = 1$).

| Dataset | olsalg | nnlsalg | maxalg | seqalg | weightalg |
|---|---|---|---|---|---|
| Level00-1d | 7.7 | 29.7 | 31.7 | 7.8 | 6.8 |
| | ±0.5 | ±0.9 | ±1.3 | ±0.5 | ±0.5 |
| Level01-1d | 7.7 | 8.8 | 9.4 | 7.8 | 7.7 |
| | ±0.5 | ±0.5 | ±0.7 | ±0.5 | ±0.5 |
| Level16-1d | 7.7 | 7.7 | 6.9 | 7.8 | 7.7 |
| | ±0.5 | ±0.5 | ±0.8 | ±0.5 | ±0.5 |
| Level32-1d | 7.7 | 7.7 | 8.3 | 7.8 | 7.7 |
| | ±0.5 | ±0.5 | ±1.0 | ±0.5 | ±0.5 |
| SplitStairs-1d | 7.7 | 8.4 | 8.4 | 7.8 | 7.7 |
| | ±0.5 | ±0.5 | ±0.7 | ±0.5 | ±0.5 |
| Stair-1d | 7.7 | 7.7 | 7.6 | 7.8 | 7.7 |
| | ±0.5 | ±0.5 | ±0.8 | ±0.5 | ±0.5 |
| Step16-1d | 7.7 | 8.3 | 8.4 | 7.8 | 7.7 |
| | ±0.5 | ±0.5 | ±0.7 | ±0.5 | ±0.5 |
| Step50-1d | 7.7 | 8.3 | 8.9 | 7.8 | 7.8 |
| | ±0.5 | ±0.5 | ±0.7 | ±0.5 | ±0.5 |

Table 2. Squared Error (with standard deviations). Sum Query. 1-d datasets. Lap Mechanism ($\epsilon = 1$).

| Dataset | olsalg | | nnlsalg | | maxalg | | seqalg | | weightalg | |
|---|---|---|---|---|---|---|---|---|---|---|
| | Total | Max | Total | Max | Total | Max | Total | Max | Total | Max |
| Level00-1d | 3157.5 | 37.0 | 245.7 | 115.2 | 250.2 | 121.2 | 231.2 | 158.1 | 45.0 | 24.8 |
| | ±22.2 | ±2.3 | ±5.5 | ±3.7 | ±8.4 | ±5.8 | ±5.1 | ±4.3 | ±2.9 | ±1.8 |
| Level01-1d | 3157.5 | 37.0 | 744.2 | 43.6 | 733.0 | 42.6 | 723.9 | 44.6 | 691.9 | 31.9 |
| | ±22.2 | ±2.3 | ±10.5 | ±2.5 | ±13.2 | ±2.9 | ±10.3 | ±2.5 | ±10.4 | ±2.2 |
| Level16-1d | 3157.5 | 37.0 | 3016.8 | 35.6 | 3113.9 | 38.8 | 3016.8 | 35.6 | 3021.0 | 35.6 |
| | ±22.2 | ±2.3 | ±19.3 | ±2.1 | ±31.7 | ±3.8 | ±19.3 | ±2.1 | ±19.4 | ±2.1 |
| Level32-1d | 3157.5 | 37.0 | 3151.8 | 37.0 | 3145.6 | 40.3 | 3151.8 | 37.0 | 3131.8 | 36.7 |
| | ±22.2 | ±2.3 | ±21.8 | ±2.3 | ±37.3 | ±5.0 | ±21.8 | ±2.3 | ±21.7 | ±2.3 |
| SplitStairs-1d | 3157.5 | 37.0 | 2053.5 | 37.5 | 2053.1 | 39.1 | 2053.0 | 37.6 | 2126.2 | 45.9 |
| | ±22.2 | ±2.3 | ±17.0 | ±2.5 | ±21.6 | ±3.1 | ±17.0 | ±2.5 | ±17.7 | ±2.7 |
| Stair-1d | 3157.5 | 37.0 | 3057.2 | 36.3 | 3063.7 | 40.2 | 3057.2 | 36.3 | 3074.2 | 36.5 |
| | ±22.2 | ±2.3 | ±21.4 | ±2.7 | ±31.8 | ±4.8 | ±21.4 | ±2.7 | ±21.5 | ±2.5 |
| Step16-1d | 3157.5 | 37.0 | 2142.9 | 35.8 | 2153.2 | 36.5 | 2142.5 | 35.8 | 2161.2 | 36.6 |
| | ±22.2 | ±2.3 | ±16.4 | ±2.3 | ±20.9 | ±2.7 | ±16.4 | ±2.3 | ±17.0 | ±2.5 |
| Step50-1d | 3157.5 | 37.0 | 2245.9 | 38.4 | 2258.7 | 38.8 | 2245.9 | 38.5 | 1814.2 | 41.3 |
| | ±22.2 | ±2.3 | ±19.0 | ±2.7 | ±24.8 | ±3.3 | ±18.9 | ±2.7 | ±22.1 | ±4.1 |

TABLE 3. Squared Errors (with standard deviations). Id Query. 1-d datasets. Lap Mechanism ($\epsilon = 0.5$).

| Dataset | olsalg | nnlsalg | maxalg | seqalg | weightalg |
|---|---|---|---|---|---|
| Level00-1d | 30.9 | 118.9 | 116.6 | 31.3 | 26.9 |
| | ±1.9 | ±3.7 | ±5.7 | ±1.9 | ±1.8 |
| Level01-1d | 30.9 | 44.2 | 43.8 | 31.2 | 31.0 |
| | ±1.9 | ±2.4 | ±3.1 | ±1.9 | ±1.9 |
| Level16-1d | 30.9 | 30.9 | 34.7 | 31.2 | 31.0 |
| | ±1.9 | ±1.9 | ±3.3 | ±1.9 | ±1.9 |
| Level32-1d | 30.9 | 30.9 | 33.3 | 31.2 | 30.9 |
| | ±1.9 | ±1.9 | ±3.7 | ±1.9 | ±1.9 |
| SplitStairs-1d | 30.9 | 33.8 | 33.7 | 31.2 | 31.0 |
| | ±1.9 | ±2.1 | ±2.6 | ±1.9 | ±1.9 |
| Stair-1d | 30.9 | 30.9 | 33.9 | 31.2 | 30.9 |
| | ±1.9 | ±1.9 | ±3.2 | ±1.9 | ±1.9 |
| Step16-1d | 30.9 | 33.5 | 30.5 | 31.2 | 31.0 |
| | ±1.9 | ±2.1 | ±2.3 | ±1.9 | ±1.9 |
| Step50-1d | 30.9 | 33.4 | 31.9 | 31.2 | 31.1 |
| | ±1.9 | ±2.1 | ±2.5 | ±1.9 | ±1.9 |

TABLE 4. Squared Error (with standard deviations). Sum Query. 1-d datasets. Lap Mechanism ($\epsilon = 0.5$).

| Dataset | olsalg | | nnlsalg | | maxalg | | seqalg | | weightalg | |
|---|---|---|---|---|---|---|---|---|---|---|
| | Total | Max | Total | Max | Total | Max | Total | Max | Total | Max |
| Level00-1d | 78938.2 | 924.2 | 6142.8 | 2879.9 | 5628.8 | 2631.4 | 5787.0 | 3953.1 | 1121.6 | 617.5 |
| | ±554.5 | ±58.3 | ±138.0 | ±93.5 | ±328.5 | ±232.4 | ±127.3 | ±107.5 | ±72.4 | ±46.0 |
| Level01-1d | 78938.2 | 924.2 | 8021.2 | 2029.4 | 8061.8 | 2028.1 | 7109.2 | 2344.4 | 4107.4 | 787.6 |
| | ±554.5 | ±58.3 | ±167.0 | ±80.8 | ±404.4 | ±175.3 | ±151.5 | ±86.2 | ±127.9 | ±54.3 |
| Level16-1d | 78938.2 | 924.2 | 39776.1 | 826.1 | 39237.6 | 839.7 | 39689.7 | 827.1 | 39637.2 | 797.0 |
| | ±554.5 | ±58.3 | ±348.5 | ±56.2 | ±609.1 | ±116.8 | ±347.6 | ±56.2 | ±351.9 | ±55.8 |
| Level32-1d | 78938.2 | 924.2 | 56529.7 | 798.5 | 57415.6 | 781.4 | 56511.8 | 798.6 | 56541.6 | 797.0 |
| | ±554.5 | ±58.3 | ±385.1 | ±55.7 | ±669.3 | ±94.1 | ±384.9 | ±55.7 | ±387.8 | ±55.8 |
| SplitStairs-1d | 78938.2 | 924.2 | 34261.9 | 929.3 | 33432.3 | 1284.1 | 34129.4 | 935.2 | 34009.3 | 797.0 |
| | ±554.5 | ±58.3 | ±310.5 | ±58.8 | ±627.5 | ±178.6 | ±308.6 | ±58.9 | ±317.7 | ±55.8 |
| Stair-1d | 78938.2 | 924.2 | 62862.6 | 884.3 | 62820.4 | 966.5 | 62856.9 | 885.2 | 63033.7 | 892.8 |
| | ±554.5 | ±58.3 | ±429.5 | ±62.7 | ±726.0 | ±126.3 | ±429.6 | ±62.7 | ±435.1 | ±64.1 |
| Step16-1d | 78938.2 | 924.2 | 27709.8 | 962.5 | 28196.6 | 1199.7 | 27460.6 | 971.2 | 27113.5 | 797.0 |
| | ±554.5 | ±58.3 | ±291.7 | ±59.5 | ±773.9 | ±207.2 | ±289.0 | ±59.7 | ±294.4 | ±55.8 |
| Step50-1d | 78938.2 | 924.2 | 47908.3 | 848.4 | 48145.1 | 889.8 | 47873.6 | 850.2 | 47937.5 | 797.0 |
| | ±554.5 | ±58.3 | ±360.7 | ±57.0 | ±617.2 | ±85.8 | ±359.9 | ±57.1 | ±367.2 | ±55.8 |

TABLE 5. Squared Errors (with standard deviations). Id Query. 1-d datasets. Lap Mechanism ($\epsilon = 0.1$).

| Dataset | olsalg | nnlsalg | maxalg | seqalg | weightalg |
|---|---|---|---|---|---|
| Level00-1d | 771.8 | 2972.2 | 2857.8 | 782.8 | 670.3 |
| | ±47.6 | ±93.3 | ±246.6 | ±48.7 | ±46.2 |
| Level01-1d | 771.8 | 2090.0 | 1846.9 | 780.9 | 780.0 |
| | ±47.6 | ±80.3 | ±149.2 | ±48.5 | ±47.8 |
| Level16-1d | 771.8 | 817.0 | 861.1 | 780.9 | 774.3 |
| | ±47.6 | ±50.4 | ±89.6 | ±48.5 | ±48.0 |
| Level32-1d | 771.8 | 780.5 | 779.3 | 780.9 | 774.3 |
| | ±47.6 | ±48.5 | ±74.1 | ±48.5 | ±48.0 |
| SplitStairs-1d | 771.8 | 927.5 | 819.0 | 780.9 | 774.3 |
| | ±47.6 | ±54.5 | ±107.5 | ±48.5 | ±48.0 |
| Stair-1d | 771.8 | 780.0 | 732.1 | 781.5 | 774.3 |
| | ±47.6 | ±48.5 | ±81.3 | ±48.5 | ±48.0 |
| Step16-1d | 771.8 | 970.1 | 932.6 | 780.9 | 774.3 |
| | ±47.6 | ±56.0 | ±138.7 | ±48.5 | ±48.0 |
| Step50-1d | 771.8 | 847.9 | 822.8 | 780.9 | 774.3 |
| | ±47.6 | ±52.0 | ±82.5 | ±48.5 | ±48.0 |

TABLE 6. Squared Error (with standard deviations). Sum Query. 1-d datasets. Lap Mechanism ($\epsilon = 0.1$).

| Dataset | olsalg | | nnlsalg | | maxalg | | seqalg | | weightalg | |
|---|---|---|---|---|---|---|---|---|---|---|
| | Total | Max | Total | Max | Total | Max | Total | Max | Total | Max |
| Level00-2d | 264.7 | 31.2 | 75.5 | 18.9 | 87.2 | 23.9 | 88.1 | 51.4 | 29.7 | 12.7 |
| | ±5.2 | ±1.9 | ±1.9 | ±0.9 | ±10.2 | ±5.1 | ±6.2 | ±4.8 | ±1.7 | ±0.7 |
| Level01-2d | 264.7 | 31.2 | 190.6 | 22.0 | 180.3 | 20.5 | 253.9 | 30.4 | 245.7 | 28.1 |
| | ±5.2 | ±1.9 | ±3.5 | ±1.3 | ±7.3 | ±2.8 | ±5.8 | ±2.3 | ±5.3 | ±1.8 |
| Level16-2d | 264.7 | 31.2 | 264.1 | 31.2 | 276.9 | 32.1 | 291.3 | 34.5 | 290.8 | 34.4 |
| | ±5.2 | ±1.9 | ±5.2 | ±1.9 | ±11.6 | ±4.6 | ±6.0 | ±2.2 | ±6.0 | ±2.2 |
| Level32-2d | 264.7 | 31.2 | 264.7 | 31.2 | 278.1 | 31.4 | 291.5 | 34.6 | 266.8 | 31.6 |
| | ±5.2 | ±1.9 | ±5.2 | ±1.9 | ±10.3 | ±3.6 | ±6.0 | ±2.2 | ±5.3 | ±1.9 |
| SplitStairs-2d | 264.7 | 31.2 | 241.9 | 28.3 | 245.0 | 28.9 | 290.7 | 33.9 | 289.4 | 34.3 |
| | ±5.2 | ±1.9 | ±4.6 | ±1.7 | ±7.3 | ±2.5 | ±6.2 | ±2.2 | ±6.0 | ±2.2 |
| Stair-2d | 264.7 | 31.2 | 264.0 | 31.1 | 273.6 | 31.9 | 291.7 | 34.6 | 287.8 | 34.3 |
| | ±5.2 | ±1.9 | ±5.2 | ±1.9 | ±14.0 | ±4.7 | ±6.0 | ±2.2 | ±6.0 | ±2.2 |
| Step16-2d | 264.7 | 31.2 | 246.6 | 28.5 | 247.1 | 27.7 | 293.1 | 33.8 | 290.8 | 34.5 |
| | ±5.2 | ±1.9 | ±4.8 | ±1.7 | ±11.2 | ±4.1 | ±6.3 | ±2.2 | ±6.0 | ±2.2 |
| Step50-2d | 264.7 | 31.2 | 247.4 | 28.6 | 240.2 | 30.3 | 291.6 | 34.9 | 246.5 | 28.3 |
| | ±5.2 | ±1.9 | ±4.8 | ±1.7 | ±10.2 | ±4.6 | ±6.0 | ±2.2 | ±4.9 | ±1.7 |

TABLE 7. Squared Errors (with standard deviations). Marg1 Query. 2-d datasets. Lap Mechanism ($\epsilon = 1$).

| Dataset | olsalg | | nnlsalg | | maxalg | | seqalg | | weightalg | |
|---|---|---|---|---|---|---|---|---|---|---|
| | Total | Max | Total | Max | Total | Max | Total | Max | Total | Max |
| Level00-2d | 253.7 | 26.7 | 73.6 | 18.3 | 88.1 | 26.3 | 76.0 | 45.4 | 31.0 | 12.9 |
| | ±5.1 | ±1.5 | ±1.8 | ±0.9 | ±10.2 | ±4.6 | ±5.6 | ±4.7 | ±1.9 | ±0.7 |
| Level01-2d | 253.7 | 26.7 | 184.4 | 20.9 | 182.4 | 21.6 | 245.5 | 28.9 | 235.8 | 26.1 |
| | ±5.1 | ±1.5 | ±3.5 | ±1.2 | ±7.9 | ±2.9 | ±5.7 | ±2.1 | ±5.2 | ±1.6 |
| Level16-2d | 253.7 | 26.7 | 253.2 | 26.6 | 250.8 | 33.4 | 279.2 | 29.5 | 278.6 | 29.4 |
| | ±5.1 | ±1.5 | ±5.1 | ±1.5 | ±10.4 | ±4.3 | ±5.9 | ±1.8 | ±6.0 | ±1.8 |
| Level32-2d | 253.7 | 26.7 | 253.7 | 26.7 | 251.5 | 29.5 | 279.3 | 29.5 | 256.0 | 27.0 |
| | ±5.1 | ±1.5 | ±5.1 | ±1.5 | ±9.8 | ±3.4 | ±5.9 | ±1.8 | ±5.2 | ±1.6 |
| SplitStairs-2d | 253.7 | 26.7 | 279.9 | 35.5 | 291.7 | 37.0 | 188.1 | 30.1 | 160.6 | 28.0 |
| | ±5.1 | ±1.5 | ±4.9 | ±1.7 | ±8.1 | ±3.5 | ±4.8 | ±1.7 | ±4.8 | ±1.6 |
| Stair-2d | 253.7 | 26.7 | 253.0 | 26.7 | 280.3 | 41.9 | 279.3 | 29.5 | 262.3 | 28.4 |
| | ±5.1 | ±1.5 | ±5.1 | ±1.5 | ±16.9 | ±7.7 | ±5.9 | ±1.8 | ±5.4 | ±1.8 |
| Step16-2d | 253.7 | 26.7 | 290.1 | 33.9 | 301.3 | 36.6 | 219.0 | 29.8 | 185.4 | 28.3 |
| | ±5.1 | ±1.5 | ±5.1 | ±1.9 | ±12.1 | ±4.0 | ±5.4 | ±2.0 | ±5.0 | ±1.9 |
| Step50-2d | 253.7 | 26.7 | 291.2 | 34.0 | 300.8 | 37.4 | 215.5 | 29.6 | 207.2 | 27.6 |
| | ±5.1 | ±1.5 | ±5.1 | ±1.9 | ±11.5 | ±5.5 | ±5.2 | ±1.9 | ±5.0 | ±1.8 |

TABLE 8. Squared Errors (with standard deviations). Marg2 Query. 2-d datasets. Lap Mechanism ($\epsilon = 1$).

| Dataset | olsalg | | nnlsalg | | maxalg | | seqalg | | weightalg | |
|---|---|---|---|---|---|---|---|---|---|---|
| | Total | Max | Total | Max | Total | Max | Total | Max | Total | Max |
| Level00-2d | 2629.1 | 31.0 | 86.1 | 36.9 | 106.7 | 47.5 | 107.7 | 75.8 | 39.9 | 19.6 |
| | ±16.7 | ±2.1 | ±1.9 | ±1.5 | ±11.4 | ±8.1 | ±6.3 | ±5.8 | ±1.9 | ±1.3 |
| Level01-2d | 2629.1 | 31.0 | 442.6 | 26.3 | 442.8 | 30.9 | 381.7 | 26.9 | 395.2 | 29.5 |
| | ±16.7 | ±2.1 | ±4.6 | ±1.6 | ±10.5 | ±4.8 | ±4.2 | ±1.8 | ±5.4 | ±2.3 |
| Level16-2d | 2629.1 | 31.0 | 2556.7 | 29.4 | 2665.4 | 42.8 | 2565.1 | 29.6 | 2690.1 | 35.1 |
| | ±16.7 | ±2.1 | ±15.1 | ±1.7 | ±33.4 | ±7.6 | ±14.9 | ±1.7 | ±17.4 | ±2.6 |
| Level32-2d | 2629.1 | 31.0 | 2626.4 | 31.0 | 2711.8 | 34.6 | 2631.7 | 31.1 | 2999.9 | 36.8 |
| | ±16.7 | ±2.1 | ±16.5 | ±2.1 | ±33.2 | ±4.8 | ±16.2 | ±2.0 | ±24.5 | ±4.1 |
| SplitStairs-2d | 2629.1 | 31.0 | 1195.5 | 27.0 | 1206.8 | 27.6 | 1141.5 | 27.0 | 1710.2 | 62.2 |
| | ±16.7 | ±2.1 | ±9.6 | ±1.6 | ±14.6 | ±2.6 | ±9.3 | ±1.6 | ±16.0 | ±3.8 |
| Stair-2d | 2629.1 | 31.0 | 2536.8 | 29.8 | 2504.3 | 40.7 | 2538.8 | 29.8 | 3621.0 | 94.1 |
| | ±16.7 | ±2.1 | ±16.0 | ±2.0 | ±42.0 | ±14.5 | ±15.7 | ±1.9 | ±22.6 | ±4.0 |
| Step16-2d | 2629.1 | 31.0 | 1334.2 | 37.9 | 1353.0 | 36.1 | 1296.8 | 54.0 | 1344.1 | 29.1 |
| | ±16.7 | ±2.1 | ±10.2 | ±2.0 | ±23.8 | ±3.9 | ±10.1 | ±2.5 | ±12.1 | ±2.2 |
| Step50-2d | 2629.1 | 31.0 | 1368.3 | 37.9 | 1370.9 | 40.2 | 1330.9 | 53.6 | 1242.2 | 29.2 |
| | ±16.7 | ±2.1 | ±11.3 | ±2.0 | ±26.4 | ±4.2 | ±10.8 | ±2.4 | ±10.0 | ±2.1 |

TABLE 9. Squared Errors (with standard deviations). Id Query. 2-d datasets. Lap Mechanism ($\epsilon = 1$).

| Dataset | olsalg | nnlsalg | maxalg | seqalg | weightalg |
|---|---|---|---|---|---|
| Level00-2d | 25.3 | 115.5 | 109.6 | 39.5 | 27.3 |
| | ±1.4 | ±3.1 | ±15.4 | ±4.2 | ±1.7 |
| Level01-2d | 25.3 | 35.3 | 32.9 | 32.4 | 26.4 |
| | ±1.4 | ±1.8 | ±3.8 | ±2.3 | ±1.6 |
| Level16-2d | 25.3 | 25.3 | 23.5 | 31.3 | 25.4 |
| | ±1.4 | ±1.4 | ±2.5 | ±1.9 | ±1.4 |
| Level32-2d | 25.3 | 25.3 | 21.4 | 31.2 | 25.3 |
| | ±1.4 | ±1.4 | ±2.3 | ±1.9 | ±1.4 |
| SplitStairs-2d | 25.3 | 35.9 | 37.4 | 30.5 | 25.3 |
| | ±1.4 | ±1.8 | ±3.0 | ±1.9 | ±1.5 |
| Stair-2d | 25.3 | 25.3 | 29.0 | 31.3 | 25.4 |
| | ±1.4 | ±1.4 | ±4.1 | ±1.9 | ±1.4 |
| Step16-2d | 25.3 | 33.6 | 27.8 | 31.6 | 25.4 |
| | ±1.4 | ±1.8 | ±3.1 | ±2.0 | ±1.5 |
| Step50-2d | 25.3 | 33.6 | 36.6 | 31.5 | 27.7 |
| | ±1.4 | ±1.8 | ±3.5 | ±2.0 | ±1.6 |

TABLE 10. Squared Error (with standard deviations). Sum Query. 2-d datasets. Lap Mechanism ($\epsilon = 1$).

| | olsalg | | nnlsalg | | maxalg | | seqalg | | weightalg | |
|---|---|---|---|---|---|---|---|---|---|---|
| **Dataset** | Total | Max | Total | Max | Total | Max | Total | Max | Total | Max |
| Level00-2d | 1058.9 | 124.7 | 301.9 | 75.8 | 376.1 | 105.5 | 351.1 | 170.4 | 118.5 | 50.8 |
| | ±20.9 | ±7.6 | ±7.5 | ±3.6 | ±67.4 | ±42.0 | ±18.6 | ±12.8 | ±6.9 | ±3.0 |
| Level01-2d | 1058.9 | 124.7 | 585.6 | 75.9 | 595.1 | 80.5 | 784.9 | 112.8 | 727.4 | 91.6 |
| | ±20.9 | ±7.6 | ±11.3 | ±4.2 | ±35.3 | ±15.6 | ±19.9 | ±8.7 | ±19.8 | ±6.1 |
| Level16-2d | 1058.9 | 124.7 | 1041.2 | 123.2 | 1049.6 | 117.3 | 1166.1 | 138.3 | 1163.7 | 137.8 |
| | ±20.9 | ±7.6 | ±20.6 | ±7.5 | ±30.8 | ±10.6 | ±23.9 | ±8.7 | ±24.1 | ±8.7 |
| Level32-2d | 1058.9 | 124.7 | 1056.4 | 124.7 | 1020.9 | 123.4 | 1164.3 | 136.8 | 1163.7 | 137.8 |
| | ±20.9 | ±7.6 | ±20.9 | ±7.6 | ±35.7 | ±13.6 | ±23.9 | ±8.6 | ±24.1 | ±8.7 |
| SplitStairs-2d | 1058.9 | 124.7 | 953.0 | 111.1 | 998.2 | 115.7 | 1167.9 | 135.9 | 1163.1 | 137.7 |
| | ±20.9 | ±7.6 | ±18.1 | ±6.6 | ±33.3 | ±11.6 | ±26.0 | ±9.3 | ±24.1 | ±8.7 |
| Stair-2d | 1058.9 | 124.7 | 1051.8 | 123.6 | 987.3 | 116.5 | 1166.7 | 138.4 | 1163.4 | 137.8 |
| | ±20.9 | ±7.6 | ±20.8 | ±7.6 | ±35.1 | ±11.4 | ±23.9 | ±8.7 | ±24.1 | ±8.7 |
| Step16-2d | 1058.9 | 124.7 | 962.8 | 111.8 | 963.3 | 110.9 | 1178.1 | 137.5 | 1161.8 | 137.9 |
| | ±20.9 | ±7.6 | ±18.6 | ±6.8 | ±39.4 | ±12.9 | ±26.0 | ±9.4 | ±24.1 | ±8.7 |
| Step50-2d | 1058.9 | 124.7 | 989.2 | 114.3 | 960.6 | 108.0 | 1166.2 | 139.7 | 1163.4 | 138.0 |
| | ±20.9 | ±7.6 | ±19.1 | ±6.9 | ±40.7 | ±14.0 | ±24.2 | ±8.8 | ±24.1 | ±8.7 |

TABLE 11. Squared Errors (with standard deviations). Marg1 Query. 2-d datasets. Lap Mechanism ($\epsilon = 0.5$).

| | olsalg | | nnlsalg | | maxalg | | seqalg | | weightalg | |
|---|---|---|---|---|---|---|---|---|---|---|
| **Dataset** | Total | Max | Total | Max | Total | Max | Total | Max | Total | Max |
| Level00-2d | 1014.9 | 106.8 | 294.2 | 73.4 | 435.8 | 120.0 | 320.8 | 158.0 | 123.4 | 51.4 |
| | ±20.4 | ±6.2 | ±7.3 | ±3.4 | ±72.5 | ±30.6 | ±17.6 | ±12.2 | ±7.5 | ±2.8 |
| Level01-2d | 1014.9 | 106.8 | 568.8 | 75.4 | 562.0 | 69.0 | 753.1 | 99.2 | 705.9 | 94.3 |
| | ±20.4 | ±6.2 | ±11.3 | ±4.2 | ±34.2 | ±16.0 | ±18.1 | ±7.3 | ±19.7 | ±5.8 |
| Level16-2d | 1014.9 | 106.8 | 997.1 | 105.0 | 1033.3 | 117.5 | 1117.0 | 117.8 | 1114.6 | 117.7 |
| | ±20.4 | ±6.2 | ±20.1 | ±6.1 | ±30.7 | ±11.3 | ±23.6 | ±7.0 | ±23.9 | ±7.1 |
| Level32-2d | 1014.9 | 106.8 | 1012.8 | 106.6 | 1058.9 | 119.1 | 1116.8 | 117.0 | 1114.6 | 117.7 |
| | ±20.4 | ±6.2 | ±20.4 | ±6.2 | ±41.5 | ±13.3 | ±23.7 | ±7.0 | ±23.9 | ±7.1 |
| SplitStairs-2d | 1014.9 | 106.8 | 1093.6 | 144.5 | 1096.2 | 141.7 | 770.2 | 122.2 | 654.3 | 112.5 |
| | ±20.4 | ±6.2 | ±19.3 | ±6.8 | ±32.3 | ±12.2 | ±20.3 | ±7.0 | ±19.5 | ±6.5 |
| Stair-2d | 1014.9 | 106.8 | 1009.5 | 106.7 | 967.8 | 115.2 | 1117.5 | 117.8 | 1095.4 | 117.5 |
| | ±20.4 | ±6.2 | ±20.3 | ±6.1 | ±33.5 | ±14.8 | ±23.7 | ±7.0 | ±23.3 | ±7.0 |
| Step16-2d | 1014.9 | 106.8 | 1127.2 | 130.6 | 1195.6 | 145.6 | 884.6 | 119.8 | 740.6 | 113.1 |
| | ±20.4 | ±6.2 | ±20.1 | ±7.3 | ±43.1 | ±16.2 | ±22.5 | ±8.3 | ±20.1 | ±7.5 |
| Step50-2d | 1014.9 | 106.8 | 1164.1 | 136.0 | 1211.6 | 157.2 | 868.0 | 117.9 | 740.5 | 113.1 |
| | ±20.4 | ±6.2 | ±20.5 | ±7.6 | ±47.3 | ±24.7 | ±20.8 | ±7.6 | ±20.1 | ±7.5 |

TABLE 12. Squared Errors (with standard deviations). Marg2 Query. 2-d datasets. Lap Mechanism ($\epsilon = 0.5$).

| Dataset | olsalg | | nnlsalg | | maxalg | | seqalg | | weightalg | |
|---|---|---|---|---|---|---|---|---|---|---|
| | Total | Max | Total | Max | Total | Max | Total | Max | Total | Max |
| Level00-2d | 10516.5 | 124.0 | 344.2 | 147.4 | 443.7 | 173.1 | 437.3 | 283.6 | 159.2 | 78.4 |
| | ±66.8 | ±8.3 | ±7.7 | ±6.0 | ±77.1 | ±53.5 | ±17.3 | ±15.1 | ±7.8 | ±5.3 |
| Level01-2d | 10516.5 | 124.0 | 960.2 | 110.7 | 1049.6 | 149.7 | 775.0 | 118.4 | 790.1 | 105.6 |
| | ±66.8 | ±8.3 | ±12.8 | ±6.1 | ±44.7 | ±21.6 | ±12.1 | ±7.8 | ±19.0 | ±9.0 |
| Level16-2d | 10516.5 | 124.0 | 8551.1 | 109.9 | 8627.8 | 108.6 | 8579.5 | 109.2 | 8741.9 | 140.3 |
| | ±66.8 | ±8.3 | ±48.1 | ±6.9 | ±71.0 | ±9.9 | ±47.0 | ±6.6 | ±53.3 | ±10.2 |
| Level32-2d | 10516.5 | 124.0 | 10227.0 | 117.5 | 10545.2 | 143.4 | 10255.8 | 118.2 | 10761.4 | 140.3 |
| | ±66.8 | ±8.3 | ±60.3 | ±6.8 | ±113.8 | ±24.0 | ±59.6 | ±6.8 | ±69.6 | ±10.2 |
| SplitStairs-2d | 10516.5 | 124.0 | 4296.4 | 106.6 | 4274.2 | 113.4 | 4087.0 | 106.9 | 4501.3 | 140.3 |
| | ±66.8 | ±8.3 | ±34.1 | ±6.4 | ±59.4 | ±12.7 | ±34.0 | ±6.1 | ±42.9 | ±10.2 |
| Stair-2d | 10516.5 | 124.0 | 9650.6 | 118.7 | 9689.6 | 127.9 | 9654.7 | 118.3 | 12121.6 | 207.2 |
| | ±66.8 | ±8.3 | ±60.9 | ±7.9 | ±111.7 | ±13.3 | ±59.6 | ±7.7 | ±89.5 | ±13.3 |
| Step16-2d | 10516.5 | 124.0 | 4518.6 | 150.2 | 4593.0 | 184.9 | 4408.7 | 205.6 | 4313.0 | 106.8 |
| | ±66.8 | ±8.3 | ±32.0 | ±7.8 | ±69.6 | ±21.2 | ±32.5 | ±9.9 | ±35.2 | ±9.1 |
| Step50-2d | 10516.5 | 124.0 | 5451.1 | 151.7 | 5576.9 | 178.4 | 5301.0 | 215.9 | 6483.3 | 142.2 |
| | ±66.8 | ±8.3 | ±43.9 | ±7.8 | ±106.7 | ±26.7 | ±42.2 | ±9.7 | ±56.2 | ±10.2 |

TABLE 13. Squared Errors (with standard deviations). Id Query. 2-d datasets. Lap Mechanism ($\epsilon = 0.5$).

| Dataset | olsalg | nnlsalg | maxalg | seqalg | weightalg |
|---|---|---|---|---|---|
| Level00-2d | 101.3 | 461.9 | 533.9 | 149.2 | 108.5 |
| | ±5.7 | ±12.5 | ±76.7 | ±14.8 | ±7.0 |
| Level01-2d | 101.3 | 189.5 | 215.3 | 126.4 | 105.7 |
| | ±5.7 | ±8.6 | ±27.2 | ±8.9 | ±6.3 |
| Level16-2d | 101.3 | 101.8 | 107.0 | 125.0 | 101.6 |
| | ±5.7 | ±5.7 | ±8.6 | ±7.8 | ±5.7 |
| Level32-2d | 101.3 | 101.3 | 100.0 | 125.1 | 101.6 |
| | ±5.7 | ±5.7 | ±9.9 | ±7.8 | ±5.7 |
| SplitStairs-2d | 101.3 | 146.6 | 141.7 | 124.5 | 101.1 |
| | ±5.7 | ±7.4 | ±11.3 | ±8.5 | ±5.8 |
| Stair-2d | 101.3 | 101.5 | 100.0 | 125.0 | 101.5 |
| | ±5.7 | ±5.7 | ±9.7 | ±7.8 | ±5.7 |
| Step16-2d | 101.3 | 136.7 | 153.6 | 123.1 | 101.7 |
| | ±5.7 | ±7.2 | ±17.7 | ±8.1 | ±5.8 |
| Step50-2d | 101.3 | 134.3 | 153.3 | 124.9 | 101.7 |
| | ±5.7 | ±7.2 | ±20.2 | ±7.9 | ±5.8 |

TABLE 14. Squared Error (with standard deviations). Sum Query. 2-d datasets. Lap Mechanism ($\epsilon = 0.5$).

| Dataset | olsalg | | nnlsalg | | maxalg | | seqalg | | weightalg | |
|---|---|---|---|---|---|---|---|---|---|---|
| | Total | Max | Total | Max | Total | Max | Total | Max | Total | Max |
| Level00-2d | 26471.7 | 3118.5 | 7548.3 | 1894.1 | 5694.6 | 2431.7 | 8088.6 | 3727.6 | 2958.3 | 1269.9 |
| | ±523.4 | ±190.8 | ±186.9 | ±89.1 | ±1376.7 | ±1126.3 | ±396.5 | ±293.5 | ±173.8 | ±74.6 |
| Level01-2d | 26471.7 | 3118.5 | 8866.5 | 1805.8 | 6245.0 | 1635.9 | 9712.2 | 3237.9 | 6134.4 | 1757.0 |
| | ±523.4 | ±190.8 | ±211.7 | ±89.7 | ±1262.2 | ±398.3 | ±322.7 | ±199.7 | ±255.9 | ±105.1 |
| Level16-2d | 26471.7 | 3118.5 | 21759.9 | 2580.4 | 20142.9 | 3174.1 | 28382.5 | 3365.1 | 28246.4 | 3309.6 |
| | ±523.4 | ±190.8 | ±404.8 | ±150.6 | ±1602.1 | ±705.0 | ±556.6 | ±203.4 | ±570.3 | ±198.6 |
| Level32-2d | 26471.7 | 3118.5 | 24466.9 | 2921.1 | 24229.8 | 3687.0 | 28991.1 | 3449.8 | 28907.3 | 3427.4 |
| | ±523.4 | ±190.8 | ±475.8 | ±175.8 | ±1377.6 | ±602.9 | ±591.7 | ±216.9 | ±598.3 | ±216.0 |
| SplitStairs-2d | 26471.7 | 3118.5 | 20081.4 | 2275.5 | 28035.9 | 4753.6 | 27437.2 | 3119.0 | 26088.0 | 2901.0 |
| | ±523.4 | ±190.8 | ±364.6 | ±133.2 | ±4283.3 | ±2378.7 | ±554.6 | ±197.6 | ±544.4 | ±180.8 |
| Stair-2d | 26471.7 | 3118.5 | 25170.6 | 2959.2 | 24453.9 | 2980.0 | 29157.7 | 3457.4 | 29081.5 | 3445.1 |
| | ±523.4 | ±190.8 | ±494.3 | ±179.8 | ±1384.4 | ±678.3 | ±598.2 | ±216.8 | ±602.2 | ±216.9 |
| Step16-2d | 26471.7 | 3118.5 | 17895.7 | 2105.1 | 21071.0 | 2841.1 | 24033.8 | 2835.4 | 22665.5 | 2550.1 |
| | ±523.4 | ±190.8 | ±329.4 | ±120.1 | ±2083.2 | ±751.8 | ±472.4 | ±189.9 | ±521.5 | ±160.1 |
| Step50-2d | 26471.7 | 3118.5 | 23086.7 | 2695.9 | 23437.1 | 3793.6 | 29141.9 | 3431.7 | 28085.5 | 3320.2 |
| | ±523.4 | ±190.8 | ±439.7 | ±160.4 | ±2507.8 | ±954.8 | ±595.8 | ±216.0 | ±580.2 | ±208.3 |

TABLE 15. Squared Errors (with standard deviations). Marg1 Query. 2-d datasets. Lap Mechanism ($\epsilon = 0.1$).

| Dataset | olsalg | | nnlsalg | | maxalg | | seqalg | | weightalg | |
|---|---|---|---|---|---|---|---|---|---|---|
| | Total | Max | Total | Max | Total | Max | Total | Max | Total | Max |
| Level00-2d | 25372.5 | 2668.9 | 7355.9 | 1833.8 | 14218.1 | 9473.1 | 7665.7 | 3512.6 | 3080.0 | 1285.3 |
| | ±510.4 | ±154.1 | ±182.7 | ±85.1 | ±1243.4 | ±105.1 | ±365.1 | ±262.1 | ±187.2 | ±69.1 |
| Level01-2d | 25372.5 | 2668.9 | 8646.9 | 1759.3 | 18406.0 | 5264.1 | 9151.9 | 3042.7 | 6193.8 | 1788.8 |
| | ±510.4 | ±154.1 | ±209.6 | ±87.2 | ±5237.9 | ±3718.8 | ±301.6 | ±184.7 | ±272.2 | ±98.7 |
| Level16-2d | 25372.5 | 2668.9 | 20984.9 | 2250.9 | 23854.3 | 3431.9 | 27062.7 | 2939.7 | 26963.2 | 2873.5 |
| | ±510.4 | ±154.1 | ±398.4 | ±132.8 | ±2344.5 | ±1162.7 | ±535.8 | ±175.0 | ±555.5 | ±180.1 |
| Level32-2d | 25372.5 | 2668.9 | 23472.4 | 2481.9 | 24557.9 | 3312.0 | 27908.4 | 2954.4 | 27757.4 | 2924.1 |
| | ±510.4 | ±154.1 | ±464.9 | ±141.4 | ±1452.2 | ±534.7 | ±585.4 | ±176.2 | ±588.5 | ±174.8 |
| SplitStairs-2d | 25372.5 | 2668.9 | 22466.1 | 3354.0 | 23583.4 | 4580.8 | 18798.5 | 3011.1 | 21400.8 | 5680.5 |
| | ±510.4 | ±154.1 | ±401.0 | ±153.1 | ±3017.1 | ±1448.8 | ±473.0 | ±171.4 | ±556.3 | ±228.8 |
| Stair-2d | 25372.5 | 2668.9 | 24634.2 | 2607.9 | 24122.0 | 3112.3 | 27780.8 | 2943.5 | 27156.0 | 2924.4 |
| | ±510.4 | ±154.1 | ±480.4 | ±167.7 | ±1303.0 | ±637.1 | ±576.0 | ±175.7 | ±567.2 | ±174.2 |
| Step16-2d | 25372.5 | 2668.9 | 20455.3 | 2297.4 | 22047.5 | 4131.7 | 20990.4 | 2825.8 | 22687.6 | 3290.4 |
| | ±510.4 | ±154.1 | ±368.2 | ±117.6 | ±2756.9 | ±1537.4 | ±475.7 | ±177.9 | ±558.5 | ±210.0 |
| Step50-2d | 25372.5 | 2668.9 | 26829.3 | 3076.4 | 29360.4 | 4341.9 | 21590.8 | 2950.8 | 18446.5 | 2816.4 |
| | ±510.4 | ±154.1 | ±481.0 | ±163.5 | ±2504.2 | ±1149.5 | ±516.3 | ±189.3 | ±495.3 | ±185.0 |

TABLE 16. Squared Errors (with standard deviations). Marg2 Query. 2-d datasets. Lap Mechanism ($\epsilon = 0.1$).

| Dataset | olsalg Total | olsalg Max | nnlsalg Total | nnlsalg Max | maxalg Total | maxalg Max | seqalg Total | seqalg Max | weightalg Total | weightalg Max |
|---|---|---|---|---|---|---|---|---|---|---|
| Level00-2d | 262911.4 | 3100.0 | 8605.7 | 3684.9 | 11473.9 | 4702.8 | 10107.9 | 6442.2 | 3969.7 | 1957.5 |
|  | ±1671.2 | ±207.7 | ±192.5 | ±150.6 | ±2457.8 | ±1883.9 | ±389.6 | ±352.0 | ±195.8 | ±132.6 |
| Level01-2d | 262911.4 | 3100.0 | 10798.5 | 3274.3 | 12020.4 | 4404.7 | 10372.8 | 4925.4 | 6766.7 | 2206.0 |
|  | ±1671.2 | ±207.7 | ±214.0 | ±149.5 | ±2257.9 | ±1463.9 | ±270.5 | ±229.7 | ±260.6 | ±159.1 |
| Level16-2d | 262911.4 | 3100.0 | 68991.2 | 2595.8 | 67044.4 | 1549.9 | 63441.2 | 2564.4 | 65066.2 | 3271.1 |
|  | ±1671.2 | ±207.7 | ±615.4 | ±160.2 | ±2491.7 | ±425.1 | ±511.1 | ±150.1 | ±673.4 | ±240.5 |
| Level32-2d | 262911.4 | 3100.0 | 124733.6 | 2616.5 | 123979.9 | 2573.2 | 122210.1 | 2588.6 | 123469.4 | 3496.1 |
|  | ±1671.2 | ±207.7 | ±882.3 | ±166.1 | ±2800.4 | ±862.2 | ±817.7 | ±158.4 | ±906.2 | ±254.5 |
| SplitStairs-2d | 262911.4 | 3100.0 | 50234.5 | 2996.2 | 52915.5 | 1999.9 | 45193.8 | 3302.0 | 44590.7 | 2844.4 |
|  | ±1671.2 | ±207.7 | ±464.6 | ±156.8 | ±3467.3 | ±912.6 | ±403.3 | ±166.0 | ±550.9 | ±228.3 |
| Stair-2d | 262911.4 | 3100.0 | 155160.3 | 3097.7 | 155361.3 | 2718.6 | 153483.3 | 3581.2 | 152550.7 | 2916.9 |
|  | ±1671.2 | ±207.7 | ±997.9 | ±178.7 | ±2725.5 | ±370.6 | ±950.4 | ±187.6 | ±1064.5 | ±229.7 |
| Step16-2d | 262911.4 | 3100.0 | 37999.0 | 3529.7 | 39280.2 | 3791.0 | 34028.3 | 4686.3 | 31919.8 | 2665.9 |
|  | ±1671.2 | ±207.7 | ±408.0 | ±178.7 | ±3077.8 | ±1182.3 | ±353.0 | ±219.2 | ±511.3 | ±227.3 |
| Step50-2d | 262911.4 | 3100.0 | 89586.0 | 3702.8 | 94050.9 | 3428.8 | 86582.5 | 5201.5 | 82313.0 | 2673.2 |
|  | ±1671.2 | ±207.7 | ±674.3 | ±191.3 | ±3765.8 | ±693.1 | ±623.4 | ±235.0 | ±665.9 | ±227.0 |

TABLE 17. Squared Errors (with standard deviations). Id Query. 2-d datasets. Lap Mechanism ($\epsilon = 0.1$).

| Dataset | olsalg | nnlsalg | maxalg | seqalg | weightalg |
|---|---|---|---|---|---|
| Level00-2d | 2532.8 | 11547.9 | 21245.2 | 3869.8 | 2706.6 |
|  | ±141.5 | ±313.5 | ±204.6 | ±318.7 | ±174.9 |
| Level01-2d | 2532.8 | 8668.1 | 9288.8 | 2672.6 | 2609.7 |
|  | ±141.5 | ±281.0 | ±2043.4 | ±187.5 | ±153.9 |
| Level16-2d | 2532.8 | 3021.9 | 2354.3 | 3096.8 | 2643.9 |
|  | ±141.5 | ±164.5 | ±426.9 | ±193.2 | ±156.6 |
| Level32-2d | 2532.8 | 2656.6 | 2517.2 | 3131.0 | 2541.3 |
|  | ±141.5 | ±149.7 | ±336.6 | ±194.7 | ±142.8 |
| SplitStairs-2d | 2532.8 | 4363.3 | 3599.9 | 3038.5 | 2637.0 |
|  | ±141.5 | ±203.0 | ±809.9 | ±193.4 | ±154.5 |
| Stair-2d | 2532.8 | 2635.1 | 3014.5 | 3126.8 | 2537.6 |
|  | ±141.5 | ±148.5 | ±483.3 | ±194.2 | ±142.5 |
| Step16-2d | 2532.8 | 4413.9 | 4285.1 | 3133.3 | 2644.7 |
|  | ±141.5 | ±208.2 | ±1574.4 | ±197.5 | ±157.4 |
| Step50-2d | 2532.8 | 3540.9 | 3890.6 | 3114.6 | 2536.0 |
|  | ±141.5 | ±184.7 | ±1362.8 | ±196.2 | ±146.5 |

TABLE 18. Squared Error (with standard deviations). Sum Query. 2-d datasets. Lap Mechanism ($\epsilon = 0.1$).

| Dataset | olsalg | | nnlsalg | | maxalg | | seqalg | | weightalg | |
|---|---|---|---|---|---|---|---|---|---|---|
| | Total | Max | Total | Max | Total | Max | Total | Max | Total | Max |
| PUMA0101301 | 249.0 | 32.2 | 138.3 | 19.3 | 149.2 | 19.4 | 157.0 | 33.5 | 147.0 | 44.9 |
| | ±5.3 | ±2.0 | ±2.8 | ±1.0 | ±4.2 | ±1.3 | ±4.7 | ±2.4 | ±4.1 | ±2.1 |
| PUMA0800803 | 249.0 | 32.2 | 185.7 | 41.8 | 193.3 | 38.8 | 181.6 | 30.2 | 169.0 | 35.3 |
| | ±5.3 | ±2.0 | ±3.6 | ±2.0 | ±6.2 | ±2.7 | ±6.3 | ±2.8 | ±4.5 | ±1.9 |
| PUMA1304600 | 249.0 | 32.2 | 194.6 | 25.3 | 199.5 | 26.7 | 196.9 | 33.2 | 196.4 | 46.6 |
| | ±5.3 | ±2.0 | ±3.7 | ±1.3 | ±6.2 | ±2.4 | ±7.2 | ±2.6 | ±5.1 | ±2.5 |
| PUMA1703529 | 249.0 | 32.2 | 184.9 | 29.9 | 195.3 | 31.2 | 207.8 | 36.4 | 185.1 | 46.2 |
| | ±5.3 | ±2.0 | ±3.6 | ±1.5 | ±6.0 | ±2.4 | ±7.8 | ±3.4 | ±4.9 | ±2.5 |
| PUMA1703531 | 249.0 | 32.2 | 127.2 | 19.2 | 138.7 | 21.1 | 148.2 | 32.4 | 136.1 | 26.5 |
| | ±5.3 | ±2.0 | ±2.6 | ±1.0 | ±4.1 | ±1.6 | ±4.3 | ±2.6 | ±4.0 | ±1.4 |
| PUMA1901700 | 249.0 | 32.2 | 205.6 | 31.2 | 211.8 | 31.3 | 198.4 | 34.9 | 163.2 | 28.3 |
| | ±5.3 | ±2.0 | ±3.9 | ±1.6 | ±6.8 | ±2.9 | ±7.3 | ±3.0 | ±4.5 | ±1.7 |
| PUMA2401004 | 249.0 | 32.2 | 228.5 | 57.1 | 245.7 | 59.2 | 191.5 | 36.3 | 178.5 | 41.5 |
| | ±5.3 | ±2.0 | ±4.3 | ±2.4 | ±7.5 | ±4.2 | ±12.7 | ±5.9 | ±4.6 | ±2.0 |
| PUMA2602702 | 249.0 | 32.2 | 175.8 | 32.9 | 191.6 | 34.4 | 189.6 | 34.7 | 167.4 | 33.8 |
| | ±5.3 | ±2.0 | ±3.4 | ±1.6 | ±5.8 | ±2.7 | ±6.1 | ±2.7 | ±4.3 | ±1.6 |
| PUMA2801100 | 249.0 | 32.2 | 145.0 | 23.9 | 150.5 | 23.5 | 156.5 | 32.0 | 133.4 | 24.4 |
| | ±5.3 | ±2.0 | ±2.9 | ±1.2 | ±4.4 | ±1.6 | ±5.3 | ±2.6 | ±3.8 | ±1.3 |
| PUMA2901901 | 249.0 | 32.2 | 165.3 | 29.4 | 174.4 | 32.0 | 188.9 | 35.3 | 149.3 | 30.1 |
| | ±5.3 | ±2.0 | ±3.3 | ±1.4 | ±5.3 | ±2.2 | ±6.5 | ±3.1 | ±4.2 | ±1.8 |
| PUMA3200405 | 249.0 | 32.2 | 225.0 | 30.5 | 237.2 | 32.4 | 225.5 | 35.8 | 216.7 | 33.6 |
| | ±5.3 | ±2.0 | ±4.2 | ±1.7 | ±8.1 | ±3.1 | ±7.5 | ±3.2 | ±5.1 | ±2.1 |
| PUMA3603710 | 249.0 | 32.2 | 240.7 | 34.2 | 245.8 | 32.0 | 216.4 | 33.0 | 227.1 | 34.3 |
| | ±5.3 | ±2.0 | ±4.5 | ±1.8 | ±7.6 | ±2.8 | ±7.2 | ±2.7 | ±5.4 | ±2.2 |
| PUMA3604010 | 249.0 | 32.2 | 204.0 | 33.7 | 210.2 | 32.5 | 192.3 | 32.4 | 207.4 | 49.9 |
| | ±5.3 | ±2.0 | ±3.8 | ±1.8 | ±6.6 | ±2.9 | ±7.5 | ±2.9 | ±5.6 | ±2.6 |
| PUMA5101301 | 249.0 | 32.2 | 209.2 | 45.1 | 227.7 | 45.0 | 198.2 | 34.4 | 172.2 | 41.2 |
| | ±5.3 | ±2.0 | ±4.0 | ±2.1 | ±7.3 | ±3.4 | ±6.8 | ±3.1 | ±4.5 | ±2.3 |
| PUMA5151255 | 249.0 | 32.2 | 239.3 | 41.4 | 249.3 | 39.7 | 192.8 | 30.8 | 160.5 | 28.0 |
| | ±5.3 | ±2.0 | ±4.4 | ±2.0 | ±7.6 | ±2.9 | ±8.2 | ±3.4 | ±4.4 | ±1.6 |

TABLE 19. Squared Errors (with standard deviations). Marg1 Query. PUMS datasets. Lap Mechanism ($\epsilon = 1$).

| Dataset | olsalg Total | Max | nnlsalg Total | Max | maxalg Total | Max | seqalg Total | Max | weightalg Total | Max |
|---|---|---|---|---|---|---|---|---|---|---|
| PUMA0101301 | 656.6 | 30.8 | 199.1 | 62.3 | 223.5 | 67.5 | 185.2 | 45.9 | 152.4 | 58.3 |
| | ±8.6 | ±2.1 | ±3.4 | ±2.1 | ±5.6 | ±3.1 | ±5.1 | ±2.5 | ±4.4 | ±2.2 |
| PUMA0800803 | 656.6 | 30.8 | 292.9 | 62.7 | 312.7 | 60.6 | 319.5 | 38.5 | 239.6 | 26.2 |
| | ±8.6 | ±2.1 | ±4.2 | ±2.3 | ±7.7 | ±3.6 | ±8.8 | ±3.4 | ±5.8 | ±1.7 |
| PUMA1304600 | 656.6 | 30.8 | 315.8 | 53.3 | 325.7 | 52.2 | 293.2 | 32.5 | 288.6 | 46.8 |
| | ±8.6 | ±2.1 | ±4.4 | ±2.1 | ±7.5 | ±3.3 | ±8.7 | ±2.7 | ±6.0 | ±2.0 |
| PUMA1703529 | 656.6 | 30.8 | 291.3 | 58.6 | 308.7 | 62.3 | 277.9 | 40.2 | 217.0 | 30.3 |
| | ±8.6 | ±2.1 | ±4.3 | ±2.2 | ±7.5 | ±3.6 | ±8.3 | ±3.3 | ±5.6 | ±1.9 |
| PUMA1703531 | 656.6 | 30.8 | 186.0 | 78.1 | 205.7 | 81.9 | 134.4 | 58.5 | 77.0 | 26.2 |
| | ±8.6 | ±2.1 | ±3.3 | ±2.4 | ±5.5 | ±3.5 | ±4.2 | ±2.9 | ±3.2 | ±1.5 |
| PUMA1901700 | 656.6 | 30.8 | 337.3 | 49.5 | 359.7 | 53.7 | 350.1 | 38.1 | 301.8 | 31.3 |
| | ±8.6 | ±2.1 | ±4.6 | ±2.1 | ±8.4 | ±3.4 | ±9.5 | ±3.4 | ±6.5 | ±1.7 |
| PUMA2401004 | 656.6 | 30.8 | 387.1 | 49.0 | 400.6 | 47.9 | 534.6 | 40.7 | 447.1 | 37.4 |
| | ±8.6 | ±2.1 | ±4.9 | ±2.1 | ±8.5 | ±3.1 | ±23.8 | ±7.3 | ±8.1 | ±2.2 |
| PUMA2602702 | 656.6 | 30.8 | 257.1 | 61.9 | 271.7 | 64.9 | 257.1 | 35.2 | 208.0 | 28.8 |
| | ±8.6 | ±2.1 | ±3.9 | ±2.3 | ±6.9 | ±3.7 | ±6.9 | ±2.4 | ±5.2 | ±1.5 |
| PUMA2801100 | 656.6 | 30.8 | 212.2 | 68.6 | 228.6 | 70.6 | 174.2 | 41.6 | 136.5 | 26.5 |
| | ±8.6 | ±2.1 | ±3.5 | ±2.3 | ±6.0 | ±3.3 | ±4.8 | ±2.5 | ±3.9 | ±1.5 |
| PUMA2901901 | 656.6 | 30.8 | 256.4 | 64.8 | 278.7 | 68.3 | 235.9 | 38.9 | 197.8 | 42.7 |
| | ±8.6 | ±2.1 | ±3.9 | ±2.3 | ±7.1 | ±3.6 | ±6.5 | ±2.6 | ±5.0 | ±1.8 |
| PUMA3200405 | 656.6 | 30.8 | 406.8 | 49.7 | 422.1 | 53.9 | 403.9 | 35.1 | 393.9 | 37.9 |
| | ±8.6 | ±2.1 | ±5.3 | ±2.1 | ±10.1 | ±3.8 | ±10.6 | ±3.5 | ±7.6 | ±2.1 |
| PUMA3603710 | 656.6 | 30.8 | 445.5 | 39.0 | 475.8 | 44.0 | 466.8 | 34.4 | 414.6 | 43.3 |
| | ±8.6 | ±2.1 | ±5.7 | ±1.9 | ±10.8 | ±3.4 | ±11.4 | ±3.3 | ±8.0 | ±2.3 |
| PUMA3604010 | 656.6 | 30.8 | 330.4 | 47.7 | 357.8 | 47.6 | 353.5 | 37.2 | 328.3 | 38.9 |
| | ±8.6 | ±2.1 | ±4.5 | ±2.0 | ±8.3 | ±3.1 | ±9.6 | ±3.1 | ±6.9 | ±2.0 |
| PUMA5101301 | 656.6 | 30.8 | 330.6 | 54.9 | 348.9 | 60.2 | 390.5 | 34.2 | 326.9 | 29.6 |
| | ±8.6 | ±2.1 | ±4.5 | ±2.2 | ±8.3 | ±4.0 | ±9.8 | ±3.3 | ±6.7 | ±2.1 |
| PUMA5151255 | 656.6 | 30.8 | 398.5 | 41.6 | 412.3 | 46.3 | 465.7 | 34.7 | 424.5 | 33.0 |
| | ±8.6 | ±2.1 | ±5.1 | ±1.9 | ±9.2 | ±3.4 | ±12.8 | ±3.9 | ±7.6 | ±2.0 |

TABLE 20. Squared Errors (with standard deviations). Marg2 Query. PUMS datasets. Lap Mechanism ($\epsilon = 1$).

| Dataset | olsalg | | nnlsalg | | maxalg | | seqalg | | weightalg | |
|---|---|---|---|---|---|---|---|---|---|---|
| | Total | Max | Total | Max | Total | Max | Total | Max | Total | Max |
| PUMA0101301 | 5976.5 | 35.7 | 235.4 | 41.0 | 260.1 | 43.8 | 236.5 | 60.8 | 302.3 | 93.0 |
| | ±26.4 | ±2.6 | ±3.2 | ±1.5 | ±5.5 | ±2.3 | ±4.3 | ±2.4 | ±5.8 | ±2.9 |
| PUMA0800803 | 5976.5 | 35.7 | 385.8 | 26.6 | 412.8 | 30.3 | 401.9 | 64.5 | 352.6 | 31.1 |
| | ±26.4 | ±2.6 | ±4.2 | ±1.6 | ±8.0 | ±2.6 | ±6.8 | ±3.4 | ±6.1 | ±1.4 |
| PUMA1304600 | 5976.5 | 35.7 | 477.2 | 30.0 | 491.1 | 31.0 | 416.1 | 40.1 | 473.3 | 44.7 |
| | ±26.4 | ±2.6 | ±4.6 | ±1.7 | ±8.1 | ±3.6 | ±6.7 | ±3.9 | ±6.0 | ±2.2 |
| PUMA1703529 | 5976.5 | 35.7 | 396.6 | 28.8 | 412.5 | 28.9 | 344.7 | 30.8 | 442.2 | 56.8 |
| | ±26.4 | ±2.6 | ±4.5 | ±1.3 | ±8.0 | ±2.2 | ±6.2 | ±2.8 | ±7.1 | ±2.5 |
| PUMA1703531 | 5976.5 | 35.7 | 180.4 | 23.2 | 200.0 | 24.9 | 148.1 | 28.9 | 153.0 | 26.6 |
| | ±26.4 | ±2.6 | ±2.5 | ±1.0 | ±4.4 | ±1.5 | ±3.0 | ±1.5 | ±3.7 | ±1.2 |
| PUMA1901700 | 5976.5 | 35.7 | 515.9 | 32.0 | 527.4 | 31.0 | 451.5 | 40.8 | 554.9 | 49.7 |
| | ±26.4 | ±2.6 | ±5.0 | ±1.7 | ±8.6 | ±2.0 | ±6.8 | ±3.4 | ±8.0 | ±3.1 |
| PUMA2401004 | 5976.5 | 35.7 | 628.8 | 35.6 | 659.9 | 37.0 | 672.5 | 78.3 | 606.3 | 38.3 |
| | ±26.4 | ±2.6 | ±5.8 | ±1.7 | ±10.3 | ±3.0 | ±16.6 | ±6.7 | ±7.6 | ±2.0 |
| PUMA2602702 | 5976.5 | 35.7 | 326.3 | 26.9 | 349.1 | 27.6 | 305.2 | 40.2 | 296.3 | 37.4 |
| | ±26.4 | ±2.6 | ±3.9 | ±1.6 | ±7.1 | ±2.9 | ±5.3 | ±2.1 | ±5.3 | ±2.2 |
| PUMA2801100 | 5976.5 | 35.7 | 233.5 | 25.2 | 253.1 | 26.3 | 203.3 | 30.8 | 210.1 | 27.6 |
| | ±26.4 | ±2.6 | ±2.8 | ±0.6 | ±5.1 | ±1.2 | ±3.5 | ±0.8 | ±3.9 | ±0.8 |
| PUMA2901901 | 5976.5 | 35.7 | 307.1 | 26.9 | 333.1 | 26.8 | 279.8 | 33.4 | 276.7 | 38.0 |
| | ±26.4 | ±2.6 | ±3.8 | ±1.6 | ±6.9 | ±2.8 | ±4.9 | ±1.9 | ±5.0 | ±2.2 |
| PUMA3200405 | 5976.5 | 35.7 | 759.7 | 31.0 | 787.4 | 30.3 | 685.8 | 41.0 | 803.3 | 63.2 |
| | ±26.4 | ±2.6 | ±6.3 | ±1.8 | ±12.0 | ±2.5 | ±8.4 | ±3.2 | ±8.7 | ±3.4 |
| PUMA3603710 | 5976.5 | 35.7 | 916.9 | 35.5 | 960.2 | 35.0 | 815.7 | 45.6 | 992.8 | 52.9 |
| | ±26.4 | ±2.6 | ±7.3 | ±1.4 | ±13.3 | ±2.3 | ±9.1 | ±2.7 | ±10.2 | ±2.0 |
| PUMA3604010 | 5976.5 | 35.7 | 502.6 | 31.6 | 523.4 | 28.8 | 464.5 | 37.8 | 534.7 | 62.4 |
| | ±26.4 | ±2.6 | ±4.7 | ±1.4 | ±8.6 | ±2.5 | ±6.7 | ±2.3 | ±7.8 | ±4.0 |
| PUMA5101301 | 5976.5 | 35.7 | 510.9 | 29.3 | 529.3 | 28.7 | 506.6 | 63.4 | 472.3 | 54.7 |
| | ±26.4 | ±2.6 | ±4.9 | ±1.5 | ±8.7 | ±2.2 | ±7.2 | ±3.4 | ±6.4 | ±2.1 |
| PUMA5151255 | 5976.5 | 35.7 | 741.7 | 34.7 | 760.6 | 34.9 | 688.8 | 46.2 | 762.4 | 92.8 |
| | ±26.4 | ±2.6 | ±6.4 | ±1.9 | ±10.9 | ±2.3 | ±10.1 | ±4.0 | ±9.9 | ±6.2 |

TABLE 21. Squared Errors (with standard deviations). Id Query. PUMS datasets. Lap Mechanism ($\epsilon = 1$).

| Dataset | olsalg | nnlsalg | maxalg | seqalg | weightalg |
|---|---|---|---|---|---|
| PUMA0101301 | 26.8 | 130.2 | 139.6 | 31.9 | 28.4 |
| | ±1.5 | ±3.6 | ±5.3 | ±2.5 | ±1.6 |
| PUMA0800803 | 26.8 | 100.5 | 104.4 | 29.5 | 26.8 |
| | ±1.5 | ±3.2 | ±5.5 | ±2.5 | ±1.6 |
| PUMA1304600 | 26.8 | 89.9 | 90.6 | 30.7 | 27.7 |
| | ±1.5 | ±3.0 | ±5.0 | ±2.7 | ±1.6 |
| PUMA1703529 | 26.8 | 96.9 | 97.1 | 32.3 | 27.7 |
| | ±1.5 | ±3.2 | ±4.9 | ±3.0 | ±1.6 |
| PUMA1703531 | 26.8 | 135.7 | 137.0 | 28.6 | 27.2 |
| | ±1.5 | ±3.6 | ±5.0 | ±2.4 | ±1.6 |
| PUMA1901700 | 26.8 | 88.4 | 94.5 | 33.7 | 27.0 |
| | ±1.5 | ±3.0 | ±5.6 | ±3.3 | ±1.6 |
| PUMA2401004 | 26.8 | 83.2 | 82.2 | 25.4 | 27.6 |
| | ±1.5 | ±3.0 | ±4.8 | ±4.3 | ±1.6 |
| PUMA2602702 | 26.8 | 102.8 | 104.1 | 29.5 | 27.1 |
| | ±1.5 | ±3.2 | ±5.3 | ±2.4 | ±1.6 |
| PUMA2801100 | 26.8 | 123.5 | 120.5 | 28.4 | 27.4 |
| | ±1.5 | ±3.5 | ±4.8 | ±2.4 | ±1.6 |
| PUMA2901901 | 26.8 | 109.4 | 106.7 | 29.1 | 27.3 |
| | ±1.5 | ±3.3 | ±4.7 | ±2.4 | ±1.6 |
| PUMA3200405 | 26.8 | 72.8 | 73.3 | 31.4 | 27.4 |
| | ±1.5 | ±2.8 | ±4.6 | ±2.8 | ±1.6 |
| PUMA3603710 | 26.8 | 66.8 | 65.1 | 32.2 | 27.3 |
| | ±1.5 | ±2.7 | ±4.3 | ±2.8 | ±1.6 |
| PUMA3604010 | 26.8 | 90.3 | 88.0 | 31.7 | 27.8 |
| | ±1.5 | ±3.1 | ±4.8 | ±3.0 | ±1.6 |
| PUMA5101301 | 26.8 | 87.3 | 86.2 | 27.6 | 27.6 |
| | ±1.5 | ±3.0 | ±4.8 | ±2.6 | ±1.6 |
| PUMA5151255 | 26.8 | 73.2 | 73.9 | 34.9 | 26.8 |
| | ±1.5 | ±2.8 | ±4.7 | ±3.7 | ±1.6 |

TABLE 22. Squared Error (with standard deviations). Sum Query. PUMS datasets. Lap Mechanism ($\epsilon = 1$).

| Dataset | olsalg Total | olsalg Max | nnlsalg Total | nnlsalg Max | maxalg Total | maxalg Max | seqalg Total | seqalg Max | weightalg Total | weightalg Max |
|---|---|---|---|---|---|---|---|---|---|---|
| PUMA0101301 | 995.9 | 128.6 | 505.6 | 70.7 | 535.3 | 86.8 | 598.3 | 141.0 | 513.0 | 142.6 |
|  | ±21.1 | ±7.9 | ±10.4 | ±3.9 | ±46.0 | ±21.2 | ±24.2 | ±12.6 | ±14.5 | ±5.9 |
| PUMA0800803 | 995.9 | 128.6 | 662.4 | 143.4 | 699.6 | 142.0 | 680.0 | 136.9 | 642.3 | 134.8 |
|  | ±21.1 | ±7.9 | ±13.2 | ±6.7 | ±45.3 | ±17.9 | ±34.7 | ±20.0 | ±17.8 | ±7.2 |
| PUMA1304600 | 995.9 | 128.6 | 682.1 | 87.5 | 718.7 | 96.8 | 782.3 | 151.1 | 790.2 | 158.6 |
|  | ±21.1 | ±7.9 | ±13.1 | ±4.5 | ±41.1 | ±13.3 | ±44.8 | ±21.6 | ±20.2 | ±7.4 |
| PUMA1703529 | 995.9 | 128.6 | 669.6 | 101.9 | 710.6 | 100.6 | 730.8 | 142.4 | 729.9 | 158.1 |
|  | ±21.1 | ±7.9 | ±13.0 | ±5.1 | ±44.5 | ±14.8 | ±48.0 | ±26.0 | ±18.6 | ±7.5 |
| PUMA1703531 | 995.9 | 128.6 | 444.9 | 72.3 | 499.9 | 77.2 | 464.5 | 141.8 | 387.0 | 86.2 |
|  | ±21.1 | ±7.9 | ±9.6 | ±3.8 | ±36.4 | ±10.5 | ±17.8 | ±11.5 | ±12.7 | ±5.2 |
| PUMA1901700 | 995.9 | 128.6 | 736.9 | 107.9 | 729.2 | 98.7 | 731.7 | 126.6 | 869.0 | 185.2 |
|  | ±21.1 | ±7.9 | ±14.1 | ±5.5 | ±45.0 | ±16.0 | ±37.1 | ±15.4 | ±21.8 | ±9.2 |
| PUMA2401004 | 995.9 | 128.6 | 843.7 | 224.8 | 854.5 | 205.0 | 823.1 | 145.8 | 636.1 | 122.7 |
|  | ±21.1 | ±7.9 | ±16.0 | ±9.6 | ±49.3 | ±26.6 | ±49.1 | ±19.3 | ±17.2 | ±6.0 |
| PUMA2602702 | 995.9 | 128.6 | 613.1 | 99.9 | 594.9 | 95.9 | 695.3 | 130.5 | 686.7 | 155.5 |
|  | ±21.1 | ±7.9 | ±12.1 | ±4.8 | ±46.8 | ±18.0 | ±34.6 | ±16.3 | ±18.2 | ±7.0 |
| PUMA2801100 | 995.9 | 128.6 | 502.2 | 78.8 | 522.8 | 74.0 | 527.2 | 125.3 | 390.3 | 85.6 |
|  | ±21.1 | ±7.9 | ±10.6 | ±3.9 | ±32.8 | ±18.4 | ±26.2 | ±13.7 | ±12.5 | ±5.3 |
| PUMA2901901 | 995.9 | 128.6 | 595.2 | 94.1 | 606.8 | 105.3 | 676.0 | 125.4 | 660.7 | 140.8 |
|  | ±21.1 | ±7.9 | ±11.9 | ±4.6 | ±35.7 | ±14.4 | ±36.0 | ±12.5 | ±18.2 | ±7.7 |
| PUMA3200405 | 995.9 | 128.6 | 821.3 | 118.0 | 892.0 | 150.7 | 886.9 | 143.4 | 795.7 | 137.1 |
|  | ±21.1 | ±7.9 | ±15.7 | ±6.5 | ±67.3 | ±39.1 | ±37.8 | ±15.9 | ±20.1 | ±8.3 |
| PUMA3603710 | 995.9 | 128.6 | 904.6 | 130.5 | 868.5 | 166.7 | 773.4 | 127.3 | 811.8 | 136.9 |
|  | ±21.1 | ±7.9 | ±16.9 | ±6.8 | ±63.0 | ±26.5 | ±31.1 | ±13.4 | ±20.6 | ±8.6 |
| PUMA3604010 | 995.9 | 128.6 | 726.3 | 115.1 | 715.9 | 145.3 | 760.1 | 129.0 | 825.6 | 155.7 |
|  | ±21.1 | ±7.9 | ±13.7 | ±6.0 | ±42.4 | ±26.8 | ±33.0 | ±15.9 | ±21.0 | ±8.7 |
| PUMA5101301 | 995.9 | 128.6 | 747.0 | 157.6 | 895.4 | 205.8 | 810.9 | 138.7 | 797.2 | 173.8 |
|  | ±21.1 | ±7.9 | ±14.3 | ±7.4 | ±72.0 | ±38.5 | ±39.0 | ±16.7 | ±20.4 | ±9.4 |
| PUMA5151255 | 995.9 | 128.6 | 874.0 | 153.5 | 976.9 | 213.0 | 795.4 | 141.3 | 607.1 | 115.6 |
|  | ±21.1 | ±7.9 | ±16.3 | ±7.5 | ±63.5 | ±30.3 | ±38.1 | ±16.7 | ±17.5 | ±6.9 |

TABLE 23. Squared Errors (with standard deviations). Marg1 Query. PUMS datasets. Lap Mechanism ($\epsilon = 0.5$).

| Dataset | olsalg | | nnlsalg | | maxalg | | seqalg | | weightalg | |
|---|---|---|---|---|---|---|---|---|---|---|
| | Total | Max | Total | Max | Total | Max | Total | Max | Total | Max |
| PUMA0101301 | 2626.6 | 123.0 | 725.6 | 236.9 | 794.1 | 270.8 | 655.0 | 221.3 | 463.8 | 159.6 |
| | ±34.3 | ±8.3 | ±12.7 | ±8.1 | ±50.0 | ±34.6 | ±23.6 | ±14.7 | ±14.2 | ±4.8 |
| PUMA0800803 | 2626.6 | 123.0 | 982.6 | 254.0 | 1070.7 | 262.4 | 959.4 | 179.6 | 795.2 | 174.2 |
| | ±34.3 | ±8.3 | ±15.3 | ±9.0 | ±59.8 | ±26.7 | ±37.0 | ±19.1 | ±20.8 | ±8.2 |
| PUMA1304600 | 2626.6 | 123.0 | 1056.6 | 233.0 | 1141.7 | 261.0 | 989.6 | 157.0 | 866.7 | 154.0 |
| | ±34.3 | ±8.3 | ±15.8 | ±8.7 | ±61.9 | ±31.5 | ±44.2 | ±18.7 | ±21.5 | ±7.0 |
| PUMA1703529 | 2626.6 | 123.0 | 1017.0 | 255.2 | 1053.7 | 236.1 | 936.6 | 180.5 | 784.3 | 238.7 |
| | ±34.3 | ±8.3 | ±15.9 | ±9.1 | ±44.8 | ±20.5 | ±55.5 | ±21.8 | ±20.6 | ±9.2 |
| PUMA1703531 | 2626.6 | 123.0 | 644.0 | 275.4 | 679.4 | 255.9 | 466.5 | 264.2 | 232.5 | 98.0 |
| | ±34.3 | ±8.3 | ±12.0 | ±8.4 | ±40.8 | ±21.9 | ±18.5 | ±14.6 | ±11.8 | ±5.6 |
| PUMA1901700 | 2626.6 | 123.0 | 1137.1 | 218.9 | 1270.6 | 248.5 | 1069.0 | 161.2 | 860.8 | 113.5 |
| | ±34.3 | ±8.3 | ±16.6 | ±8.5 | ±58.9 | ±24.7 | ±42.4 | ±18.7 | ±20.8 | ±7.1 |
| PUMA2401004 | 2626.6 | 123.0 | 1348.6 | 208.8 | 1301.5 | 201.4 | 1532.4 | 123.3 | 1368.2 | 110.3 |
| | ±34.3 | ±8.3 | ±17.9 | ±8.4 | ±56.5 | ±23.4 | ±57.8 | ±15.5 | ±27.8 | ±7.4 |
| PUMA2602702 | 2626.6 | 123.0 | 855.8 | 273.2 | 823.0 | 229.4 | 731.5 | 192.3 | 546.5 | 106.4 |
| | ±34.3 | ±8.3 | ±14.4 | ±9.4 | ±54.6 | ±25.4 | ±33.8 | ±18.5 | ±16.9 | ±6.4 |
| PUMA2801100 | 2626.6 | 123.0 | 712.4 | 265.2 | 728.7 | 260.3 | 600.2 | 244.6 | 345.3 | 96.9 |
| | ±34.3 | ±8.3 | ±12.8 | ±8.8 | ±49.0 | ±34.5 | ±26.4 | ±18.3 | ±12.4 | ±5.4 |
| PUMA2901901 | 2626.6 | 123.0 | 867.5 | 278.3 | 821.8 | 244.4 | 778.2 | 200.6 | 549.8 | 107.5 |
| | ±34.3 | ±8.3 | ±14.3 | ±9.4 | ±43.2 | ±27.2 | ±36.0 | ±18.3 | ±16.0 | ±5.0 |
| PUMA3200405 | 2626.6 | 123.0 | 1407.8 | 210.4 | 1417.3 | 232.2 | 1403.1 | 154.6 | 1246.5 | 136.0 |
| | ±34.3 | ±8.3 | ±19.1 | ±8.6 | ±67.8 | ±33.0 | ±44.3 | ±15.5 | ±26.1 | ±7.7 |
| PUMA3603710 | 2626.6 | 123.0 | 1617.8 | 156.4 | 1788.8 | 208.4 | 1572.7 | 129.6 | 1506.5 | 151.0 |
| | ±34.3 | ±8.3 | ±21.5 | ±6.9 | ±86.9 | ±28.0 | ±42.6 | ±12.1 | ±31.1 | ±8.7 |
| PUMA3604010 | 2626.6 | 123.0 | 1092.7 | 213.7 | 1106.5 | 223.9 | 1136.2 | 154.5 | 969.6 | 122.3 |
| | ±34.3 | ±8.3 | ±15.7 | ±8.2 | ±43.7 | ±21.9 | ±38.1 | ±15.6 | ±23.9 | ±6.1 |
| PUMA5101301 | 2626.6 | 123.0 | 1093.7 | 241.6 | 1079.9 | 235.9 | 1049.1 | 124.3 | 941.8 | 110.5 |
| | ±34.3 | ±8.3 | ±16.2 | ±9.0 | ±59.5 | ±35.6 | ±35.5 | ±12.8 | ±23.2 | ±7.1 |
| PUMA5151255 | 2626.6 | 123.0 | 1388.6 | 183.6 | 1378.6 | 200.2 | 1523.9 | 133.0 | 1430.8 | 134.5 |
| | ±34.3 | ±8.3 | ±18.7 | ±7.9 | ±72.5 | ±28.8 | ±46.6 | ±15.4 | ±28.3 | ±8.7 |

TABLE 24. Squared Errors (with standard deviations). Marg2 Query. PUMS datasets. Lap Mechanism ($\epsilon = 0.5$).

| Dataset | olsalg | | nnlsalg | | maxalg | | seqalg | | weightalg | |
|---|---|---|---|---|---|---|---|---|---|---|
| | Total | Max | Total | Max | Total | Max | Total | Max | Total | Max |
| PUMA0101301 | 23906.2 | 142.9 | 809.0 | 135.6 | 910.7 | 144.6 | 782.9 | 179.7 | 731.3 | 209.8 |
| | ±105.7 | ±10.3 | ±11.2 | ±3.3 | ±48.1 | ±12.0 | ±19.3 | ±5.1 | ±15.4 | ±3.7 |
| PUMA0800803 | 23906.2 | 142.9 | 1179.8 | 107.5 | 1235.3 | 125.4 | 1171.7 | 189.8 | 1123.8 | 141.9 |
| | ±105.7 | ±10.3 | ±14.7 | ±6.4 | ±52.9 | ±17.4 | ±33.2 | ±14.1 | ±22.3 | ±8.7 |
| PUMA1304600 | 23906.2 | 142.9 | 1313.0 | 111.9 | 1385.5 | 142.3 | 1049.1 | 126.3 | 1264.4 | 136.0 |
| | ±105.7 | ±10.3 | ±14.3 | ±5.4 | ±50.1 | ±20.8 | ±25.4 | ±13.8 | ±20.8 | ±6.9 |
| PUMA1703529 | 23906.2 | 142.9 | 1243.8 | 105.3 | 1257.2 | 96.7 | 1019.1 | 114.3 | 1285.7 | 160.5 |
| | ±105.7 | ±10.3 | ±15.3 | ±6.3 | ±44.3 | ±12.2 | ±33.7 | ±13.6 | ±21.9 | ±4.3 |
| PUMA1703531 | 23906.2 | 142.9 | 562.2 | 94.9 | 599.0 | 72.1 | 429.9 | 112.9 | 409.8 | 78.8 |
| | ±105.7 | ±10.3 | ±8.4 | ±4.9 | ±32.8 | ±11.0 | ±12.3 | ±9.4 | ±12.0 | ±4.9 |
| PUMA1901700 | 23906.2 | 142.9 | 1516.1 | 115.9 | 1665.9 | 129.7 | 1312.1 | 156.9 | 1617.1 | 205.0 |
| | ±105.7 | ±10.3 | ±16.6 | ±4.9 | ±64.7 | ±17.9 | ±29.9 | ±13.8 | ±25.1 | ±8.6 |
| PUMA2401004 | 23906.2 | 142.9 | 1954.4 | 130.0 | 1971.8 | 147.8 | 1983.4 | 311.3 | 1760.1 | 168.9 |
| | ±105.7 | ±10.3 | ±19.4 | ±6.2 | ±59.9 | ±18.7 | ±51.6 | ±25.6 | ±26.7 | ±8.8 |
| PUMA2602702 | 23906.2 | 142.9 | 977.2 | 100.0 | 956.4 | 109.4 | 843.4 | 121.7 | 930.1 | 156.2 |
| | ±105.7 | ±10.3 | ±13.5 | ±4.7 | ±53.3 | ±19.3 | ±25.6 | ±11.3 | ±19.1 | ±6.6 |
| PUMA2801100 | 23906.2 | 142.9 | 686.9 | 97.5 | 705.7 | 79.0 | 534.2 | 92.7 | 516.0 | 78.7 |
| | ±105.7 | ±10.3 | ±9.9 | ±5.1 | ±33.6 | ±10.0 | ±16.2 | ±8.0 | ±12.2 | ±5.1 |
| PUMA2901901 | 23906.2 | 142.9 | 944.4 | 100.4 | 919.2 | 103.2 | 809.4 | 131.6 | 888.2 | 138.2 |
| | ±105.7 | ±10.3 | ±12.5 | ±4.6 | ±40.2 | ±18.0 | ±26.7 | ±14.1 | ±18.2 | ±6.6 |
| PUMA3200405 | 23906.2 | 142.9 | 2189.2 | 119.6 | 2191.5 | 134.7 | 1918.5 | 142.3 | 2336.1 | 259.1 |
| | ±105.7 | ±10.3 | ±20.5 | ±6.9 | ±77.4 | ±22.2 | ±31.4 | ±11.2 | ±29.7 | ±14.2 |
| PUMA3603710 | 23906.2 | 142.9 | 2884.1 | 119.2 | 3088.6 | 149.1 | 2484.2 | 140.7 | 2870.4 | 166.1 |
| | ±105.7 | ±10.3 | ±24.6 | ±3.5 | ±103.3 | ±29.8 | ±33.8 | ±6.1 | ±33.0 | ±3.9 |
| PUMA3604010 | 23906.2 | 142.9 | 1432.5 | 105.9 | 1442.3 | 120.6 | 1262.1 | 122.7 | 1448.6 | 194.0 |
| | ±105.7 | ±10.3 | ±14.5 | ±3.8 | ±42.7 | ±17.0 | ±22.8 | ±7.7 | ±24.0 | ±10.5 |
| PUMA5101301 | 23906.2 | 142.9 | 1474.7 | 108.3 | 1498.6 | 101.8 | 1394.5 | 203.4 | 1392.9 | 153.2 |
| | ±105.7 | ±10.3 | ±16.4 | ±6.5 | ±58.3 | ±15.8 | ±32.3 | ±15.2 | ±24.4 | ±8.0 |
| PUMA5151255 | 23906.2 | 142.9 | 2239.7 | 130.3 | 2274.1 | 124.3 | 2079.0 | 178.5 | 2123.0 | 172.8 |
| | ±105.7 | ±10.3 | ±21.4 | ±7.2 | ±79.8 | ±16.4 | ±37.8 | ±16.9 | ±29.4 | ±12.5 |

TABLE 25. Squared Errors (with standard deviations). Id Query. PUMS datasets. Lap Mechanism ($\epsilon = 0.5$).

| Dataset | olsalg | nnlsalg | maxalg | seqalg | weightalg |
|---|---|---|---|---|---|
| PUMA0101301 | 107.2 | 547.2 | 500.3 | 106.7 | 112.5 |
| | ±5.9 | ±14.5 | ±40.8 | ±10.7 | ±6.6 |
| PUMA0800803 | 107.2 | 446.1 | 571.7 | 120.3 | 107.2 |
| | ±5.9 | ±13.4 | ±55.5 | ±13.5 | ±6.4 |
| PUMA1304600 | 107.2 | 408.1 | 426.3 | 120.8 | 109.8 |
| | ±5.9 | ±12.9 | ±40.9 | ±16.7 | ±6.4 |
| PUMA1703529 | 107.2 | 435.3 | 426.3 | 134.9 | 110.9 |
| | ±5.9 | ±13.3 | ±36.8 | ±19.3 | ±6.5 |
| PUMA1703531 | 107.2 | 584.0 | 677.4 | 111.4 | 108.1 |
| | ±5.9 | ±14.9 | ±53.8 | ±8.6 | ±6.6 |
| PUMA1901700 | 107.2 | 395.1 | 443.6 | 119.1 | 110.4 |
| | ±5.9 | ±12.8 | ±42.0 | ±13.4 | ±6.4 |
| PUMA2401004 | 107.2 | 369.3 | 329.0 | 109.6 | 107.5 |
| | ±5.9 | ±12.4 | ±32.2 | ±15.2 | ±6.3 |
| PUMA2602702 | 107.2 | 467.8 | 472.0 | 146.0 | 109.2 |
| | ±5.9 | ±13.7 | ±45.0 | ±16.6 | ±6.4 |
| PUMA2801100 | 107.2 | 543.7 | 558.2 | 117.8 | 110.8 |
| | ±5.9 | ±14.5 | ±42.1 | ±13.6 | ±6.5 |
| PUMA2901901 | 107.2 | 485.2 | 464.4 | 126.5 | 110.8 |
| | ±5.9 | ±13.9 | ±38.9 | ±16.5 | ±6.5 |
| PUMA3200405 | 107.2 | 329.1 | 301.0 | 122.9 | 108.4 |
| | ±5.9 | ±11.7 | ±37.8 | ±12.5 | ±6.3 |
| PUMA3603710 | 107.2 | 300.3 | 293.3 | 85.7 | 108.8 |
| | ±5.9 | ±11.3 | ±43.4 | ±8.5 | ±6.4 |
| PUMA3604010 | 107.2 | 399.9 | 386.5 | 129.8 | 111.3 |
| | ±5.9 | ±12.9 | ±32.5 | ±14.1 | ±6.5 |
| PUMA5101301 | 107.2 | 396.1 | 369.5 | 139.2 | 107.2 |
| | ±5.9 | ±12.7 | ±47.2 | ±16.3 | ±6.3 |
| PUMA5151255 | 107.2 | 330.7 | 280.3 | 139.1 | 107.8 |
| | ±5.9 | ±11.8 | ±43.7 | ±15.4 | ±6.3 |

TABLE 26. Squared Error (with standard deviations). Sum Query. PUMS datasets. Lap Mechanism ($\epsilon = 0.5$).

| Dataset | olsalg | | nnlsalg | | maxalg | | seqalg | | weightalg | |
|---|---|---|---|---|---|---|---|---|---|---|
| | Total | Max | Total | Max | Total | Max | Total | Max | Total | Max |
| PUMA0101301 | 24896.5 | 3215.6 | 10104.5 | 1708.4 | 9642.4 | 1786.5 | 10263.6 | 3245.0 | 7632.6 | 2034.3 |
| | ±526.6 | ±198.2 | ±224.8 | ±88.2 | ±1335.2 | ±812.4 | ±553.9 | ±324.2 | ±266.6 | ±82.8 |
| PUMA0800803 | 24896.5 | 3215.6 | 12272.6 | 2263.3 | 12433.7 | 3064.4 | 14411.4 | 4356.0 | 10115.3 | 2199.8 |
| | ±526.6 | ±198.2 | ±259.8 | ±106.9 | ±1894.4 | ±1212.6 | ±2215.9 | ±1524.3 | ±330.9 | ±118.5 |
| PUMA1304600 | 24896.5 | 3215.6 | 12120.8 | 1812.5 | 11921.0 | 2081.1 | 13247.7 | 2884.2 | 12740.9 | 3563.4 |
| | ±526.6 | ±198.2 | ±256.2 | ±100.0 | ±1621.6 | ±755.3 | ±2037.6 | ±719.6 | ±399.9 | ±185.2 |
| PUMA1703529 | 24896.5 | 3215.6 | 12838.6 | 1805.1 | 13531.6 | 2312.8 | 23113.3 | 6541.3 | 13386.5 | 3221.7 |
| | ±526.6 | ±198.2 | ±266.7 | ±102.1 | ±2620.2 | ±1701.4 | ±4924.2 | ±4081.2 | ±395.3 | ±147.8 |
| PUMA1703531 | 24896.5 | 3215.6 | 9061.0 | 1801.3 | 10780.0 | 2611.4 | 7140.4 | 4292.3 | 4327.4 | 1710.4 |
| | ±526.6 | ±198.2 | ±212.9 | ±88.6 | ±1525.7 | ±755.5 | ±584.3 | ±473.2 | ±212.9 | ±94.4 |
| PUMA1901700 | 24896.5 | 3215.6 | 13529.0 | 2218.4 | 16867.9 | 3205.4 | 13848.3 | 4556.5 | 13439.0 | 3091.4 |
| | ±526.6 | ±198.2 | ±278.5 | ±111.0 | ±2285.1 | ±956.0 | ±2074.6 | ±1715.9 | ±412.2 | ±176.3 |
| PUMA2401004 | 24896.5 | 3215.6 | 15690.2 | 3323.8 | 12290.1 | 2493.8 | NA | NA | 14873.8 | 3085.1 |
| | ±526.6 | ±198.2 | ±309.9 | ±149.6 | ±1532.5 | ±591.9 | NA | NA | ±430.0 | ±171.5 |
| PUMA2602702 | 24896.5 | 3215.6 | 11788.9 | 1642.1 | 12902.9 | 2684.5 | 13206.5 | 2797.6 | 11656.5 | 3066.2 |
| | ±526.6 | ±198.2 | ±252.4 | ±82.8 | ±2428.1 | ±1818.0 | ±851.2 | ±390.7 | ±368.5 | ±158.8 |
| PUMA2801100 | 24896.5 | 3215.6 | 10658.4 | 1754.2 | 12538.7 | 2648.4 | 9076.9 | 3191.2 | 6842.2 | 2483.3 |
| | ±526.6 | ±198.2 | ±237.3 | ±92.6 | ±2033.0 | ±1364.8 | ±1023.3 | ±636.7 | ±281.6 | ±153.9 |
| PUMA2901901 | 24896.5 | 3215.6 | 11494.9 | 1741.4 | 9753.9 | 2134.0 | 13258.2 | 2941.7 | 10340.4 | 3165.1 |
| | ±526.6 | ±198.2 | ±247.7 | ±96.2 | ±1364.1 | ±939.0 | ±1759.8 | ±796.4 | ±339.4 | ±176.9 |
| PUMA3200405 | 24896.5 | 3215.6 | 15740.1 | 2388.3 | 16015.6 | 2612.2 | 15100.4 | 2439.5 | 17778.8 | 3758.6 |
| | ±526.6 | ±198.2 | ±314.0 | ±123.7 | ±1910.9 | ±754.6 | ±1151.0 | ±483.9 | ±472.0 | ±211.4 |
| PUMA3603710 | 24896.5 | 3215.6 | 17655.8 | 2756.2 | 17234.3 | 4363.1 | 14934.9 | 2884.7 | 16438.2 | 4355.5 |
| | ±526.6 | ±198.2 | ±340.1 | ±140.5 | ±1885.8 | ±1173.0 | ±944.3 | ±437.5 | ±465.3 | ±221.7 |
| PUMA3604010 | 24896.5 | 3215.6 | 11373.5 | 1971.8 | 14373.0 | 3585.1 | 15183.5 | 6053.7 | 9585.6 | 2336.1 |
| | ±526.6 | ±198.2 | ±243.6 | ±100.6 | ±2347.7 | ±799.9 | ±2446.4 | ±1818.7 | ±318.2 | ±133.7 |
| PUMA5101301 | 24896.5 | 3215.6 | 12946.4 | 2040.5 | 16033.9 | 3618.1 | 14648.9 | 2980.5 | 12533.1 | 2613.3 |
| | ±526.6 | ±198.2 | ±269.3 | ±99.5 | ±2164.8 | ±1039.4 | ±1320.7 | ±477.4 | ±389.9 | ±143.4 |
| PUMA5151255 | 24896.5 | 3215.6 | 15451.4 | 2277.8 | 14946.2 | 2335.3 | 18403.9 | 4421.7 | 17045.8 | 3732.4 |
| | ±526.6 | ±198.2 | ±299.0 | ±111.6 | ±1995.7 | ±1283.8 | ±2462.0 | ±1386.8 | ±455.8 | ±210.5 |

TABLE 27. Squared Errors (with standard deviations). Marg1 Query. PUMS datasets. Lap Mechanism ($\epsilon = 0.1$).

| Dataset | olsalg | | nnlsalg | | maxalg | | seqalg | | weightalg | |
|---|---|---|---|---|---|---|---|---|---|---|
| | Total | Max | Total | Max | Total | Max | Total | Max | Total | Max |
| PUMA0101301 | 65664.1 | 3075.7 | 13618.3 | 5141.4 | 12947.3 | 4395.2 | 14169.5 | 8114.8 | 4949.6 | 2036.7 |
| | ±856.9 | ±207.6 | ±265.2 | ±171.8 | ±1779.9 | ±1258.0 | ±710.6 | ±552.1 | ±257.0 | ±109.7 |
| PUMA0800803 | 65664.1 | 3075.7 | 17302.2 | 5898.5 | 23152.7 | 9272.9 | 22604.8 | 8350.0 | 9168.6 | 2488.8 |
| | ±856.9 | ±207.6 | ±307.3 | ±197.0 | ±3745.8 | ±2955.5 | ±2651.9 | ±1504.3 | ±322.7 | ±140.7 |
| PUMA1304600 | 65664.1 | 3075.7 | 17227.7 | 6608.7 | 20876.1 | 7232.8 | 13908.9 | 5583.9 | 7834.3 | 2652.8 |
| | ±856.9 | ±207.6 | ±317.2 | ±217.4 | ±2272.5 | ±1183.0 | ±1901.8 | ±1061.2 | ±336.7 | ±154.9 |
| PUMA1703529 | 65664.1 | 3075.7 | 18583.7 | 7148.5 | 18695.9 | 7183.1 | 21515.9 | 10223.1 | 8709.0 | 2650.2 |
| | ±856.9 | ±207.6 | ±331.2 | ±232.3 | ±3023.3 | ±2228.9 | ±4189.6 | ±3443.0 | ±328.2 | ±153.8 |
| PUMA1703531 | 65664.1 | 3075.7 | 11527.0 | 4025.8 | 14654.2 | 4099.9 | 13799.8 | 9634.8 | 3623.6 | 1572.5 |
| | ±856.9 | ±207.6 | ±234.9 | ±140.2 | ±2447.6 | ±841.7 | ±920.3 | ±816.6 | ±212.2 | ±85.9 |
| PUMA1901700 | 65664.1 | 3075.7 | 20261.4 | 5898.7 | 24119.9 | 6681.5 | 15818.3 | 4194.7 | 14173.4 | 4738.1 |
| | ±856.9 | ±207.6 | ±340.4 | ±205.9 | ±3161.8 | ±1720.6 | ±2428.3 | ±1077.2 | ±427.6 | ±165.0 |
| PUMA2401004 | 65664.1 | 3075.7 | 20905.0 | 6110.6 | 25347.4 | 9196.5 | NA | NA | 14298.2 | 2593.0 |
| | ±856.9 | ±207.6 | ±355.5 | ±219.8 | ±3640.7 | ±2861.7 | NA | NA | ±454.6 | ±156.8 |
| PUMA2602702 | 65664.1 | 3075.7 | 16674.9 | 7245.2 | 21024.3 | 9251.2 | 14273.5 | 7653.5 | 5933.2 | 2502.9 |
| | ±856.9 | ±207.6 | ±316.2 | ±228.0 | ±2925.9 | ±1894.1 | ±1014.7 | ±734.0 | ±302.6 | ±150.8 |
| PUMA2801100 | 65664.1 | 3075.7 | 14301.5 | 5872.8 | 14027.9 | 4455.6 | 11696.7 | 7521.4 | 5157.7 | 2406.6 |
| | ±856.9 | ±207.6 | ±283.8 | ±197.2 | ±2102.9 | ±708.5 | ±1756.1 | ±1579.4 | ±274.7 | ±148.7 |
| PUMA2901901 | 65664.1 | 3075.7 | 16288.1 | 6994.3 | 15740.0 | 7336.8 | 14045.6 | 7836.8 | 5810.7 | 2504.9 |
| | ±856.9 | ±207.6 | ±309.5 | ±221.6 | ±1973.1 | ±1226.4 | ±1511.6 | ±1276.2 | ±293.7 | ±151.8 |
| PUMA3200405 | 65664.1 | 3075.7 | 24070.3 | 5824.6 | 25938.7 | 6332.2 | 20598.9 | 4370.2 | 13952.7 | 2838.4 |
| | ±856.9 | ±207.6 | ±385.7 | ±211.4 | ±2629.4 | ±1481.1 | ±1450.7 | ±796.4 | ±432.2 | ±163.2 |
| PUMA3603710 | 65664.1 | 3075.7 | 27996.4 | 3696.8 | 26465.1 | 4220.0 | 28859.3 | 4474.6 | 25595.0 | 5328.0 |
| | ±856.9 | ±207.6 | ±412.8 | ±175.5 | ±2164.4 | ±1013.1 | ±1207.7 | ±443.4 | ±636.8 | ±242.6 |
| PUMA3604010 | 65664.1 | 3075.7 | 15549.5 | 5083.6 | 21388.8 | 9090.9 | 14231.8 | 6216.5 | 7884.6 | 2332.7 |
| | ±856.9 | ±207.6 | ±286.7 | ±173.8 | ±3358.4 | ±2540.9 | ±3120.2 | ±2530.2 | ±321.5 | ±123.7 |
| PUMA5101301 | 65664.1 | 3075.7 | 17886.8 | 6799.0 | 17689.4 | 6431.7 | 18006.2 | 7912.6 | 8521.4 | 2602.1 |
| | ±856.9 | ±207.6 | ±328.1 | ±222.8 | ±2057.8 | ±1296.3 | ±1483.5 | ±1077.3 | ±352.3 | ±153.2 |
| PUMA5151255 | 65664.1 | 3075.7 | 21460.6 | 5928.5 | 22491.7 | 4344.7 | 23194.9 | 5064.6 | 15497.7 | 2792.9 |
| | ±856.9 | ±207.6 | ±353.2 | ±213.6 | ±2941.4 | ±984.4 | ±2668.4 | ±1066.6 | ±471.0 | ±167.7 |

TABLE 28. Squared Errors (with standard deviations). Marg2 Query. PUMS datasets. Lap Mechanism ($\epsilon = 0.1$).

| Dataset | olsalg Total | Max | nnlsalg Total | Max | maxalg Total | Max | seqalg Total | Max | weightalg Total | Max |
|---|---|---|---|---|---|---|---|---|---|---|
| PUMA0101301 | 597654.9 | 3571.4 | 13058.7 | 3275.2 | 12947.8 | 2686.7 | 12159.6 | 4872.7 | 8158.8 | 2002.2 |
| | ±2643.1 | ±258.6 | ±212.1 | ±139.8 | ±1344.0 | ±534.8 | ±483.3 | ±382.6 | ±265.3 | ±53.1 |
| PUMA0800803 | 597654.9 | 3571.4 | 16513.8 | 2093.3 | 19653.0 | 2554.7 | 16523.9 | 2745.4 | 13274.0 | 2113.7 |
| | ±2643.1 | ±258.6 | ±218.7 | ±57.2 | ±2044.3 | ±517.1 | ±1327.2 | ±339.2 | ±308.9 | ±116.5 |
| PUMA1304600 | 597654.9 | 3571.4 | 16512.9 | 2471.6 | 19550.9 | 2443.4 | 11209.4 | 1850.4 | 13004.8 | 2832.2 |
| | ±2643.1 | ±258.6 | ±252.0 | ±127.2 | ±2103.0 | ±766.4 | ±1090.8 | ±619.7 | ±372.3 | ±154.7 |
| PUMA1703529 | 597654.9 | 3571.4 | 18223.1 | 2469.7 | 17509.2 | 3273.1 | 18088.1 | 3634.6 | 15570.9 | 3262.1 |
| | ±2643.1 | ±258.6 | ±267.3 | ±128.6 | ±2257.0 | ±1119.2 | ±2026.1 | ±846.3 | ±389.0 | ±123.8 |
| PUMA1703531 | 597654.9 | 3571.4 | 10469.3 | 3760.6 | 12319.8 | 5015.3 | 11123.4 | 7551.8 | 4729.1 | 1786.9 |
| | ±2643.1 | ±258.6 | ±200.1 | ±146.2 | ±1748.9 | ±1444.0 | ±695.8 | ±633.9 | ±217.2 | ±115.7 |
| PUMA1901700 | 597654.9 | 3571.4 | 20648.5 | 2575.2 | 25415.7 | 3731.7 | 15792.2 | 2338.8 | 18722.1 | 3023.7 |
| | ±2643.1 | ±258.6 | ±263.9 | ±85.9 | ±2715.8 | ±839.2 | ±1412.7 | ±719.2 | ±419.4 | ±103.2 |
| PUMA2401004 | 597654.9 | 3571.4 | 23552.1 | 2736.6 | 28577.1 | 5984.3 | NA | NA | 21406.2 | 2722.5 |
| | ±2643.1 | ±258.6 | ±297.3 | ±138.8 | ±3444.6 | ±2504.7 | NA | NA | ±461.3 | ±150.8 |
| PUMA2602702 | 597654.9 | 3571.4 | 15417.7 | 2472.0 | 17737.0 | 2903.3 | 12490.9 | 3014.8 | 12100.4 | 2944.3 |
| | ±2643.1 | ±258.6 | ±233.6 | ±106.5 | ±2589.0 | ±1479.7 | ±582.1 | ±353.3 | ±340.3 | ±139.3 |
| PUMA2801100 | 597654.9 | 3571.4 | 13627.0 | 3034.9 | 15786.7 | 3214.9 | 10505.2 | 3516.3 | 8371.2 | 2902.2 |
| | ±2643.1 | ±258.6 | ±244.3 | ±135.3 | ±2597.8 | ±1419.6 | ±965.0 | ±728.6 | ±310.7 | ±178.4 |
| PUMA2901901 | 597654.9 | 3571.4 | 14875.2 | 2616.8 | 15098.1 | 3670.4 | 12787.3 | 3334.6 | 10742.9 | 2868.6 |
| | ±2643.1 | ±258.6 | ±242.2 | ±128.8 | ±1833.4 | ±1328.1 | ±1298.7 | ±783.1 | ±319.5 | ±157.8 |
| PUMA3200405 | 597654.9 | 3571.4 | 28252.4 | 2907.8 | 28794.9 | 3166.4 | 21533.9 | 2275.9 | 27848.0 | 3658.7 |
| | ±2643.1 | ±258.6 | ±353.8 | ±139.2 | ±2598.2 | ±1216.4 | ±947.7 | ±436.1 | ±583.4 | ±218.5 |
| PUMA3603710 | 597654.9 | 3571.4 | 33268.3 | 2874.8 | 36198.8 | 3875.9 | 25180.8 | 2377.2 | 27205.8 | 2303.0 |
| | ±2643.1 | ±258.6 | ±358.4 | ±141.9 | ±2310.7 | ±1074.2 | ±642.8 | ±247.8 | ±497.9 | ±140.2 |
| PUMA3604010 | 597654.9 | 3571.4 | 14244.6 | 2455.6 | 15963.1 | 3455.2 | 12436.4 | 4212.1 | 10295.9 | 2073.2 |
| | ±2643.1 | ±258.6 | ±214.9 | ±123.9 | ±1570.4 | ±813.0 | ±1549.7 | ±1214.8 | ±309.5 | ±131.9 |
| PUMA5101301 | 597654.9 | 3571.4 | 17418.0 | 2445.3 | 18585.4 | 2733.3 | 14108.6 | 3130.0 | 14449.9 | 2557.0 |
| | ±2643.1 | ±258.6 | ±239.2 | ±97.4 | ±1980.0 | ±722.1 | ±720.0 | ±452.5 | ±365.9 | ±113.3 |
| PUMA5151255 | 597654.9 | 3571.4 | 24694.4 | 2627.0 | 26043.6 | 3246.3 | 23306.6 | 4012.0 | 23083.9 | 3001.6 |
| | ±2643.1 | ±258.6 | ±296.5 | ±126.8 | ±2410.3 | ±918.1 | ±1799.3 | ±1065.6 | ±458.9 | ±155.0 |

TABLE 29. Squared Errors (with standard deviations). Id Query. PUMS datasets. Lap Mechanism ($\epsilon = 0.1$).

| Dataset | olsalg | nnlsalg | maxalg | seqalg | weightalg |
|---|---|---|---|---|---|
| PUMA0101301 | 2680.2 | 15540.2 | 20191.3 | 3486.7 | 2738.3 |
| | ±147.2 | ±380.4 | ±2841.2 | ±396.5 | ±171.6 |
| PUMA0800803 | 2680.2 | 13761.1 | 13153.1 | 4435.7 | 2701.3 |
| | ±147.2 | ±363.5 | ±2622.8 | ±1003.9 | ±160.4 |
| PUMA1304600 | 2680.2 | 13594.4 | 20043.4 | 5018.9 | 2729.4 |
| | ±147.2 | ±361.8 | ±2638.0 | ±2324.5 | ±165.1 |
| PUMA1703529 | 2680.2 | 13502.4 | 10246.0 | 4174.4 | 2736.0 |
| | ±147.2 | ±361.4 | ±1979.8 | ±1681.4 | ±161.9 |
| PUMA1703531 | 2680.2 | 16585.6 | 21135.7 | 3336.8 | 2716.7 |
| | ±147.2 | ±390.2 | ±3283.1 | ±499.2 | ±172.9 |
| PUMA1901700 | 2680.2 | 12647.7 | 22102.9 | 3468.7 | 2702.0 |
| | ±147.2 | ±352.4 | ±3586.2 | ±1376.7 | ±159.3 |
| PUMA2401004 | 2680.2 | 11795.3 | 8553.0 | NA | 2635.9 |
| | ±147.2 | ±343.1 | ±1790.3 | NA | ±158.0 |
| PUMA2602702 | 2680.2 | 14423.8 | 12298.0 | 2602.7 | 2670.9 |
| | ±147.2 | ±370.3 | ±2152.6 | ±384.0 | ±164.4 |
| PUMA2801100 | 2680.2 | 15642.4 | 15246.7 | 5985.2 | 2748.1 |
| | ±147.2 | ±381.3 | ±2409.8 | ±1344.1 | ±167.5 |
| PUMA2901901 | 2680.2 | 14649.7 | 11092.6 | 3400.2 | 2664.6 |
| | ±147.2 | ±372.4 | ±1482.9 | ±978.1 | ±163.3 |
| PUMA3200405 | 2680.2 | 11154.8 | 10611.7 | 2968.6 | 2756.5 |
| | ±147.2 | ±334.4 | ±1679.4 | ±512.8 | ±161.9 |
| PUMA3603710 | 2680.2 | 10040.7 | 9504.2 | 3296.2 | 2731.3 |
| | ±147.2 | ±322.4 | ±1724.7 | ±413.7 | ±160.6 |
| PUMA3604010 | 2680.2 | 13625.9 | 10093.9 | 5440.1 | 2742.4 |
| | ±147.2 | ±362.5 | ±1664.1 | ±2001.7 | ±162.1 |
| PUMA5101301 | 2680.2 | 13157.3 | 12595.6 | 3420.9 | 2698.9 |
| | ±147.2 | ±357.3 | ±2072.2 | ±582.7 | ±161.8 |
| PUMA5151255 | 2680.2 | 11160.6 | 13465.7 | 2661.5 | 2738.0 |
| | ±147.2 | ±335.1 | ±2485.5 | ±698.7 | ±161.8 |

TABLE 30. Squared Error (with standard deviations). Sum Query. PUMS datasets. Lap Mechanism ($\epsilon = 0.1$).

## 2. zCDP Experiments

| Dataset | olsalg | | nnlsalg | | maxalg | | seqalg | | weightalg | |
|---|---|---|---|---|---|---|---|---|---|---|
| | Total | Max | Total | Max | Total | Max | Total | Max | Total | Max |
| Level00-1d | 198.8 | 2.2 | 10.7 | 6.3 | 10.9 | 6.5 | 10.3 | 8.0 | 2.5 | 1.7 |
| | ±0.9 | ±0.1 | ±0.2 | ±0.2 | ±0.3 | ±0.2 | ±0.2 | ±0.2 | ±0.1 | ±0.1 |
| Level01-1d | 198.8 | 2.2 | 112.6 | 2.1 | 112.9 | 2.1 | 112.4 | 2.1 | 112.4 | 2.1 |
| | ±0.9 | ±0.1 | ±0.6 | ±0.1 | ±0.6 | ±0.1 | ±0.6 | ±0.1 | ±0.6 | ±0.1 |
| Level16-1d | 198.8 | 2.2 | 198.8 | 2.2 | 201.5 | 2.9 | 198.8 | 2.2 | 198.8 | 2.2 |
| | ±0.9 | ±0.1 | ±0.9 | ±0.1 | ±2.4 | ±0.3 | ±0.9 | ±0.1 | ±0.9 | ±0.1 |
| Level32-1d | 198.8 | 2.2 | 198.8 | 2.2 | 197.8 | 2.3 | 198.8 | 2.2 | 198.8 | 2.2 |
| | ±0.9 | ±0.1 | ±0.9 | ±0.1 | ±1.2 | ±0.1 | ±0.9 | ±0.1 | ±0.9 | ±0.1 |
| SplitStairs-1d | 198.8 | 2.2 | 136.1 | 2.4 | 136.6 | 2.4 | 136.1 | 2.4 | 117.8 | 5.1 |
| | ±0.9 | ±0.1 | ±0.8 | ±0.1 | ±0.9 | ±0.1 | ±0.8 | ±0.1 | ±0.7 | ±0.1 |
| Stair-1d | 198.8 | 2.2 | 198.5 | 2.2 | 198.3 | 2.6 | 198.5 | 2.2 | 198.6 | 2.2 |
| | ±0.9 | ±0.1 | ±0.9 | ±0.1 | ±1.6 | ±0.2 | ±0.9 | ±0.1 | ±0.9 | ±0.1 |
| Step16-1d | 198.8 | 2.2 | 139.4 | 2.3 | 138.8 | 2.4 | 139.4 | 2.3 | 107.1 | 2.2 |
| | ±0.9 | ±0.1 | ±0.8 | ±0.1 | ±0.9 | ±0.1 | ±0.8 | ±0.1 | ±0.7 | ±0.1 |
| Step50-1d | 198.8 | 2.2 | 139.4 | 2.3 | 138.8 | 2.3 | 139.4 | 2.3 | 107.1 | 2.2 |
| | ±0.9 | ±0.1 | ±0.8 | ±0.1 | ±0.9 | ±0.1 | ±0.8 | ±0.1 | ±0.7 | ±0.1 |

TABLE 31. Squared Errors (with standard deviations). Id Query. 1-d datasets. Gauss Mechanism ($\rho = 0.5$).

| Dataset | olsalg | nnlsalg | maxalg | seqalg | weightalg |
|---|---|---|---|---|---|
| Level00-1d | 2.0 | 6.1 | 6.1 | 2.0 | 1.6 |
| | ±0.1 | ±0.2 | ±0.2 | ±0.1 | ±0.1 |
| Level01-1d | 2.0 | 2.0 | 1.9 | 2.0 | 2.0 |
| | ±0.1 | ±0.1 | ±0.1 | ±0.1 | ±0.1 |
| Level16-1d | 2.0 | 2.0 | 2.1 | 2.0 | 2.0 |
| | ±0.1 | ±0.1 | ±0.2 | ±0.1 | ±0.1 |
| Level32-1d | 2.0 | 2.0 | 2.1 | 2.0 | 2.0 |
| | ±0.1 | ±0.1 | ±0.1 | ±0.1 | ±0.1 |
| SplitStairs-1d | 2.0 | 2.1 | 2.0 | 2.0 | 2.0 |
| | ±0.1 | ±0.1 | ±0.1 | ±0.1 | ±0.1 |
| Stair-1d | 2.0 | 2.0 | 1.8 | 2.0 | 2.0 |
| | ±0.1 | ±0.1 | ±0.1 | ±0.1 | ±0.1 |
| Step16-1d | 2.0 | 2.1 | 2.1 | 2.0 | 2.0 |
| | ±0.1 | ±0.1 | ±0.1 | ±0.1 | ±0.1 |
| Step50-1d | 2.0 | 2.1 | 2.1 | 2.0 | 2.0 |
| | ±0.1 | ±0.1 | ±0.1 | ±0.1 | ±0.1 |

TABLE 32. Squared Error (with standard deviations). Sum Query. 1-d datasets. Gauss Mechanism ($\rho = 0.5$).

| Dataset | olsalg Total | olsalg Max | nnlsalg Total | nnlsalg Max | maxalg Total | maxalg Max | seqalg Total | seqalg Max | weightalg Total | weightalg Max |
|---|---|---|---|---|---|---|---|---|---|---|
| Level00-1d | 795.2 | 8.8 | 42.7 | 25.4 | 42.4 | 24.8 | 41.3 | 31.9 | 9.6 | 6.7 |
|  | ±3.5 | ±0.4 | ±0.9 | ±0.8 | ±1.3 | ±1.1 | ±0.9 | ±0.9 | ±0.4 | ±0.4 |
| Level01-1d | 795.2 | 8.8 | 252.9 | 9.8 | 251.4 | 9.8 | 250.1 | 9.8 | 246.4 | 8.3 |
|  | ±3.5 | ±0.4 | ±1.8 | ±0.5 | ±2.3 | ±0.6 | ±1.8 | ±0.5 | ±1.8 | ±0.4 |
| Level16-1d | 795.2 | 8.8 | 795.2 | 8.8 | 787.7 | 9.6 | 795.2 | 8.8 | 795.2 | 8.8 |
|  | ±3.5 | ±0.4 | ±3.5 | ±0.4 | ±6.3 | ±0.8 | ±3.5 | ±0.4 | ±3.5 | ±0.4 |
| Level32-1d | 795.2 | 8.8 | 795.2 | 8.8 | 791.3 | 10.1 | 795.2 | 8.8 | 795.2 | 8.8 |
|  | ±3.5 | ±0.4 | ±3.5 | ±0.4 | ±6.6 | ±0.8 | ±3.5 | ±0.4 | ±3.5 | ±0.4 |
| SplitStairs-1d | 795.2 | 8.8 | 534.9 | 9.5 | 534.2 | 9.4 | 534.8 | 9.5 | 496.4 | 16.8 |
|  | ±3.5 | ±0.4 | ±3.0 | ±0.4 | ±4.0 | ±0.5 | ±3.0 | ±0.4 | ±2.9 | ±0.6 |
| Stair-1d | 795.2 | 8.8 | 788.9 | 8.8 | 788.5 | 10.3 | 788.9 | 8.8 | 791.2 | 8.8 |
|  | ±3.5 | ±0.4 | ±3.5 | ±0.4 | ±6.6 | ±0.9 | ±3.5 | ±0.4 | ±3.5 | ±0.4 |
| Step16-1d | 795.2 | 8.8 | 557.7 | 9.3 | 553.8 | 9.5 | 557.7 | 9.3 | 485.9 | 10.4 |
|  | ±3.5 | ±0.4 | ±3.1 | ±0.4 | ±4.2 | ±0.6 | ±3.1 | ±0.4 | ±4.3 | ±0.8 |
| Step50-1d | 795.2 | 8.8 | 557.7 | 9.3 | 558.7 | 9.8 | 557.7 | 9.3 | 421.4 | 8.7 |
|  | ±3.5 | ±0.4 | ±3.1 | ±0.4 | ±4.2 | ±0.6 | ±3.1 | ±0.4 | ±2.6 | ±0.4 |

TABLE 33. Squared Errors (with standard deviations). Id Query. 1-d datasets. Gauss Mechanism ($\rho = 0.125$).

| Dataset | olsalg | nnlsalg | maxalg | seqalg | weightalg |
|---|---|---|---|---|---|
| Level00-1d | 7.9 | 24.2 | 24.5 | 8.0 | 6.1 |
|  | ±0.3 | ±0.7 | ±1.1 | ±0.3 | ±0.3 |
| Level01-1d | 7.9 | 9.2 | 8.7 | 8.0 | 8.0 |
|  | ±0.3 | ±0.4 | ±0.5 | ±0.3 | ±0.3 |
| Level16-1d | 7.9 | 7.9 | 7.8 | 8.0 | 7.9 |
|  | ±0.3 | ±0.3 | ±0.6 | ±0.3 | ±0.3 |
| Level32-1d | 7.9 | 7.9 | 7.8 | 8.0 | 7.9 |
|  | ±0.3 | ±0.3 | ±0.6 | ±0.3 | ±0.3 |
| SplitStairs-1d | 7.9 | 8.5 | 8.7 | 8.0 | 7.9 |
|  | ±0.3 | ±0.4 | ±0.5 | ±0.3 | ±0.3 |
| Stair-1d | 7.9 | 7.9 | 7.0 | 8.0 | 7.9 |
|  | ±0.3 | ±0.3 | ±0.6 | ±0.3 | ±0.3 |
| Step16-1d | 7.9 | 8.5 | 8.9 | 8.0 | 7.9 |
|  | ±0.3 | ±0.4 | ±0.5 | ±0.3 | ±0.3 |
| Step50-1d | 7.9 | 8.5 | 8.2 | 8.0 | 7.9 |
|  | ±0.3 | ±0.4 | ±0.5 | ±0.3 | ±0.3 |

TABLE 34. Squared Error (with standard deviations). Sum Query. 1-d datasets. Gauss Mechanism ($\rho = 0.125$).

| | olsalg | | nnlsalg | | maxalg | | seqalg | | weightalg | |
|---|---|---|---|---|---|---|---|---|---|---|
| **Dataset** | Total | Max | Total | Max | Total | Max | Total | Max | Total | Max |
| Level00-1d | 19881.2 | 221.2 | 1068.5 | 633.9 | 1127.8 | 676.4 | 1033.3 | 796.6 | 235.6 | 164.6 |
| | ±88.4 | ±8.9 | ±23.3 | ±21.0 | ±44.9 | ±41.2 | ±23.5 | ±22.5 | ±9.7 | ±9.0 |
| Level01-1d | 19881.2 | 221.2 | 1855.9 | 411.3 | 1841.0 | 412.4 | 1704.9 | 435.4 | 1297.2 | 207.0 |
| | ±88.4 | ±8.9 | ±26.7 | ±17.1 | ±36.7 | ±24.0 | ±25.7 | ±17.6 | ±21.0 | ±10.2 |
| Level16-1d | 19881.2 | 221.2 | 15398.7 | 208.7 | 15426.4 | 205.1 | 15392.9 | 208.7 | 15431.7 | 207.1 |
| | ±88.4 | ±8.9 | ±63.7 | ±10.3 | ±88.4 | ±14.3 | ±63.7 | ±10.3 | ±64.7 | ±10.2 |
| Level32-1d | 19881.2 | 221.2 | 19474.4 | 217.5 | 19480.1 | 235.4 | 19476.2 | 217.7 | 19541.2 | 218.6 |
| | ±88.4 | ±8.9 | ±82.9 | ±8.5 | ±144.6 | ±20.8 | ±82.9 | ±8.5 | ±83.2 | ±8.6 |
| SplitStairs-1d | 19881.2 | 221.2 | 11138.3 | 247.3 | 11121.5 | 257.9 | 11123.5 | 248.3 | 11862.4 | 299.9 |
| | ±88.4 | ±8.9 | ±62.4 | ±10.9 | ±90.3 | ±18.9 | ±62.3 | ±11.0 | ±65.6 | ±11.5 |
| Stair-1d | 19881.2 | 221.2 | 18450.3 | 218.5 | 18449.4 | 233.5 | 18449.8 | 218.5 | 18805.1 | 227.6 |
| | ±88.4 | ±8.9 | ±83.0 | ±9.3 | ±125.1 | ±14.9 | ±83.0 | ±9.3 | ±84.5 | ±9.8 |
| Step16-1d | 19881.2 | 221.2 | 10057.4 | 234.0 | 10128.1 | 240.9 | 10026.5 | 235.0 | 10069.1 | 207.1 |
| | ±88.4 | ±8.9 | ±50.2 | ±11.7 | ±70.4 | ±17.2 | ±49.9 | ±11.8 | ±52.2 | ±10.2 |
| Step50-1d | 19881.2 | 221.2 | 13929.6 | 232.0 | 13848.4 | 240.1 | 13929.3 | 232.3 | 16780.0 | 305.4 |
| | ±88.4 | ±8.9 | ±76.7 | ±10.0 | ±109.4 | ±15.1 | ±76.8 | ±10.1 | ±83.1 | ±11.4 |

TABLE 35. Squared Errors (with standard deviations). Id Query. 1-d datasets. Gauss Mechanism ($\rho = 0.005$).

| Dataset | olsalg | nnlsalg | maxalg | seqalg | weightalg |
|---|---|---|---|---|---|
| Level00-1d | 198.0 | 606.0 | 663.1 | 201.2 | 150.7 |
| | ±8.4 | ±18.0 | ±35.5 | ±8.6 | ±6.8 |
| Level01-1d | 198.0 | 390.2 | 400.7 | 201.2 | 199.1 |
| | ±8.4 | ±14.2 | ±20.3 | ±8.6 | ±8.5 |
| Level16-1d | 198.0 | 198.3 | 205.4 | 201.2 | 199.1 |
| | ±8.4 | ±8.4 | ±11.9 | ±8.6 | ±8.5 |
| Level32-1d | 198.0 | 198.0 | 192.3 | 201.2 | 199.0 |
| | ±8.4 | ±8.4 | ±16.1 | ±8.6 | ±8.5 |
| SplitStairs-1d | 198.0 | 220.4 | 217.9 | 201.2 | 199.1 |
| | ±8.4 | ±9.2 | ±13.1 | ±8.6 | ±8.5 |
| Stair-1d | 198.0 | 198.4 | 206.5 | 201.2 | 198.6 |
| | ±8.4 | ±8.4 | ±12.9 | ±8.6 | ±8.4 |
| Step16-1d | 198.0 | 221.2 | 221.5 | 201.2 | 199.1 |
| | ±8.4 | ±9.2 | ±13.8 | ±8.6 | ±8.5 |
| Step50-1d | 198.0 | 211.9 | 220.9 | 201.2 | 198.9 |
| | ±8.4 | ±8.8 | ±13.0 | ±8.6 | ±8.5 |

TABLE 36. Squared Error (with standard deviations). Sum Query. 1-d datasets. Gauss Mechanism ($\rho = 0.005$).

| Dataset | olsalg Total | olsalg Max | nnlsalg Total | nnlsalg Max | maxalg Total | maxalg Max | seqalg Total | seqalg Max | weightalg Total | weightalg Max |
|---|---|---|---|---|---|---|---|---|---|---|
| Level00-2d | 33.3 | 3.7 | 8.3 | 2.3 | 8.8 | 2.5 | 9.6 | 6.0 | 3.2 | 1.6 |
|  | ±0.5 | ±0.2 | ±0.2 | ±0.1 | ±0.3 | ±0.2 | ±0.3 | ±0.3 | ±0.1 | ±0.1 |
| Level01-2d | 33.3 | 3.7 | 30.0 | 3.3 | 29.8 | 3.3 | 36.9 | 4.0 | 36.5 | 4.0 |
|  | ±0.5 | ±0.2 | ±0.4 | ±0.2 | ±0.8 | ±0.3 | ±0.5 | ±0.2 | ±0.5 | ±0.2 |
| Level16-2d | 33.3 | 3.7 | 33.3 | 3.7 | 33.4 | 3.8 | 37.0 | 4.0 | 33.3 | 3.7 |
|  | ±0.5 | ±0.2 | ±0.5 | ±0.2 | ±0.6 | ±0.2 | ±0.5 | ±0.2 | ±0.5 | ±0.2 |
| Level32-2d | 33.3 | 3.7 | 33.3 | 3.7 | 33.7 | 3.8 | 37.0 | 4.0 | 33.3 | 3.7 |
|  | ±0.5 | ±0.2 | ±0.5 | ±0.2 | ±0.6 | ±0.2 | ±0.5 | ±0.2 | ±0.5 | ±0.2 |
| SplitStairs-2d | 33.3 | 3.7 | 30.8 | 3.4 | 30.3 | 3.4 | 37.0 | 4.0 | 32.4 | 3.6 |
|  | ±0.5 | ±0.2 | ±0.4 | ±0.2 | ±0.7 | ±0.3 | ±0.5 | ±0.2 | ±0.5 | ±0.2 |
| Stair-2d | 33.3 | 3.7 | 33.3 | 3.7 | 35.2 | 3.6 | 37.0 | 4.0 | 33.5 | 3.7 |
|  | ±0.5 | ±0.2 | ±0.5 | ±0.2 | ±0.8 | ±0.2 | ±0.5 | ±0.2 | ±0.5 | ±0.2 |
| Step16-2d | 33.3 | 3.7 | 31.0 | 3.4 | 30.3 | 3.8 | 36.7 | 4.0 | 30.9 | 3.4 |
|  | ±0.5 | ±0.2 | ±0.4 | ±0.2 | ±0.8 | ±0.3 | ±0.5 | ±0.2 | ±0.4 | ±0.2 |
| Step50-2d | 33.3 | 3.7 | 31.0 | 3.4 | 30.7 | 3.5 | 36.8 | 4.0 | 31.0 | 3.4 |
|  | ±0.5 | ±0.2 | ±0.4 | ±0.2 | ±0.9 | ±0.3 | ±0.5 | ±0.2 | ±0.4 | ±0.2 |

TABLE 37. Squared Errors (with standard deviations). Marg1 Query. 2-d datasets. Gauss Mechanism ($\rho = 0.5$).

| Dataset | olsalg Total | olsalg Max | nnlsalg Total | nnlsalg Max | maxalg Total | maxalg Max | seqalg Total | seqalg Max | weightalg Total | weightalg Max |
|---|---|---|---|---|---|---|---|---|---|---|
| Level00-2d | 32.8 | 3.6 | 8.2 | 2.3 | 8.7 | 2.5 | 8.7 | 5.5 | 3.1 | 1.5 |
|  | ±0.5 | ±0.2 | ±0.2 | ±0.1 | ±0.3 | ±0.2 | ±0.3 | ±0.3 | ±0.1 | ±0.1 |
| Level01-2d | 32.8 | 3.6 | 29.7 | 3.2 | 30.5 | 3.3 | 35.9 | 3.8 | 35.6 | 3.8 |
|  | ±0.5 | ±0.2 | ±0.4 | ±0.1 | ±0.8 | ±0.3 | ±0.5 | ±0.2 | ±0.5 | ±0.2 |
| Level16-2d | 32.8 | 3.6 | 32.8 | 3.6 | 33.4 | 3.6 | 35.9 | 3.8 | 32.8 | 3.6 |
|  | ±0.5 | ±0.2 | ±0.5 | ±0.2 | ±0.6 | ±0.2 | ±0.5 | ±0.2 | ±0.5 | ±0.2 |
| Level32-2d | 32.8 | 3.6 | 32.8 | 3.6 | 33.7 | 3.9 | 35.9 | 3.8 | 32.8 | 3.6 |
|  | ±0.5 | ±0.2 | ±0.5 | ±0.2 | ±0.6 | ±0.2 | ±0.5 | ±0.2 | ±0.5 | ±0.2 |
| SplitStairs-2d | 32.8 | 3.6 | 36.3 | 4.6 | 37.3 | 4.7 | 24.6 | 3.9 | 21.8 | 3.6 |
|  | ±0.5 | ±0.2 | ±0.5 | ±0.2 | ±0.9 | ±0.4 | ±0.4 | ±0.2 | ±0.4 | ±0.2 |
| Stair-2d | 32.8 | 3.6 | 32.8 | 3.6 | 33.7 | 3.8 | 35.9 | 3.8 | 32.8 | 3.6 |
|  | ±0.5 | ±0.2 | ±0.5 | ±0.2 | ±0.7 | ±0.3 | ±0.5 | ±0.2 | ±0.5 | ±0.2 |
| Step16-2d | 32.8 | 3.6 | 35.8 | 4.4 | 34.7 | 4.8 | 27.2 | 3.9 | 25.5 | 3.7 |
|  | ±0.5 | ±0.2 | ±0.5 | ±0.2 | ±0.9 | ±0.4 | ±0.4 | ±0.2 | ±0.4 | ±0.2 |
| Step50-2d | 32.8 | 3.6 | 35.8 | 4.4 | 37.0 | 4.7 | 27.2 | 3.8 | 25.5 | 3.7 |
|  | ±0.5 | ±0.2 | ±0.5 | ±0.2 | ±1.1 | ±0.5 | ±0.4 | ±0.2 | ±0.4 | ±0.2 |

TABLE 38. Squared Errors (with standard deviations). Marg2 Query. 2-d datasets. Gauss Mechanism ($\rho = 0.5$).

| Dataset | olsalg | | nnlsalg | | maxalg | | seqalg | | weightalg | |
|---|---|---|---|---|---|---|---|---|---|---|
| | Total | Max | Total | Max | Total | Max | Total | Max | Total | Max |
| Level00-2d | 330.1 | 3.6 | 9.7 | 4.6 | 10.9 | 5.4 | 12.4 | 9.1 | 4.3 | 2.3 |
| | ±1.5 | ±0.2 | ±0.2 | ±0.2 | ±0.4 | ±0.3 | ±0.3 | ±0.3 | ±0.1 | ±0.1 |
| Level01-2d | 330.1 | 3.6 | 139.0 | 3.1 | 139.9 | 3.0 | 137.0 | 3.0 | 139.1 | 4.1 |
| | ±1.5 | ±0.2 | ±0.7 | ±0.1 | ±1.4 | ±0.3 | ±0.7 | ±0.1 | ±0.8 | ±0.2 |
| Level16-2d | 330.1 | 3.6 | 330.1 | 3.6 | 334.5 | 4.0 | 330.8 | 3.6 | 330.1 | 3.6 |
| | ±1.5 | ±0.2 | ±1.5 | ±0.2 | ±2.0 | ±0.2 | ±1.5 | ±0.2 | ±1.5 | ±0.2 |
| Level32-2d | 330.1 | 3.6 | 330.1 | 3.6 | 333.1 | 3.7 | 330.8 | 3.6 | 330.1 | 3.6 |
| | ±1.5 | ±0.2 | ±1.5 | ±0.2 | ±1.9 | ±0.2 | ±1.5 | ±0.2 | ±1.5 | ±0.2 |
| SplitStairs-2d | 330.1 | 3.6 | 158.9 | 3.3 | 159.7 | 3.6 | 153.5 | 3.3 | 201.3 | 9.7 |
| | ±1.5 | ±0.2 | ±1.0 | ±0.1 | ±1.7 | ±0.3 | ±1.0 | ±0.1 | ±1.4 | ±0.3 |
| Stair-2d | 330.1 | 3.6 | 328.8 | 3.6 | 333.3 | 3.9 | 329.4 | 3.6 | 344.7 | 6.5 |
| | ±1.5 | ±0.2 | ±1.5 | ±0.2 | ±2.3 | ±0.3 | ±1.5 | ±0.2 | ±1.5 | ±0.3 |
| Step16-2d | 330.1 | 3.6 | 168.6 | 4.8 | 170.3 | 5.1 | 165.8 | 6.6 | 164.0 | 3.6 |
| | ±1.5 | ±0.2 | ±1.0 | ±0.2 | ±2.0 | ±0.4 | ±1.0 | ±0.2 | ±1.0 | ±0.2 |
| Step50-2d | 330.1 | 3.6 | 168.6 | 4.8 | 170.9 | 5.1 | 165.6 | 6.6 | 164.0 | 3.6 |
| | ±1.5 | ±0.2 | ±1.0 | ±0.2 | ±2.0 | ±0.4 | ±1.0 | ±0.2 | ±1.0 | ±0.2 |

TABLE 39. Squared Errors (with standard deviations). Id Query. 2-d datasets. Gauss Mechanism ($\rho = 0.5$).

| Dataset | olsalg | nnlsalg | maxalg | seqalg | weightalg |
|---|---|---|---|---|---|
| Level00-2d | 3.1 | 13.5 | 13.6 | 3.7 | 2.9 |
| | ±0.1 | ±0.3 | ±0.5 | ±0.2 | ±0.1 |
| Level01-2d | 3.1 | 3.3 | 3.3 | 4.0 | 3.1 |
| | ±0.1 | ±0.1 | ±0.3 | ±0.2 | ±0.1 |
| Level16-2d | 3.1 | 3.1 | 3.0 | 4.0 | 3.1 |
| | ±0.1 | ±0.1 | ±0.2 | ±0.2 | ±0.1 |
| Level32-2d | 3.1 | 3.1 | 3.0 | 4.0 | 3.1 |
| | ±0.1 | ±0.1 | ±0.2 | ±0.2 | ±0.1 |
| SplitStairs-2d | 3.1 | 4.1 | 3.9 | 4.0 | 3.1 |
| | ±0.1 | ±0.2 | ±0.3 | ±0.2 | ±0.1 |
| Stair-2d | 3.1 | 3.1 | 3.1 | 4.0 | 3.1 |
| | ±0.1 | ±0.1 | ±0.2 | ±0.2 | ±0.1 |
| Step16-2d | 3.1 | 3.9 | 4.1 | 4.0 | 3.3 |
| | ±0.1 | ±0.2 | ±0.3 | ±0.2 | ±0.1 |
| Step50-2d | 3.1 | 3.9 | 4.1 | 4.0 | 3.3 |
| | ±0.1 | ±0.2 | ±0.4 | ±0.2 | ±0.1 |

TABLE 40. Squared Error (with standard deviations). Sum Query. 2-d datasets. Gauss Mechanism ($\rho = 0.5$).

| Dataset | olsalg Total | olsalg Max | nnlsalg Total | nnlsalg Max | maxalg Total | maxalg Max | seqalg Total | seqalg Max | weightalg Total | weightalg Max |
|---|---|---|---|---|---|---|---|---|---|---|
| Level00-2d | 133.3 | 14.9 | 33.2 | 9.1 | 34.5 | 9.4 | 31.4 | 21.1 | 12.3 | 6.5 |
|  | ±1.9 | ±0.7 | ±0.7 | ±0.4 | ±2.6 | ±1.5 | ±2.0 | ±1.8 | ±0.5 | ±0.3 |
| Level01-2d | 133.3 | 14.9 | 104.5 | 11.4 | 104.9 | 12.0 | 146.7 | 15.5 | 144.3 | 15.7 |
|  | ±1.9 | ±0.7 | ±1.5 | ±0.5 | ±3.0 | ±1.0 | ±2.2 | ±0.7 | ±2.0 | ±0.6 |
| Level16-2d | 133.3 | 14.9 | 133.3 | 14.9 | 137.1 | 14.9 | 147.9 | 16.1 | 143.2 | 15.6 |
|  | ±1.9 | ±0.7 | ±1.9 | ±0.7 | ±3.3 | ±1.2 | ±2.1 | ±0.7 | ±2.0 | ±0.7 |
| Level32-2d | 133.3 | 14.9 | 133.3 | 14.9 | 133.5 | 15.1 | 147.9 | 16.1 | 133.3 | 14.9 |
|  | ±1.9 | ±0.7 | ±1.9 | ±0.7 | ±3.2 | ±1.1 | ±2.1 | ±0.7 | ±1.9 | ±0.7 |
| SplitStairs-2d | 133.3 | 14.9 | 122.6 | 13.6 | 123.5 | 13.7 | 148.0 | 16.1 | 137.0 | 14.8 |
|  | ±1.9 | ±0.7 | ±1.7 | ±0.6 | ±3.0 | ±1.0 | ±2.1 | ±0.7 | ±1.9 | ±0.6 |
| Stair-2d | 133.3 | 14.9 | 133.2 | 14.8 | 132.8 | 16.1 | 147.7 | 16.1 | 137.5 | 15.1 |
|  | ±1.9 | ±0.7 | ±1.9 | ±0.7 | ±4.5 | ±1.6 | ±2.1 | ±0.7 | ±1.9 | ±0.7 |
| Step16-2d | 133.3 | 14.9 | 123.8 | 13.7 | 124.2 | 15.7 | 148.1 | 16.2 | 142.9 | 15.5 |
|  | ±1.9 | ±0.7 | ±1.8 | ±0.6 | ±4.2 | ±1.7 | ±2.1 | ±0.7 | ±2.0 | ±0.7 |
| Step50-2d | 133.3 | 14.9 | 123.8 | 13.7 | 120.1 | 13.9 | 147.4 | 16.0 | 123.8 | 13.6 |
|  | ±1.9 | ±0.7 | ±1.8 | ±0.6 | ±4.3 | ±1.6 | ±2.1 | ±0.7 | ±1.8 | ±0.6 |

TABLE 41. Squared Errors (with standard deviations). Marg1 Query. 2-d datasets. Gauss Mechanism ($\rho = 0.125$).

| Dataset | olsalg Total | olsalg Max | nnlsalg Total | nnlsalg Max | maxalg Total | maxalg Max | seqalg Total | seqalg Max | weightalg Total | weightalg Max |
|---|---|---|---|---|---|---|---|---|---|---|
| Level00-2d | 131.4 | 14.3 | 32.9 | 9.1 | 35.9 | 8.6 | 33.2 | 22.1 | 11.8 | 6.2 |
|  | ±1.8 | ±0.6 | ±0.7 | ±0.4 | ±2.5 | ±1.2 | ±1.9 | ±1.7 | ±0.5 | ±0.3 |
| Level01-2d | 131.4 | 14.3 | 103.5 | 11.2 | 107.4 | 11.8 | 144.2 | 15.5 | 140.8 | 14.9 |
|  | ±1.8 | ±0.6 | ±1.4 | ±0.5 | ±3.0 | ±1.0 | ±2.1 | ±0.7 | ±1.9 | ±0.6 |
| Level16-2d | 131.4 | 14.3 | 131.4 | 14.3 | 131.1 | 15.0 | 143.7 | 15.3 | 140.4 | 15.1 |
|  | ±1.8 | ±0.6 | ±1.8 | ±0.6 | ±3.1 | ±1.1 | ±2.0 | ±0.7 | ±2.0 | ±0.7 |
| Level32-2d | 131.4 | 14.3 | 131.4 | 14.3 | 129.5 | 14.3 | 143.7 | 15.3 | 131.4 | 14.3 |
|  | ±1.8 | ±0.6 | ±1.8 | ±0.6 | ±3.2 | ±1.1 | ±2.0 | ±0.7 | ±1.8 | ±0.6 |
| SplitStairs-2d | 131.4 | 14.3 | 144.3 | 18.5 | 151.0 | 19.2 | 98.6 | 15.5 | 79.5 | 13.5 |
|  | ±1.8 | ±0.6 | ±2.0 | ±0.8 | ±3.6 | ±1.5 | ±1.7 | ±0.7 | ±1.5 | ±0.6 |
| Stair-2d | 131.4 | 14.3 | 131.2 | 14.3 | 136.7 | 16.1 | 143.5 | 15.3 | 132.4 | 14.3 |
|  | ±1.8 | ±0.6 | ±1.8 | ±0.6 | ±4.8 | ±1.7 | ±2.0 | ±0.7 | ±1.9 | ±0.6 |
| Step16-2d | 131.4 | 14.3 | 143.1 | 17.7 | 146.9 | 19.0 | 108.6 | 15.5 | 89.5 | 14.8 |
|  | ±1.8 | ±0.6 | ±2.0 | ±0.8 | ±4.8 | ±2.0 | ±1.8 | ±0.7 | ±1.6 | ±0.7 |
| Step50-2d | 131.4 | 14.3 | 143.1 | 17.7 | 146.5 | 19.1 | 109.0 | 15.5 | 101.9 | 15.0 |
|  | ±1.8 | ±0.6 | ±2.0 | ±0.8 | ±5.2 | ±1.9 | ±1.8 | ±0.7 | ±1.7 | ±0.7 |

TABLE 42. Squared Errors (with standard deviations). Marg2 Query. 2-d datasets. Gauss Mechanism ($\rho = 0.125$).

| Dataset | olsalg | | nnlsalg | | maxalg | | seqalg | | weightalg | |
|---|---|---|---|---|---|---|---|---|---|---|
| | Total | Max | Total | Max | Total | Max | Total | Max | Total | Max |
| Level00-2d | 1320.4 | 14.3 | 38.9 | 18.5 | 43.6 | 19.6 | 43.3 | 32.7 | 16.6 | 8.9 |
| | ±5.9 | ±0.6 | ±0.8 | ±0.6 | ±2.8 | ±2.2 | ±2.2 | ±2.1 | ±0.5 | ±0.4 |
| Level01-2d | 1320.4 | 14.3 | 281.3 | 12.6 | 286.1 | 13.4 | 270.3 | 11.8 | 277.1 | 15.3 |
| | ±5.9 | ±0.6 | ±2.0 | ±0.6 | ±4.3 | ±1.2 | ±2.1 | ±0.6 | ±2.2 | ±0.7 |
| Level16-2d | 1320.4 | 14.3 | 1320.3 | 14.3 | 1331.2 | 16.0 | 1323.0 | 14.4 | 2483.6 | 29.6 |
| | ±5.9 | ±0.6 | ±5.9 | ±0.6 | ±10.2 | ±1.3 | ±5.9 | ±0.6 | ±15.7 | ±1.8 |
| Level32-2d | 1320.4 | 14.3 | 1320.4 | 14.3 | 1343.8 | 16.0 | 1323.0 | 14.4 | 1320.4 | 14.3 |
| | ±5.9 | ±0.6 | ±5.9 | ±0.6 | ±10.2 | ±1.2 | ±5.9 | ±0.6 | ±5.9 | ±0.6 |
| SplitStairs-2d | 1320.4 | 14.3 | 620.8 | 13.3 | 618.9 | 14.9 | 600.4 | 13.3 | 981.1 | 46.0 |
| | ±5.9 | ±0.6 | ±3.8 | ±0.6 | ±6.6 | ±1.0 | ±3.8 | ±0.6 | ±7.2 | ±1.8 |
| Stair-2d | 1320.4 | 14.3 | 1300.6 | 14.4 | 1305.3 | 17.7 | 1302.2 | 14.4 | 1713.8 | 52.9 |
| | ±5.9 | ±0.6 | ±5.8 | ±0.6 | ±14.3 | ±1.9 | ±5.9 | ±0.6 | ±9.3 | ±2.3 |
| Step16-2d | 1320.4 | 14.3 | 674.5 | 19.0 | 673.4 | 19.5 | 662.2 | 26.4 | 1269.5 | 27.7 |
| | ±5.9 | ±0.6 | ±4.0 | ±0.7 | ±9.5 | ±1.7 | ±4.1 | ±0.9 | ±10.9 | ±1.7 |
| Step50-2d | 1320.4 | 14.3 | 674.5 | 19.0 | 663.2 | 19.0 | 663.4 | 26.5 | 657.9 | 14.5 |
| | ±5.9 | ±0.6 | ±4.0 | ±0.7 | ±10.3 | ±1.8 | ±4.1 | ±0.9 | ±4.0 | ±0.7 |

TABLE 43. Squared Errors (with standard deviations). Id Query. 2-d datasets. Gauss Mechanism ($\rho = 0.125$).

| Dataset | olsalg | nnlsalg | maxalg | seqalg | weightalg |
|---|---|---|---|---|---|
| Level00-2d | 12.4 | 53.8 | 60.2 | 14.8 | 11.0 |
| | ±0.5 | ±1.3 | ±5.4 | ±1.2 | ±0.5 |
| Level01-2d | 12.4 | 15.1 | 14.6 | 16.1 | 13.0 |
| | ±0.5 | ±0.6 | ±1.4 | ±0.7 | ±0.5 |
| Level16-2d | 12.4 | 12.4 | 13.6 | 16.1 | 12.4 |
| | ±0.5 | ±0.5 | ±1.0 | ±0.7 | ±0.5 |
| Level32-2d | 12.4 | 12.4 | 13.5 | 16.1 | 12.4 |
| | ±0.5 | ±0.5 | ±1.0 | ±0.7 | ±0.5 |
| SplitStairs-2d | 12.4 | 16.4 | 17.1 | 16.2 | 12.1 |
| | ±0.5 | ±0.7 | ±1.2 | ±0.7 | ±0.5 |
| Stair-2d | 12.4 | 12.4 | 14.7 | 16.1 | 12.4 |
| | ±0.5 | ±0.5 | ±1.4 | ±0.7 | ±0.5 |
| Step16-2d | 12.4 | 15.7 | 15.5 | 16.0 | 12.3 |
| | ±0.5 | ±0.7 | ±1.6 | ±0.7 | ±0.5 |
| Step50-2d | 12.4 | 15.7 | 18.9 | 16.0 | 13.0 |
| | ±0.5 | ±0.7 | ±2.1 | ±0.7 | ±0.5 |

TABLE 44. Squared Error (with standard deviations). Sum Query. 2-d datasets. Gauss Mechanism ($\rho = 0.125$).

| Dataset | olsalg | | nnlsalg | | maxalg | | seqalg | | weightalg | |
|---|---|---|---|---|---|---|---|---|---|---|
| | Total | Max | Total | Max | Total | Max | Total | Max | Total | Max |
| Level00-2d | 3333.2 | 371.8 | 829.4 | 227.4 | 483.9 | 105.7 | 898.3 | 460.5 | 304.9 | 163.2 |
| | ±47.1 | ±16.8 | ±17.9 | ±10.2 | ±79.2 | ±39.9 | ±39.4 | ±32.3 | ±11.8 | ±7.8 |
| Level01-2d | 3333.2 | 371.8 | 1363.7 | 219.6 | 1339.7 | 230.6 | 1805.8 | 360.8 | 1433.1 | 275.5 |
| | ±47.1 | ±16.8 | ±23.3 | ±10.4 | ±80.1 | ±37.0 | ±35.7 | ±19.8 | ±30.6 | ±13.4 |
| Level16-2d | 3333.2 | 371.8 | 3179.0 | 353.9 | 3153.5 | 354.3 | 3696.4 | 403.3 | 3686.8 | 403.1 |
| | ±47.1 | ±16.8 | ±45.0 | ±16.2 | ±63.8 | ±23.7 | ±52.4 | ±17.1 | ±52.2 | ±17.1 |
| Level32-2d | 3333.2 | 371.8 | 3311.1 | 369.9 | 3311.5 | 361.1 | 3698.6 | 404.0 | 3687.2 | 403.2 |
| | ±47.1 | ±16.8 | ±46.8 | ±16.8 | ±74.9 | ±24.4 | ±52.4 | ±17.2 | ±52.2 | ±17.1 |
| SplitStairs-2d | 3333.2 | 371.8 | 2927.4 | 322.8 | 2868.5 | 324.6 | 3693.3 | 407.5 | 3666.8 | 401.4 |
| | ±47.1 | ±16.8 | ±41.1 | ±14.7 | ±79.1 | ±29.9 | ±54.3 | ±17.9 | ±51.8 | ±17.0 |
| Stair-2d | 3333.2 | 371.8 | 3291.8 | 364.2 | 3291.3 | 367.4 | 3696.7 | 403.7 | 3681.4 | 402.8 |
| | ±47.1 | ±16.8 | ±46.6 | ±16.5 | ±76.5 | ±26.6 | ±52.4 | ±17.1 | ±52.0 | ±17.0 |
| Step16-2d | 3333.2 | 371.8 | 2854.0 | 317.0 | 3037.9 | 402.3 | 3692.5 | 394.1 | 3599.4 | 394.6 |
| | ±47.1 | ±16.8 | ±40.8 | ±14.5 | ±96.2 | ±40.8 | ±54.2 | ±17.1 | ±51.6 | ±17.4 |
| Step50-2d | 3333.2 | 371.8 | 3094.0 | 342.3 | 3209.5 | 382.7 | 3687.0 | 397.7 | 3677.2 | 402.1 |
| | ±47.1 | ±16.8 | ±44.0 | ±15.7 | ±106.5 | ±40.2 | ±52.3 | ±16.7 | ±52.1 | ±17.0 |

TABLE 45. Squared Errors (with standard deviations). Marg1 Query. 2-d datasets. Gauss Mechanism ($\rho = 0.005$).

| Dataset | olsalg | | nnlsalg | | maxalg | | seqalg | | weightalg | |
|---|---|---|---|---|---|---|---|---|---|---|
| | Total | Max | Total | Max | Total | Max | Total | Max | Total | Max |
| Level00-2d | 3283.8 | 356.4 | 821.8 | 228.5 | 650.6 | 147.9 | 898.6 | 468.2 | 291.4 | 154.5 |
| | ±46.1 | ±15.9 | ±16.8 | ±9.7 | ±98.1 | ±41.6 | ±36.9 | ±30.1 | ±11.3 | ±7.3 |
| Level01-2d | 3283.8 | 356.4 | 1345.2 | 219.2 | 1296.5 | 206.0 | 1781.8 | 359.2 | 1398.4 | 264.8 |
| | ±46.1 | ±15.9 | ±21.9 | ±9.5 | ±76.4 | ±34.6 | ±33.6 | ±17.7 | ±28.9 | ±12.5 |
| Level16-2d | 3283.8 | 356.4 | 3129.3 | 342.2 | 3094.7 | 354.0 | 3592.6 | 383.0 | 3583.0 | 382.4 |
| | ±46.1 | ±15.9 | ±43.9 | ±15.2 | ±60.9 | ±23.0 | ±50.0 | ±16.7 | ±49.9 | ±16.7 |
| Level32-2d | 3283.8 | 356.4 | 3258.6 | 354.4 | 3365.3 | 381.5 | 3589.9 | 382.4 | 3583.4 | 382.5 |
| | ±46.1 | ±15.9 | ±45.7 | ±15.8 | ±76.7 | ±26.2 | ±50.1 | ±16.7 | ±49.9 | ±16.7 |
| SplitStairs-2d | 3283.8 | 356.4 | 3378.9 | 472.8 | 3460.6 | 521.1 | 2496.5 | 393.4 | 2006.2 | 356.3 |
| | ±46.1 | ±15.9 | ±48.3 | ±19.6 | ±101.2 | ±44.1 | ±44.6 | ±17.7 | ±38.2 | ±15.5 |
| Stair-2d | 3283.8 | 356.4 | 3263.7 | 356.8 | 3281.3 | 369.4 | 3594.4 | 383.2 | 3554.7 | 383.2 |
| | ±46.1 | ±15.9 | ±45.9 | ±16.1 | ±77.9 | ±28.9 | ±50.1 | ±16.7 | ±49.4 | ±16.7 |
| Step16-2d | 3283.8 | 356.4 | 3259.0 | 405.0 | 3346.3 | 419.0 | 2739.2 | 397.4 | 2238.9 | 372.4 |
| | ±46.1 | ±15.9 | ±47.1 | ±18.3 | ±106.5 | ±39.5 | ±46.5 | ±18.3 | ±39.2 | ±16.7 |
| Step50-2d | 3283.8 | 356.4 | 3574.7 | 442.9 | 3734.3 | 459.1 | 2716.8 | 385.5 | 2240.0 | 372.5 |
| | ±46.1 | ±15.9 | ±51.1 | ±20.2 | ±121.2 | ±45.7 | ±44.5 | ±17.3 | ±39.3 | ±16.7 |

TABLE 46. Squared Errors (with standard deviations). Marg2 Query. 2-d datasets. Gauss Mechanism ($\rho = 0.005$).

| Dataset | olsalg | | nnlsalg | | maxalg | | seqalg | | weightalg | |
|---|---|---|---|---|---|---|---|---|---|---|
| | Total | Max | Total | Max | Total | Max | Total | Max | Total | Max |
| Level00-2d | 33009.3 | 358.7 | 973.2 | 462.3 | 628.1 | 230.7 | 1171.9 | 771.7 | 411.4 | 221.1 |
| | ±146.9 | ±15.4 | ±18.9 | ±15.7 | ±80.9 | ±51.1 | ±41.0 | ±37.7 | ±13.7 | ±10.6 |
| Level01-2d | 33009.3 | 358.7 | 1815.7 | 367.7 | 1981.2 | 462.2 | 1716.6 | 427.0 | 1493.4 | 290.4 |
| | ±146.9 | ±15.4 | ±24.3 | ±14.5 | ±101.3 | ±66.5 | ±28.2 | ±18.3 | ±27.7 | ±14.3 |
| Level16-2d | 33009.3 | 358.7 | 20740.6 | 319.2 | 20784.7 | 316.8 | 20633.6 | 316.0 | 21079.8 | 419.2 |
| | ±146.9 | ±15.4 | ±90.7 | ±13.9 | ±128.9 | ±19.3 | ±90.1 | ±13.8 | ±100.5 | ±18.3 |
| Level32-2d | 33009.3 | 358.7 | 30426.8 | 333.5 | 30995.1 | 362.3 | 30442.5 | 331.9 | 32278.1 | 419.3 |
| | ±146.9 | ±15.4 | ±124.4 | ±14.6 | ±205.0 | ±22.2 | ±124.6 | ±14.6 | ±142.7 | ±18.3 |
| SplitStairs-2d | 33009.3 | 358.7 | 11784.0 | 334.9 | 11908.5 | 325.0 | 11304.8 | 334.2 | 12703.8 | 403.3 |
| | ±146.9 | ±15.4 | ±72.0 | ±14.6 | ±146.7 | ±31.0 | ±73.8 | ±15.1 | ±87.9 | ±17.5 |
| Stair-2d | 33009.3 | 358.7 | 28790.5 | 354.2 | 28682.7 | 369.2 | 28727.4 | 358.2 | 37500.2 | 605.3 |
| | ±146.9 | ±15.4 | ±129.8 | ±15.1 | ±220.7 | ±26.8 | ±129.7 | ±15.6 | ±204.4 | ±38.3 |
| Step16-2d | 33009.3 | 358.7 | 10758.2 | 463.8 | 10818.2 | 458.5 | 10647.5 | 636.2 | 10298.1 | 305.9 |
| | ±146.9 | ±15.4 | ±60.9 | ±18.1 | ±133.8 | ±40.1 | ±65.0 | ±22.5 | ±67.0 | ±15.0 |
| Step50-2d | 33009.3 | 358.7 | 16789.7 | 475.1 | 17644.5 | 489.9 | 16504.9 | 658.7 | 20888.5 | 444.8 |
| | ±146.9 | ±15.4 | ±98.9 | ±18.6 | ±239.6 | ±44.4 | ±99.3 | ±22.3 | ±123.7 | ±18.6 |

TABLE 47. Squared Errors (with standard deviations). Id Query. 2-d datasets. Gauss Mechanism ($\rho = 0.005$).

| Dataset | olsalg | nnlsalg | maxalg | seqalg | weightalg |
|---|---|---|---|---|---|
| Level00-2d | 310.7 | 1343.7 | 1084.0 | 450.9 | 268.9 |
| | ±13.1 | ±32.6 | ±171.2 | ±26.4 | ±12.1 |
| Level01-2d | 310.7 | 717.3 | 836.6 | 392.6 | 326.6 |
| | ±13.1 | ±24.7 | ±111.0 | ±18.8 | ±13.7 |
| Level16-2d | 310.7 | 312.5 | 304.3 | 402.3 | 312.0 |
| | ±13.1 | ±13.1 | ±19.0 | ±17.2 | ±13.2 |
| Level32-2d | 310.7 | 310.3 | 311.6 | 403.1 | 312.0 |
| | ±13.1 | ±13.1 | ±23.2 | ±17.2 | ±13.2 |
| SplitStairs-2d | 310.7 | 436.9 | 465.0 | 405.5 | 299.4 |
| | ±13.1 | ±17.7 | ±39.7 | ±17.9 | ±12.4 |
| Stair-2d | 310.7 | 311.0 | 307.5 | 402.6 | 311.8 |
| | ±13.1 | ±13.1 | ±22.0 | ±17.2 | ±13.2 |
| Step16-2d | 310.7 | 419.7 | 465.2 | 393.4 | 300.4 |
| | ±13.1 | ±17.2 | ±44.7 | ±17.3 | ±12.7 |
| Step50-2d | 310.7 | 393.1 | 414.4 | 398.5 | 300.7 |
| | ±13.1 | ±16.3 | ±40.1 | ±16.8 | ±12.7 |

TABLE 48. Squared Error (with standard deviations). Sum Query. 2-d datasets. Gauss Mechanism ($\rho = 0.005$).

| Dataset | olsalg | | nnlsalg | | maxalg | | seqalg | | weightalg | |
|---|---|---|---|---|---|---|---|---|---|---|
| | Total | Max | Total | Max | Total | Max | Total | Max | Total | Max |
| PUMA0101301 | 31.7 | 3.8 | 18.2 | 3.1 | 18.2 | 3.1 | 20.7 | 3.8 | 14.7 | 3.3 |
| | ±0.5 | ±0.2 | ±0.3 | ±0.1 | ±0.3 | ±0.1 | ±0.5 | ±0.2 | ±0.3 | ±0.2 |
| PUMA0800803 | 31.7 | 3.8 | 25.6 | 5.8 | 25.5 | 5.7 | 25.6 | 3.6 | 22.5 | 3.8 |
| | ±0.5 | ±0.2 | ±0.4 | ±0.3 | ±0.4 | ±0.3 | ±0.6 | ±0.2 | ±0.4 | ±0.2 |
| PUMA1304600 | 31.7 | 3.8 | 26.3 | 3.7 | 26.2 | 3.6 | 25.1 | 3.8 | 24.1 | 3.9 |
| | ±0.5 | ±0.2 | ±0.4 | ±0.2 | ±0.4 | ±0.2 | ±0.7 | ±0.3 | ±0.4 | ±0.1 |
| PUMA1703529 | 31.7 | 3.8 | 25.6 | 4.7 | 25.6 | 4.6 | 24.7 | 3.8 | 22.8 | 3.6 |
| | ±0.5 | ±0.2 | ±0.4 | ±0.2 | ±0.4 | ±0.2 | ±0.5 | ±0.2 | ±0.4 | ±0.2 |
| PUMA1703531 | 31.7 | 3.8 | 18.0 | 2.4 | 18.0 | 2.4 | 23.0 | 3.7 | 18.9 | 4.5 |
| | ±0.5 | ±0.2 | ±0.3 | ±0.1 | ±0.3 | ±0.1 | ±0.5 | ±0.2 | ±0.3 | ±0.2 |
| PUMA1901700 | 31.7 | 3.8 | 27.4 | 4.6 | 27.4 | 4.5 | 24.0 | 3.8 | 20.9 | 3.6 |
| | ±0.5 | ±0.2 | ±0.4 | ±0.2 | ±0.5 | ±0.2 | ±0.6 | ±0.2 | ±0.4 | ±0.2 |
| PUMA2401004 | 31.7 | 3.8 | 29.4 | 6.9 | 29.2 | 6.8 | 24.2 | 3.9 | 21.5 | 3.6 |
| | ±0.5 | ±0.2 | ±0.5 | ±0.3 | ±0.5 | ±0.3 | ±0.7 | ±0.3 | ±0.4 | ±0.2 |
| PUMA2602702 | 31.7 | 3.8 | 25.5 | 5.2 | 25.3 | 5.1 | 24.2 | 3.8 | 20.4 | 3.7 |
| | ±0.5 | ±0.2 | ±0.4 | ±0.2 | ±0.4 | ±0.2 | ±0.6 | ±0.2 | ±0.4 | ±0.2 |
| PUMA2801100 | 31.7 | 3.8 | 20.3 | 3.9 | 20.3 | 3.9 | 22.3 | 3.7 | 18.8 | 4.0 |
| | ±0.5 | ±0.2 | ±0.3 | ±0.2 | ±0.4 | ±0.2 | ±0.5 | ±0.2 | ±0.3 | ±0.2 |
| PUMA2901901 | 31.7 | 3.8 | 23.3 | 4.6 | 23.2 | 4.6 | 25.2 | 3.7 | 22.8 | 4.3 |
| | ±0.5 | ±0.2 | ±0.4 | ±0.2 | ±0.4 | ±0.2 | ±0.6 | ±0.2 | ±0.4 | ±0.2 |
| PUMA3200405 | 31.7 | 3.8 | 29.6 | 4.1 | 29.5 | 4.1 | 28.9 | 3.7 | 26.5 | 3.9 |
| | ±0.5 | ±0.2 | ±0.5 | ±0.2 | ±0.5 | ±0.2 | ±0.6 | ±0.2 | ±0.5 | ±0.2 |
| PUMA3603710 | 31.7 | 3.8 | 31.3 | 4.7 | 31.4 | 4.8 | 29.6 | 4.0 | 27.6 | 4.0 |
| | ±0.5 | ±0.2 | ±0.5 | ±0.2 | ±0.5 | ±0.2 | ±0.7 | ±0.3 | ±0.5 | ±0.2 |
| PUMA3604010 | 31.7 | 3.8 | 26.6 | 4.0 | 26.9 | 4.1 | 23.0 | 3.7 | 20.0 | 3.4 |
| | ±0.5 | ±0.2 | ±0.4 | ±0.2 | ±0.5 | ±0.2 | ±0.6 | ±0.2 | ±0.4 | ±0.2 |
| PUMA5101301 | 31.7 | 3.8 | 28.2 | 6.0 | 27.8 | 5.9 | 25.8 | 3.8 | 23.2 | 3.6 |
| | ±0.5 | ±0.2 | ±0.4 | ±0.3 | ±0.5 | ±0.3 | ±0.6 | ±0.2 | ±0.4 | ±0.2 |
| PUMA5151255 | 31.7 | 3.8 | 31.6 | 5.4 | 31.3 | 5.4 | 25.5 | 4.0 | 23.0 | 3.6 |
| | ±0.5 | ±0.2 | ±0.5 | ±0.2 | ±0.5 | ±0.3 | ±0.6 | ±0.2 | ±0.4 | ±0.2 |

TABLE 49. Squared Errors (with standard deviations). Marg1 Query. PUMS datasets. Gauss Mechanism ($\rho = 0.5$).

| Dataset | olsalg | | nnlsalg | | maxalg | | seqalg | | weightalg | |
|---|---|---|---|---|---|---|---|---|---|---|
| | Total | Max | Total | Max | Total | Max | Total | Max | Total | Max |
| PUMA0101301 | 83.3 | 3.7 | 26.2 | 7.5 | 26.3 | 7.6 | 27.2 | 5.5 | 16.5 | 3.1 |
| | ±0.8 | ±0.2 | ±0.4 | ±0.3 | ±0.4 | ±0.3 | ±0.5 | ±0.3 | ±0.3 | ±0.1 |
| PUMA0800803 | 83.3 | 3.7 | 44.3 | 7.6 | 43.8 | 7.5 | 51.8 | 4.3 | 44.4 | 5.9 |
| | ±0.8 | ±0.2 | ±0.5 | ±0.3 | ±0.5 | ±0.3 | ±0.8 | ±0.2 | ±0.5 | ±0.2 |
| PUMA1304600 | 83.3 | 3.7 | 46.8 | 6.2 | 46.5 | 6.2 | 52.4 | 4.6 | 47.2 | 5.2 |
| | ±0.8 | ±0.2 | ±0.5 | ±0.2 | ±0.5 | ±0.3 | ±1.0 | ±0.3 | ±0.5 | ±0.2 |
| PUMA1703529 | 83.3 | 3.7 | 44.5 | 6.5 | 44.2 | 6.4 | 46.8 | 4.1 | 41.2 | 5.7 |
| | ±0.8 | ±0.2 | ±0.5 | ±0.2 | ±0.5 | ±0.3 | ±0.7 | ±0.2 | ±0.5 | ±0.2 |
| PUMA1703531 | 83.3 | 3.7 | 26.6 | 9.8 | 26.6 | 9.7 | 21.2 | 5.7 | 16.6 | 4.2 |
| | ±0.8 | ±0.2 | ±0.4 | ±0.3 | ±0.4 | ±0.3 | ±0.4 | ±0.3 | ±0.3 | ±0.1 |
| PUMA1901700 | 83.3 | 3.7 | 48.7 | 5.8 | 48.8 | 5.8 | 51.1 | 4.2 | 43.7 | 5.6 |
| | ±0.8 | ±0.2 | ±0.5 | ±0.2 | ±0.6 | ±0.2 | ±0.8 | ±0.2 | ±0.5 | ±0.2 |
| PUMA2401004 | 83.3 | 3.7 | 55.1 | 5.7 | 54.7 | 5.6 | 74.9 | 4.2 | 62.4 | 5.2 |
| | ±0.8 | ±0.2 | ±0.5 | ±0.2 | ±0.6 | ±0.3 | ±1.2 | ±0.3 | ±0.6 | ±0.2 |
| PUMA2602702 | 83.3 | 3.7 | 42.2 | 6.7 | 42.2 | 6.7 | 49.9 | 4.5 | 40.9 | 3.7 |
| | ±0.8 | ±0.2 | ±0.5 | ±0.2 | ±0.5 | ±0.3 | ±0.7 | ±0.3 | ±0.5 | ±0.2 |
| PUMA2801100 | 83.3 | 3.7 | 30.6 | 7.9 | 30.7 | 8.0 | 31.3 | 5.2 | 25.3 | 5.3 |
| | ±0.8 | ±0.2 | ±0.4 | ±0.3 | ±0.4 | ±0.3 | ±0.5 | ±0.3 | ±0.4 | ±0.2 |
| PUMA2901901 | 83.3 | 3.7 | 37.7 | 7.6 | 37.5 | 7.5 | 37.2 | 4.7 | 27.6 | 4.0 |
| | ±0.8 | ±0.2 | ±0.5 | ±0.3 | ±0.5 | ±0.3 | ±0.7 | ±0.3 | ±0.4 | ±0.2 |
| PUMA3200405 | 83.3 | 3.7 | 57.3 | 5.8 | 57.0 | 5.8 | 61.6 | 4.1 | 56.7 | 5.4 |
| | ±0.8 | ±0.2 | ±0.6 | ±0.2 | ±0.6 | ±0.3 | ±0.9 | ±0.3 | ±0.6 | ±0.2 |
| PUMA3603710 | 83.3 | 3.7 | 63.4 | 5.0 | 63.1 | 5.0 | 70.1 | 4.4 | 65.1 | 4.8 |
| | ±0.8 | ±0.2 | ±0.6 | ±0.2 | ±0.7 | ±0.2 | ±1.0 | ±0.3 | ±0.7 | ±0.2 |
| PUMA3604010 | 83.3 | 3.7 | 47.3 | 5.0 | 46.9 | 4.9 | 53.4 | 4.3 | 41.2 | 4.7 |
| | ±0.8 | ±0.2 | ±0.5 | ±0.2 | ±0.6 | ±0.2 | ±0.8 | ±0.2 | ±0.5 | ±0.2 |
| PUMA5101301 | 83.3 | 3.7 | 50.8 | 6.1 | 50.6 | 6.2 | 68.5 | 4.2 | 62.5 | 5.0 |
| | ±0.8 | ±0.2 | ±0.5 | ±0.2 | ±0.6 | ±0.3 | ±1.0 | ±0.3 | ±0.6 | ±0.2 |
| PUMA5151255 | 83.3 | 3.7 | 59.6 | 4.8 | 59.5 | 4.9 | 73.8 | 4.3 | 69.2 | 4.6 |
| | ±0.8 | ±0.2 | ±0.6 | ±0.2 | ±0.6 | ±0.2 | ±0.9 | ±0.2 | ±0.7 | ±0.2 |

TABLE 50. Squared Errors (with standard deviations). Marg2 Query. PUMS datasets. Gauss Mechanism ($\rho = 0.5$).

| Dataset | olsalg | | nnlsalg | | maxalg | | seqalg | | weightalg | |
|---|---|---|---|---|---|---|---|---|---|---|
| | Total | Max | Total | Max | Total | Max | Total | Max | Total | Max |
| PUMA0101301 | 746.5 | 3.8 | 33.9 | 4.5 | 34.2 | 4.6 | 37.2 | 6.6 | 33.2 | 5.0 |
| | ±2.3 | ±0.2 | ±0.3 | ±0.2 | ±0.4 | ±0.2 | ±0.5 | ±0.3 | ±0.5 | ±0.3 |
| PUMA0800803 | 746.5 | 3.8 | 68.5 | 4.3 | 68.8 | 4.4 | 76.0 | 9.1 | 74.9 | 6.7 |
| | ±2.3 | ±0.2 | ±0.5 | ±0.2 | ±0.6 | ±0.2 | ±0.8 | ±0.4 | ±0.7 | ±0.3 |
| PUMA1304600 | 746.5 | 3.8 | 88.0 | 4.4 | 87.9 | 4.4 | 87.7 | 5.5 | 106.0 | 8.1 |
| | ±2.3 | ±0.2 | ±0.6 | ±0.2 | ±0.7 | ±0.2 | ±1.0 | ±0.3 | ±0.9 | ±0.4 |
| PUMA1703529 | 746.5 | 3.8 | 73.3 | 3.7 | 73.1 | 3.7 | 71.5 | 5.3 | 77.9 | 5.3 |
| | ±2.3 | ±0.2 | ±0.5 | ±0.2 | ±0.6 | ±0.2 | ±0.7 | ±0.3 | ±0.7 | ±0.2 |
| PUMA1703531 | 746.5 | 3.8 | 31.6 | 4.4 | 31.9 | 4.4 | 31.5 | 5.8 | 30.2 | 5.3 |
| | ±2.3 | ±0.2 | ±0.3 | ±0.1 | ±0.3 | ±0.1 | ±0.4 | ±0.1 | ±0.4 | ±0.1 |
| PUMA1901700 | 746.5 | 3.8 | 91.3 | 3.9 | 91.8 | 4.0 | 88.8 | 5.0 | 104.5 | 9.8 |
| | ±2.3 | ±0.2 | ±0.6 | ±0.2 | ±0.7 | ±0.2 | ±0.9 | ±0.3 | ±0.9 | ±0.5 |
| PUMA2401004 | 746.5 | 3.8 | 103.2 | 4.7 | 103.1 | 4.7 | 114.6 | 9.0 | 114.5 | 6.3 |
| | ±2.3 | ±0.2 | ±0.7 | ±0.2 | ±0.8 | ±0.2 | ±1.3 | ±0.5 | ±0.8 | ±0.3 |
| PUMA2602702 | 746.5 | 3.8 | 65.1 | 3.6 | 65.1 | 3.7 | 67.3 | 7.0 | 70.3 | 8.8 |
| | ±2.3 | ±0.2 | ±0.5 | ±0.2 | ±0.5 | ±0.2 | ±0.7 | ±0.3 | ±0.7 | ±0.4 |
| PUMA2801100 | 746.5 | 3.8 | 40.7 | 3.9 | 40.9 | 4.0 | 42.8 | 5.8 | 40.1 | 4.7 |
| | ±2.3 | ±0.2 | ±0.4 | ±0.2 | ±0.4 | ±0.2 | ±0.5 | ±0.3 | ±0.5 | ±0.2 |
| PUMA2901901 | 746.5 | 3.8 | 54.3 | 3.4 | 54.2 | 3.4 | 56.5 | 6.2 | 57.0 | 6.5 |
| | ±2.3 | ±0.2 | ±0.4 | ±0.1 | ±0.5 | ±0.1 | ±0.7 | ±0.3 | ±0.6 | ±0.3 |
| PUMA3200405 | 746.5 | 3.8 | 131.0 | 4.3 | 131.4 | 4.2 | 130.1 | 5.2 | 155.2 | 7.6 |
| | ±2.3 | ±0.2 | ±0.8 | ±0.2 | ±0.9 | ±0.2 | ±1.2 | ±0.2 | ±1.1 | ±0.5 |
| PUMA3603710 | 746.5 | 3.8 | 158.4 | 4.4 | 158.3 | 4.4 | 157.1 | 5.6 | 191.7 | 9.2 |
| | ±2.3 | ±0.2 | ±0.9 | ±0.2 | ±1.0 | ±0.2 | ±1.3 | ±0.3 | ±1.2 | ±0.4 |
| PUMA3604010 | 746.5 | 3.8 | 84.5 | 4.4 | 84.3 | 4.4 | 84.5 | 5.9 | 85.2 | 5.6 |
| | ±2.3 | ±0.2 | ±0.6 | ±0.2 | ±0.7 | ±0.2 | ±0.8 | ±0.3 | ±0.8 | ±0.2 |
| PUMA5101301 | 746.5 | 3.8 | 95.7 | 4.3 | 95.2 | 4.2 | 104.7 | 7.3 | 109.1 | 6.6 |
| | ±2.3 | ±0.2 | ±0.6 | ±0.2 | ±0.7 | ±0.2 | ±1.0 | ±0.4 | ±0.8 | ±0.2 |
| PUMA5151255 | 746.5 | 3.8 | 136.7 | 4.3 | 136.7 | 4.5 | 136.7 | 5.5 | 158.1 | 10.2 |
| | ±2.3 | ±0.2 | ±0.8 | ±0.2 | ±0.9 | ±0.2 | ±1.1 | ±0.3 | ±1.0 | ±0.5 |

TABLE 51. Squared Errors (with standard deviations). Id Query. PUMS datasets. Gauss Mechanism ($\rho = 0.5$).

| Dataset | olsalg | nnlsalg | maxalg | seqalg | weightalg |
|---|---|---|---|---|---|
| PUMA0101301 | 3.2 | 13.7 | 13.8 | 4.0 | 3.1 |
| | ±0.1 | ±0.4 | ±0.4 | ±0.2 | ±0.1 |
| PUMA0800803 | 3.2 | 9.9 | 10.1 | 4.1 | 3.2 |
| | ±0.1 | ±0.3 | ±0.3 | ±0.2 | ±0.1 |
| PUMA1304600 | 3.2 | 8.8 | 8.9 | 4.3 | 3.3 |
| | ±0.1 | ±0.3 | ±0.3 | ±0.3 | ±0.1 |
| PUMA1703529 | 3.2 | 9.2 | 9.5 | 4.1 | 3.2 |
| | ±0.1 | ±0.3 | ±0.3 | ±0.2 | ±0.1 |
| PUMA1703531 | 3.2 | 13.6 | 13.8 | 3.9 | 3.2 |
| | ±0.1 | ±0.4 | ±0.4 | ±0.2 | ±0.1 |
| PUMA1901700 | 3.2 | 8.9 | 8.9 | 4.0 | 3.2 |
| | ±0.1 | ±0.3 | ±0.3 | ±0.2 | ±0.1 |
| PUMA2401004 | 3.2 | 8.2 | 8.1 | 3.9 | 3.3 |
| | ±0.1 | ±0.3 | ±0.3 | ±0.3 | ±0.1 |
| PUMA2602702 | 3.2 | 9.7 | 9.8 | 4.1 | 3.2 |
| | ±0.1 | ±0.3 | ±0.3 | ±0.2 | ±0.1 |
| PUMA2801100 | 3.2 | 12.4 | 12.6 | 3.9 | 3.2 |
| | ±0.1 | ±0.3 | ±0.4 | ±0.2 | ±0.1 |
| PUMA2901901 | 3.2 | 10.7 | 10.8 | 4.0 | 3.2 |
| | ±0.1 | ±0.3 | ±0.3 | ±0.2 | ±0.1 |
| PUMA3200405 | 3.2 | 7.4 | 7.2 | 4.0 | 3.3 |
| | ±0.1 | ±0.3 | ±0.3 | ±0.3 | ±0.1 |
| PUMA3603710 | 3.2 | 6.7 | 6.7 | 3.9 | 3.3 |
| | ±0.1 | ±0.2 | ±0.3 | ±0.2 | ±0.1 |
| PUMA3604010 | 3.2 | 9.1 | 9.3 | 3.9 | 3.2 |
| | ±0.1 | ±0.3 | ±0.3 | ±0.2 | ±0.1 |
| PUMA5101301 | 3.2 | 8.5 | 8.6 | 4.2 | 3.2 |
| | ±0.1 | ±0.3 | ±0.3 | ±0.3 | ±0.1 |
| PUMA5151255 | 3.2 | 7.1 | 7.2 | 4.2 | 3.2 |
| | ±0.1 | ±0.3 | ±0.3 | ±0.2 | ±0.1 |

TABLE 52. Squared Error (with standard deviations). Sum Query. PUMS datasets. Gauss Mechanism ($\rho = 0.5$).

| Dataset | olsalg | | nnlsalg | | maxalg | | seqalg | | weightalg | |
|---------|--------|-----|---------|-----|--------|-----|--------|-----|-----------|-----|
| | Total | Max | Total | Max | Total | Max | Total | Max | Total | Max |
| PUMA0101301 | 127.0 | 15.2 | 67.2 | 10.7 | 67.3 | 10.9 | 80.3 | 15.0 | 62.1 | 19.4 |
| | ±1.9 | ±0.6 | ±1.1 | ±0.4 | ±1.1 | ±0.5 | ±2.0 | ±1.0 | ±1.2 | ±0.6 |
| PUMA0800803 | 127.0 | 15.2 | 92.9 | 22.0 | 91.8 | 21.3 | 94.7 | 14.9 | 89.6 | 19.5 |
| | ±1.9 | ±0.6 | ±1.5 | ±0.9 | ±1.6 | ±1.0 | ±2.3 | ±0.9 | ±1.5 | ±0.7 |
| PUMA1304600 | 127.0 | 15.2 | 97.4 | 13.9 | 97.7 | 13.9 | 97.0 | 15.1 | 84.9 | 14.2 |
| | ±1.9 | ±0.6 | ±1.5 | ±0.7 | ±1.6 | ±0.7 | ±2.3 | ±0.9 | ±1.6 | ±0.6 |
| PUMA1703529 | 127.0 | 15.2 | 92.3 | 16.3 | 91.5 | 15.4 | 97.5 | 15.4 | 77.8 | 14.2 |
| | ±1.9 | ±0.6 | ±1.5 | ±0.8 | ±1.5 | ±0.7 | ±2.3 | ±0.9 | ±1.5 | ±0.7 |
| PUMA1703531 | 127.0 | 15.2 | 63.3 | 8.6 | 63.1 | 8.7 | 83.2 | 15.2 | 75.8 | 17.8 |
| | ±1.9 | ±0.6 | ±1.0 | ±0.4 | ±1.1 | ±0.4 | ±1.9 | ±0.9 | ±1.3 | ±0.7 |
| PUMA1901700 | 127.0 | 15.2 | 102.1 | 17.1 | 101.4 | 16.3 | 91.4 | 15.1 | 78.0 | 14.4 |
| | ±1.9 | ±0.6 | ±1.6 | ±0.8 | ±1.7 | ±0.8 | ±2.4 | ±1.1 | ±1.5 | ±0.7 |
| PUMA2401004 | 127.0 | 15.2 | 111.2 | 28.6 | 111.2 | 28.9 | 97.1 | 16.6 | 79.8 | 16.4 |
| | ±1.9 | ±0.6 | ±1.8 | ±1.1 | ±1.9 | ±1.3 | ±3.1 | ±1.5 | ±1.6 | ±0.8 |
| PUMA2602702 | 127.0 | 15.2 | 89.4 | 18.2 | 88.6 | 17.8 | 95.0 | 14.9 | 81.5 | 22.4 |
| | ±1.9 | ±0.6 | ±1.4 | ±0.8 | ±1.5 | ±0.8 | ±2.1 | ±0.8 | ±1.5 | ±0.8 |
| PUMA2801100 | 127.0 | 15.2 | 72.7 | 13.6 | 73.1 | 13.7 | 81.8 | 15.1 | 74.8 | 18.1 |
| | ±1.9 | ±0.6 | ±1.2 | ±0.6 | ±1.3 | ±0.7 | ±1.8 | ±0.9 | ±1.4 | ±0.7 |
| PUMA2901901 | 127.0 | 15.2 | 82.6 | 16.3 | 82.5 | 16.1 | 91.2 | 15.1 | 71.0 | 14.4 |
| | ±1.9 | ±0.6 | ±1.4 | ±0.7 | ±1.4 | ±0.8 | ±2.1 | ±0.9 | ±1.3 | ±0.7 |
| PUMA3200405 | 127.0 | 15.2 | 112.7 | 15.8 | 112.5 | 15.8 | 112.0 | 15.6 | 115.9 | 19.3 |
| | ±1.9 | ±0.6 | ±1.7 | ±0.8 | ±1.8 | ±0.8 | ±2.5 | ±1.1 | ±1.8 | ±0.7 |
| PUMA3603710 | 127.0 | 15.2 | 119.5 | 18.1 | 120.1 | 18.3 | 114.7 | 15.2 | 112.0 | 16.9 |
| | ±1.9 | ±0.6 | ±1.8 | ±0.7 | ±1.9 | ±0.8 | ±2.6 | ±1.0 | ±1.8 | ±0.6 |
| PUMA3604010 | 127.0 | 15.2 | 99.4 | 15.6 | 100.5 | 16.0 | 97.7 | 15.7 | 71.7 | 14.1 |
| | ±1.9 | ±0.6 | ±1.5 | ±0.6 | ±1.6 | ±0.7 | ±3.1 | ±1.2 | ±1.4 | ±0.6 |
| PUMA5101301 | 127.0 | 15.2 | 104.3 | 23.4 | 103.7 | 23.2 | 95.6 | 14.8 | 77.9 | 14.3 |
| | ±1.9 | ±0.6 | ±1.7 | ±1.0 | ±1.7 | ±1.0 | ±2.3 | ±0.9 | ±1.4 | ±0.7 |
| PUMA5151255 | 127.0 | 15.2 | 118.5 | 21.7 | 118.6 | 21.9 | 98.2 | 15.6 | 80.9 | 14.4 |
| | ±1.9 | ±0.6 | ±1.8 | ±1.0 | ±2.0 | ±1.1 | ±2.5 | ±1.1 | ±1.5 | ±0.7 |

TABLE 53. Squared Errors (with standard deviations). Marg1 Query. PUMS datasets. Gauss Mechanism ($\rho = 0.125$).

| Dataset | olsalg | | nnlsalg | | maxalg | | seqalg | | weightalg | |
|---|---|---|---|---|---|---|---|---|---|---|
| | Total | Max | Total | Max | Total | Max | Total | Max | Total | Max |
| PUMA0101301 | 333.4 | 15.0 | 92.7 | 30.4 | 93.2 | 30.6 | 88.7 | 24.1 | 46.7 | 12.9 |
| | ±3.0 | ±0.7 | ±1.4 | ±1.0 | ±1.5 | ±1.0 | ±2.2 | ±1.3 | ±1.1 | ±0.7 |
| PUMA0800803 | 333.4 | 15.0 | 150.1 | 30.5 | 149.4 | 30.4 | 167.3 | 17.9 | 136.1 | 18.9 |
| | ±3.0 | ±0.7 | ±1.8 | ±1.0 | ±1.8 | ±1.1 | ±2.8 | ±1.1 | ±1.9 | ±0.7 |
| PUMA1304600 | 333.4 | 15.0 | 162.7 | 25.6 | 162.4 | 25.2 | 166.2 | 18.4 | 135.8 | 16.2 |
| | ±3.0 | ±0.7 | ±1.9 | ±0.9 | ±2.0 | ±1.0 | ±2.8 | ±1.1 | ±1.9 | ±0.8 |
| PUMA1703529 | 333.4 | 15.0 | 146.4 | 27.9 | 146.4 | 27.7 | 143.0 | 18.8 | 114.3 | 16.5 |
| | ±3.0 | ±0.7 | ±1.8 | ±1.0 | ±1.9 | ±1.0 | ±2.5 | ±1.1 | ±1.7 | ±0.7 |
| PUMA1703531 | 333.4 | 15.0 | 88.6 | 38.5 | 89.0 | 38.9 | 64.2 | 27.6 | 39.5 | 13.1 |
| | ±3.0 | ±0.7 | ±1.4 | ±1.2 | ±1.5 | ±1.2 | ±1.6 | ±1.3 | ±0.9 | ±0.6 |
| PUMA1901700 | 333.4 | 15.0 | 172.9 | 23.5 | 172.9 | 23.0 | 181.4 | 19.0 | 171.9 | 23.6 |
| | ±3.0 | ±0.7 | ±2.0 | ±0.9 | ±2.1 | ±0.9 | ±3.3 | ±1.1 | ±2.1 | ±0.8 |
| PUMA2401004 | 333.4 | 15.0 | 200.6 | 23.7 | 197.8 | 23.2 | 274.7 | 17.8 | 235.5 | 19.8 |
| | ±3.0 | ±0.7 | ±2.0 | ±0.9 | ±2.2 | ±1.0 | ±5.0 | ±1.4 | ±2.5 | ±0.7 |
| PUMA2602702 | 333.4 | 15.0 | 131.3 | 28.8 | 131.4 | 29.1 | 138.9 | 17.8 | 113.1 | 19.3 |
| | ±3.0 | ±0.7 | ±1.6 | ±1.0 | ±1.7 | ±1.1 | ±2.3 | ±1.0 | ±1.6 | ±0.6 |
| PUMA2801100 | 333.4 | 15.0 | 103.3 | 32.8 | 103.9 | 33.2 | 94.0 | 22.4 | 72.0 | 17.4 |
| | ±3.0 | ±0.7 | ±1.5 | ±1.1 | ±1.5 | ±1.1 | ±1.8 | ±1.1 | ±1.3 | ±0.6 |
| PUMA2901901 | 333.4 | 15.0 | 126.3 | 30.7 | 126.4 | 30.4 | 118.4 | 19.4 | 98.8 | 25.2 |
| | ±3.0 | ±0.7 | ±1.7 | ±1.0 | ±1.8 | ±1.1 | ±2.3 | ±1.1 | ±1.7 | ±0.9 |
| PUMA3200405 | 333.4 | 15.0 | 208.9 | 24.4 | 207.9 | 24.3 | 217.0 | 17.2 | 202.7 | 23.2 |
| | ±3.0 | ±0.7 | ±2.2 | ±0.9 | ±2.4 | ±1.0 | ±3.4 | ±1.0 | ±2.4 | ±0.7 |
| PUMA3603710 | 333.4 | 15.0 | 230.1 | 19.6 | 227.7 | 19.5 | 244.3 | 17.1 | 210.8 | 19.9 |
| | ±3.0 | ±0.7 | ±2.3 | ±0.8 | ±2.4 | ±0.8 | ±3.8 | ±1.0 | ±2.4 | ±0.7 |
| PUMA3604010 | 333.4 | 15.0 | 168.2 | 22.2 | 167.9 | 21.8 | 187.5 | 17.2 | 180.1 | 24.1 |
| | ±3.0 | ±0.7 | ±1.9 | ±0.9 | ±2.0 | ±0.9 | ±4.0 | ±1.2 | ±2.2 | ±0.7 |
| PUMA5101301 | 333.4 | 15.0 | 174.6 | 26.0 | 175.1 | 26.0 | 222.8 | 17.0 | 186.1 | 18.0 |
| | ±3.0 | ±0.7 | ±1.8 | ±1.0 | ±2.0 | ±1.0 | ±3.3 | ±1.1 | ±2.2 | ±0.7 |
| PUMA5151255 | 333.4 | 15.0 | 210.4 | 20.1 | 210.0 | 19.4 | 256.9 | 15.9 | 222.7 | 15.8 |
| | ±3.0 | ±0.7 | ±2.1 | ±0.8 | ±2.2 | ±0.8 | ±3.8 | ±1.1 | ±2.4 | ±0.6 |

TABLE 54. Squared Errors (with standard deviations). Marg2 Query. PUMS datasets. Gauss Mechanism ($\rho = 0.125$).

| Dataset | olsalg | | nnlsalg | | maxalg | | seqalg | | weightalg | |
|---|---|---|---|---|---|---|---|---|---|---|
| | Total | Max | Total | Max | Total | Max | Total | Max | Total | Max |
| PUMA0101301 | 2985.8 | 15.2 | 113.2 | 19.9 | 113.6 | 20.0 | 125.5 | 30.6 | 117.4 | 23.0 |
| | ±9.0 | ±0.7 | ±1.3 | ±0.7 | ±1.4 | ±0.7 | ±2.2 | ±1.3 | ±2.3 | ±1.6 |
| PUMA0800803 | 2985.8 | 15.2 | 205.7 | 14.9 | 206.3 | 15.0 | 228.1 | 34.9 | 215.7 | 17.6 |
| | ±9.0 | ±0.7 | ±1.8 | ±0.6 | ±1.9 | ±0.7 | ±2.9 | ±1.6 | ±2.3 | ±0.7 |
| PUMA1304600 | 2985.8 | 15.2 | 267.2 | 14.9 | 266.7 | 14.9 | 257.3 | 19.8 | 287.8 | 28.2 |
| | ±9.0 | ±0.7 | ±2.0 | ±0.4 | ±2.1 | ±0.7 | ±2.7 | ±1.0 | ±2.9 | ±1.8 |
| PUMA1703529 | 2985.8 | 15.2 | 209.3 | 14.4 | 209.5 | 14.8 | 200.2 | 18.2 | 232.9 | 21.6 |
| | ±9.0 | ±0.7 | ±1.8 | ±0.6 | ±1.9 | ±0.7 | ±2.4 | ±0.8 | ±2.7 | ±1.3 |
| PUMA1703531 | 2985.8 | 15.2 | 91.7 | 11.8 | 92.1 | 11.9 | 85.9 | 15.3 | 90.3 | 17.9 |
| | ±9.0 | ±0.7 | ±1.0 | ±0.5 | ±1.1 | ±0.5 | ±1.6 | ±0.9 | ±1.4 | ±0.7 |
| PUMA1901700 | 2985.8 | 15.2 | 281.8 | 15.8 | 282.6 | 15.7 | 271.6 | 19.6 | 331.1 | 35.3 |
| | ±9.0 | ±0.7 | ±2.1 | ±0.7 | ±2.2 | ±0.7 | ±3.1 | ±1.1 | ±3.4 | ±2.2 |
| PUMA2401004 | 2985.8 | 15.2 | 331.5 | 18.6 | 330.8 | 18.5 | 370.7 | 38.5 | 355.6 | 26.7 |
| | ±9.0 | ±0.7 | ±2.4 | ±0.8 | ±2.6 | ±0.9 | ±4.9 | ±2.2 | ±2.9 | ±0.9 |
| PUMA2602702 | 2985.8 | 15.2 | 174.4 | 12.7 | 174.8 | 12.6 | 175.4 | 22.7 | 171.2 | 16.3 |
| | ±9.0 | ±0.7 | ±1.5 | ±0.6 | ±1.6 | ±0.6 | ±2.1 | ±1.1 | ±1.9 | ±0.9 |
| PUMA2801100 | 2985.8 | 15.2 | 123.4 | 15.7 | 124.1 | 15.8 | 124.4 | 20.8 | 128.2 | 19.0 |
| | ±9.0 | ±0.7 | ±1.3 | ±0.5 | ±1.3 | ±0.5 | ±1.7 | ±0.7 | ±1.7 | ±0.6 |
| PUMA2901901 | 2985.8 | 15.2 | 159.0 | 14.4 | 159.9 | 14.5 | 158.5 | 21.5 | 157.3 | 19.1 |
| | ±9.0 | ±0.7 | ±1.5 | ±0.5 | ±1.6 | ±0.6 | ±2.1 | ±0.9 | ±1.9 | ±0.7 |
| PUMA3200405 | 2985.8 | 15.2 | 418.3 | 15.2 | 419.7 | 15.2 | 412.5 | 19.1 | 498.0 | 44.6 |
| | ±9.0 | ±0.7 | ±2.7 | ±0.6 | ±2.9 | ±0.7 | ±3.8 | ±1.0 | ±4.1 | ±2.4 |
| PUMA3603710 | 2985.8 | 15.2 | 497.3 | 17.5 | 496.9 | 17.6 | 486.4 | 24.8 | 591.7 | 41.3 |
| | ±9.0 | ±0.7 | ±3.0 | ±0.7 | ±3.1 | ±0.8 | ±4.3 | ±1.3 | ±4.7 | ±2.4 |
| PUMA3604010 | 2985.8 | 15.2 | 268.2 | 16.3 | 268.5 | 16.4 | 269.2 | 21.7 | 286.9 | 22.8 |
| | ±9.0 | ±0.7 | ±2.1 | ±0.7 | ±2.2 | ±0.7 | ±3.9 | ±1.4 | ±3.0 | ±0.8 |
| PUMA5101301 | 2985.8 | 15.2 | 283.7 | 15.9 | 285.3 | 16.3 | 303.7 | 32.4 | 314.7 | 41.2 |
| | ±9.0 | ±0.7 | ±2.0 | ±0.7 | ±2.2 | ±0.7 | ±3.2 | ±1.6 | ±2.8 | ±1.6 |
| PUMA5151255 | 2985.8 | 15.2 | 407.9 | 16.9 | 409.8 | 16.9 | 406.0 | 23.7 | 456.2 | 38.9 |
| | ±9.0 | ±0.7 | ±2.6 | ±0.7 | ±2.8 | ±0.8 | ±3.9 | ±1.3 | ±3.7 | ±2.2 |

TABLE 55. Squared Errors (with standard deviations). Id Query. PUMS datasets. Gauss Mechanism ($\rho = 0.125$).

| Dataset | olsalg | nnlsalg | maxalg | seqalg | weightalg |
|---|---|---|---|---|---|
| PUMA0101301 | 12.9 | 57.8 | 58.5 | 15.9 | 12.0 |
| | ±0.6 | ±1.5 | ±1.5 | ±0.9 | ±0.5 |
| PUMA0800803 | 12.9 | 43.5 | 44.1 | 15.5 | 12.6 |
| | ±0.6 | ±1.3 | ±1.4 | ±0.9 | ±0.5 |
| PUMA1304600 | 12.9 | 38.8 | 39.2 | 16.9 | 12.9 |
| | ±0.6 | ±1.2 | ±1.3 | ±1.0 | ±0.6 |
| PUMA1703529 | 12.9 | 41.7 | 42.5 | 16.1 | 12.7 |
| | ±0.6 | ±1.2 | ±1.4 | ±0.9 | ±0.5 |
| PUMA1703531 | 12.9 | 59.2 | 60.5 | 15.7 | 12.0 |
| | ±0.6 | ±1.5 | ±1.6 | ±0.9 | ±0.5 |
| PUMA1901700 | 12.9 | 38.8 | 39.8 | 16.4 | 12.9 |
| | ±0.6 | ±1.2 | ±1.3 | ±1.1 | ±0.5 |
| PUMA2401004 | 12.9 | 35.9 | 35.3 | 15.5 | 12.9 |
| | ±0.6 | ±1.1 | ±1.2 | ±1.1 | ±0.5 |
| PUMA2602702 | 12.9 | 44.0 | 44.7 | 15.6 | 12.9 |
| | ±0.6 | ±1.3 | ±1.4 | ±0.9 | ±0.5 |
| PUMA2801100 | 12.9 | 54.0 | 54.9 | 15.2 | 12.4 |
| | ±0.6 | ±1.4 | ±1.5 | ±0.8 | ±0.5 |
| PUMA2901901 | 12.9 | 47.5 | 48.3 | 15.3 | 12.6 |
| | ±0.6 | ±1.3 | ±1.4 | ±0.9 | ±0.5 |
| PUMA3200405 | 12.9 | 31.9 | 31.9 | 15.8 | 13.2 |
| | ±0.6 | ±1.1 | ±1.2 | ±0.9 | ±0.6 |
| PUMA3603710 | 12.9 | 29.2 | 29.1 | 15.9 | 13.1 |
| | ±0.6 | ±1.0 | ±1.1 | ±1.1 | ±0.6 |
| PUMA3604010 | 12.9 | 39.8 | 40.5 | 16.9 | 12.9 |
| | ±0.6 | ±1.2 | ±1.3 | ±1.2 | ±0.6 |
| PUMA5101301 | 12.9 | 37.5 | 37.9 | 16.6 | 12.8 |
| | ±0.6 | ±1.2 | ±1.3 | ±1.0 | ±0.5 |
| PUMA5151255 | 12.9 | 31.6 | 32.3 | 17.8 | 12.9 |
| | ±0.6 | ±1.1 | ±1.2 | ±1.1 | ±0.5 |

TABLE 56. Squared Error (with standard deviations). Sum Query. PUMS datasets. Gauss Mechanism ($\rho = 0.125$).

| Dataset | olsalg | | nnlsalg | | maxalg | | seqalg | | weightalg | |
|---|---|---|---|---|---|---|---|---|---|---|
| | Total | Max | Total | Max | Total | Max | Total | Max | Total | Max |
| PUMA0101301 | 3174.8 | 380.7 | 1344.9 | 199.2 | 1505.8 | 251.8 | 1562.0 | 442.9 | 1207.2 | 399.7 |
| | ±47.5 | ±16.2 | ±23.2 | ±8.9 | ±158.4 | ±47.2 | ±53.2 | ±34.9 | ±25.7 | ±16.2 |
| PUMA0800803 | 3174.8 | 380.7 | 1762.3 | 389.7 | 1680.5 | 324.8 | 1818.6 | 415.2 | 1696.3 | 451.1 |
| | ±47.5 | ±16.2 | ±30.3 | ±17.2 | ±149.1 | ±67.6 | ±90.6 | ±45.0 | ±34.1 | ±16.8 |
| PUMA1304600 | 3174.8 | 380.7 | 1775.7 | 247.9 | 2110.7 | 343.6 | 1929.3 | 347.3 | 2028.5 | 467.0 |
| | ±47.5 | ±16.2 | ±28.5 | ±11.4 | ±184.6 | ±86.6 | ±138.7 | ±67.3 | ±35.5 | ±17.5 |
| PUMA1703529 | 3174.8 | 380.7 | 1812.6 | 292.8 | 1834.8 | 263.1 | 2186.2 | 389.2 | 1846.4 | 436.3 |
| | ±47.5 | ±16.2 | ±29.4 | ±13.4 | ±195.6 | ±83.8 | ±160.6 | ±75.4 | ±33.8 | ±15.5 |
| PUMA1703531 | 3174.8 | 380.7 | 1110.1 | 188.2 | 1168.5 | 219.9 | 1011.6 | 343.5 | 713.8 | 208.6 |
| | ±47.5 | ±16.2 | ±20.8 | ±8.5 | ±142.7 | ±83.6 | ±42.3 | ±26.9 | ±18.1 | ±9.7 |
| PUMA1901700 | 3174.8 | 380.7 | 1979.9 | 311.2 | 1908.6 | 269.1 | 2335.2 | 383.6 | 2363.4 | 540.7 |
| | ±47.5 | ±16.2 | ±31.5 | ±14.4 | ±229.9 | ±74.5 | ±127.2 | ±43.8 | ±39.9 | ±18.1 |
| PUMA2401004 | 3174.8 | 380.7 | 2281.1 | 639.7 | 2368.5 | 690.8 | 1842.9 | 395.6 | 1622.9 | 382.8 |
| | ±47.5 | ±16.2 | ±38.2 | ±25.8 | ±168.3 | ±118.2 | ±180.6 | ±81.7 | ±33.6 | ±17.4 |
| PUMA2602702 | 3174.8 | 380.7 | 1629.5 | 261.4 | 1806.0 | 287.5 | 1860.2 | 370.3 | 1740.1 | 451.8 |
| | ±47.5 | ±16.2 | ±27.2 | ±11.9 | ±189.2 | ±106.0 | ±70.2 | ±37.4 | ±31.8 | ±17.0 |
| PUMA2801100 | 3174.8 | 380.7 | 1319.0 | 230.1 | 1161.6 | 169.5 | 1431.0 | 443.3 | 806.3 | 226.8 |
| | ±47.5 | ±16.2 | ±23.7 | ±10.3 | ±134.6 | ±44.2 | ±91.7 | ±61.5 | ±19.9 | ±10.2 |
| PUMA2901901 | 3174.8 | 380.7 | 1589.1 | 252.8 | 1381.1 | 271.1 | 1804.1 | 389.6 | 1630.8 | 419.8 |
| | ±47.5 | ±16.2 | ±26.7 | ±11.5 | ±115.4 | ±61.1 | ±85.5 | ±46.2 | ±31.2 | ±15.7 |
| PUMA3200405 | 3174.8 | 380.7 | 2237.7 | 355.0 | 2238.9 | 433.1 | 2449.2 | 386.6 | 2337.5 | 546.3 |
| | ±47.5 | ±16.2 | ±35.5 | ±16.6 | ±316.3 | ±138.5 | ±96.6 | ±39.1 | ±41.9 | ±20.9 |
| PUMA3603710 | 3174.8 | 380.7 | 2529.8 | 400.5 | 2924.0 | 491.2 | 2202.8 | 353.6 | 2217.6 | 469.4 |
| | ±47.5 | ±16.2 | ±40.0 | ±16.6 | ±254.4 | ±131.1 | ±59.8 | ±24.5 | ±41.3 | ±20.1 |
| PUMA3604010 | 3174.8 | 380.7 | 1830.6 | 261.3 | 1827.3 | 312.6 | 2013.5 | 400.3 | 2156.8 | 481.7 |
| | ±47.5 | ±16.2 | ±28.8 | ±11.6 | ±203.0 | ±97.0 | ±99.2 | ±49.2 | ±37.0 | ±16.5 |
| PUMA5101301 | 3174.8 | 380.7 | 1959.8 | 404.0 | 1697.0 | 303.9 | 2176.5 | 438.7 | 2157.6 | 470.0 |
| | ±47.5 | ±16.2 | ±32.3 | ±18.1 | ±170.9 | ±67.6 | ±106.1 | ±45.2 | ±38.1 | ±17.5 |
| PUMA5151255 | 3174.8 | 380.7 | 2361.5 | 429.3 | 2671.4 | 472.6 | 2358.7 | 427.8 | 2053.0 | 494.5 |
| | ±47.5 | ±16.2 | ±37.7 | ±19.5 | ±270.5 | ±135.9 | ±81.7 | ±35.3 | ±47.1 | ±26.5 |

TABLE 57. Squared Errors (with standard deviations). Marg1 Query. PUMS datasets. Gauss Mechanism ($\rho = 0.005$).

| Dataset | olsalg Total | olsalg Max | nnlsalg Total | nnlsalg Max | maxalg Total | maxalg Max | seqalg Total | seqalg Max | weightalg Total | weightalg Max |
|---|---|---|---|---|---|---|---|---|---|---|
| PUMA0101301 | 8334.3 | 374.7 | 1768.8 | 680.6 | 1876.2 | 828.8 | 1423.4 | 698.7 | 836.0 | 297.5 |
|  | ±75.7 | ±16.7 | ±28.9 | ±22.2 | ±165.2 | ±133.0 | ±46.2 | ±38.8 | ±20.8 | ±13.0 |
| PUMA0800803 | 8334.3 | 374.7 | 2421.6 | 763.4 | 2315.0 | 721.9 | 2582.6 | 687.0 | 1659.1 | 512.2 |
|  | ±75.7 | ±16.7 | ±35.7 | ±25.1 | ±214.7 | ±160.4 | ±112.5 | ±70.1 | ±32.5 | ±15.5 |
| PUMA1304600 | 8334.3 | 374.7 | 2479.2 | 761.8 | 3053.7 | 1120.9 | 2188.5 | 605.8 | 1685.0 | 338.0 |
|  | ±75.7 | ±16.7 | ±35.0 | ±25.3 | ±261.0 | ±204.7 | ±125.5 | ±75.1 | ±31.9 | ±14.7 |
| PUMA1703529 | 8334.3 | 374.7 | 2570.7 | 817.3 | 2759.4 | 787.0 | 2156.1 | 726.9 | 1712.5 | 691.7 |
|  | ±75.7 | ±16.7 | ±38.0 | ±26.6 | ±234.6 | ±135.2 | ±194.0 | ±150.5 | ±31.5 | ±17.8 |
| PUMA1703531 | 8334.3 | 374.7 | 1501.2 | 666.1 | 1703.8 | 763.7 | 1306.0 | 834.6 | 471.0 | 262.8 |
|  | ±75.7 | ±16.7 | ±26.4 | ±20.7 | ±196.4 | ±155.8 | ±53.1 | ±45.6 | ±16.1 | ±11.2 |
| PUMA1901700 | 8334.3 | 374.7 | 2816.2 | 726.2 | 2986.8 | 766.6 | 2980.6 | 679.5 | 1609.1 | 347.6 |
|  | ±75.7 | ±16.7 | ±39.1 | ±24.8 | ±296.6 | ±194.0 | ±165.7 | ±89.8 | ±30.5 | ±14.8 |
| PUMA2401004 | 8334.3 | 374.7 | 3427.8 | 666.8 | 3418.7 | 699.5 | 3659.6 | 474.0 | 3122.8 | 323.2 |
|  | ±75.7 | ±16.7 | ±39.9 | ±23.9 | ±189.2 | ±129.1 | ±185.0 | ±87.9 | ±45.2 | ±14.4 |
| PUMA2602702 | 8334.3 | 374.7 | 2097.6 | 876.0 | 2295.3 | 942.6 | 1511.3 | 676.2 | 936.0 | 320.3 |
|  | ±75.7 | ±16.7 | ±34.3 | ±27.4 | ±233.6 | ±177.8 | ±64.0 | ±50.7 | ±23.6 | ±14.7 |
| PUMA2801100 | 8334.3 | 374.7 | 1714.4 | 743.1 | 1583.8 | 649.3 | 1437.2 | 814.4 | 618.2 | 265.5 |
|  | ±75.7 | ±16.7 | ±29.6 | ±23.6 | ±174.8 | ±140.0 | ±89.1 | ±76.2 | ±18.0 | ±11.4 |
| PUMA2901901 | 8334.3 | 374.7 | 2105.3 | 878.9 | 2443.2 | 1240.6 | 1775.1 | 824.5 | 965.8 | 324.8 |
|  | ±75.7 | ±16.7 | ±34.0 | ±27.3 | ±186.2 | ±160.8 | ±112.8 | ±96.2 | ±23.3 | ±14.2 |
| PUMA3200405 | 8334.3 | 374.7 | 3527.2 | 678.6 | 3383.2 | 746.2 | 3425.1 | 462.4 | 2803.8 | 424.5 |
|  | ±75.7 | ±16.7 | ±43.3 | ±24.3 | ±371.5 | ±244.4 | ±96.2 | ±33.0 | ±41.9 | ±15.5 |
| PUMA3603710 | 8334.3 | 374.7 | 4395.0 | 529.9 | 4256.4 | 564.5 | 4535.3 | 443.2 | 3958.1 | 528.4 |
|  | ±75.7 | ±16.7 | ±49.6 | ±20.2 | ±307.6 | ±155.5 | ±83.6 | ±28.1 | ±52.3 | ±20.0 |
| PUMA3604010 | 8334.3 | 374.7 | 2504.2 | 687.9 | 2814.2 | 815.6 | 2576.0 | 603.8 | 1831.9 | 353.0 |
|  | ±75.7 | ±16.7 | ±33.8 | ±23.4 | ±232.6 | ±158.9 | ±104.9 | ±61.8 | ±33.1 | ±14.6 |
| PUMA5101301 | 8334.3 | 374.7 | 2589.2 | 789.0 | 2506.4 | 832.1 | 2385.2 | 570.9 | 1764.4 | 330.9 |
|  | ±75.7 | ±16.7 | ±36.4 | ±26.4 | ±233.4 | ±171.8 | ±104.4 | ±63.5 | ±34.2 | ±15.0 |
| PUMA5151255 | 8334.3 | 374.7 | 3529.9 | 619.3 | 3413.1 | 636.9 | 4129.0 | 454.4 | 3495.5 | 430.9 |
|  | ±75.7 | ±16.7 | ±41.2 | ±22.9 | ±296.6 | ±188.5 | ±94.3 | ±38.2 | ±47.4 | ±16.4 |

TABLE 58. Squared Errors (with standard deviations). Marg2 Query. PUMS datasets. Gauss Mechanism ($\rho = 0.005$).

| Dataset | olsalg | | nnlsalg | | maxalg | | seqalg | | weightalg | |
|---|---|---|---|---|---|---|---|---|---|---|
| | Total | Max | Total | Max | Total | Max | Total | Max | Total | Max |
| PUMA0101301 | 74645.2 | 380.9 | 1885.3 | 369.2 | 1922.9 | 451.9 | 1785.6 | 466.7 | 1485.5 | 420.3 |
| | ±225.9 | ±17.8 | ±24.5 | ±14.3 | ±156.6 | ±86.7 | ±47.0 | ±32.7 | ±27.9 | ±16.8 |
| PUMA0800803 | 74645.2 | 380.9 | 2763.8 | 338.4 | 2686.3 | 426.1 | 3098.6 | 516.6 | 2772.4 | 476.5 |
| | ±225.9 | ±17.8 | ±31.5 | ±14.7 | ±201.6 | ±124.7 | ±100.3 | ±46.4 | ±41.7 | ±18.0 |
| PUMA1304600 | 74645.2 | 380.9 | 2726.3 | 332.3 | 3035.0 | 416.5 | 2268.1 | 390.9 | 2698.8 | 488.4 |
| | ±225.9 | ±17.8 | ±29.1 | ±14.3 | ±199.3 | ±121.3 | ±107.1 | ±64.0 | ±38.4 | ±18.1 |
| PUMA1703529 | 74645.2 | 380.9 | 2898.4 | 320.5 | 3063.2 | 320.4 | 2738.5 | 514.2 | 2793.3 | 410.7 |
| | ±225.9 | ±17.8 | ±30.8 | ±14.3 | ±198.0 | ±85.1 | ±141.2 | ±87.3 | ±39.1 | ±14.3 |
| PUMA1703531 | 74645.2 | 380.9 | 1246.1 | 329.1 | 1423.9 | 398.7 | 1011.5 | 387.6 | 761.2 | 200.4 |
| | ±225.9 | ±17.8 | ±17.6 | ±12.2 | ±132.4 | ±100.5 | ±32.7 | ±25.7 | ±17.0 | ±9.1 |
| PUMA1901700 | 74645.2 | 380.9 | 3452.1 | 306.7 | 3623.8 | 361.0 | 3162.2 | 340.7 | 3456.1 | 439.4 |
| | ±225.9 | ±17.8 | ±34.4 | ±11.0 | ±248.2 | ±93.0 | ±112.7 | ±39.5 | ±45.7 | ±14.2 |
| PUMA2401004 | 74645.2 | 380.9 | 4529.2 | 359.5 | 4575.7 | 365.6 | 4774.0 | 820.1 | 4157.3 | 470.0 |
| | ±225.9 | ±17.8 | ±39.6 | ±15.7 | ±187.7 | ±81.9 | ±219.7 | ±133.2 | ±49.2 | ±19.4 |
| PUMA2602702 | 74645.2 | 380.9 | 2265.6 | 305.8 | 2539.8 | 385.1 | 1933.9 | 342.5 | 2182.2 | 448.6 |
| | ±225.9 | ±17.8 | ±27.9 | ±13.6 | ±204.4 | ±92.9 | ±55.6 | ±26.2 | ±34.2 | ±16.8 |
| PUMA2801100 | 74645.2 | 380.9 | 1601.8 | 318.0 | 1568.5 | 295.5 | 1363.5 | 401.9 | 1030.2 | 227.7 |
| | ±225.9 | ±17.8 | ±22.3 | ±12.3 | ±143.6 | ±48.6 | ±68.5 | ±42.3 | ±20.9 | ±9.8 |
| PUMA2901901 | 74645.2 | 380.9 | 2193.4 | 296.3 | 2100.6 | 304.0 | 1938.0 | 358.5 | 2016.3 | 425.9 |
| | ±225.9 | ±17.8 | ±26.3 | ±11.7 | ±131.7 | ±71.0 | ±70.9 | ±34.7 | ±32.1 | ±15.3 |
| PUMA3200405 | 74645.2 | 380.9 | 4765.9 | 355.4 | 4221.0 | 296.7 | 4444.7 | 467.5 | 5518.1 | 642.2 |
| | ±225.9 | ±17.8 | ±40.9 | ±15.4 | ±272.2 | ±105.7 | ±94.0 | ±47.5 | ±66.4 | ±37.7 |
| PUMA3603710 | 74645.2 | 380.9 | 6786.2 | 311.4 | 7413.6 | 462.8 | 6085.4 | 285.0 | 7136.2 | 403.9 |
| | ±225.9 | ±17.8 | ±48.8 | ±13.1 | ±345.2 | ±70.4 | ±70.2 | ±12.4 | ±63.6 | ±18.6 |
| PUMA3604010 | 74645.2 | 380.9 | 2825.3 | 328.8 | 2927.2 | 324.0 | 2616.4 | 352.4 | 2899.9 | 400.2 |
| | ±225.9 | ±17.8 | ±27.2 | ±13.3 | ±193.1 | ±92.8 | ±74.0 | ±41.5 | ±40.6 | ±16.3 |
| PUMA5101301 | 74645.2 | 380.9 | 3159.6 | 323.3 | 3111.8 | 424.3 | 3284.7 | 581.1 | 3193.0 | 457.3 |
| | ±225.9 | ±17.8 | ±32.5 | ±14.3 | ±188.5 | ±81.9 | ±108.0 | ±61.5 | ±43.2 | ±16.6 |
| PUMA5151255 | 74645.2 | 380.9 | 4997.8 | 374.8 | 5258.1 | 444.1 | 5096.9 | 480.2 | 5444.9 | 579.4 |
| | ±225.9 | ±17.8 | ±42.0 | ±16.1 | ±313.8 | ±116.0 | ±86.8 | ±41.3 | ±67.9 | ±42.3 |

TABLE 59. Squared Errors (with standard deviations). Id Query. PUMS datasets. Gauss Mechanism ($\rho = 0.005$).

| Dataset | olsalg | nnlsalg | maxalg | seqalg | weightalg |
|---|---|---|---|---|---|
| PUMA0101301 | 323.4 | 1623.3 | 1661.8 | 395.8 | 286.2 |
|  | ±13.8 | ±38.5 | ±224.8 | ±26.2 | ±12.4 |
| PUMA0800803 | 323.4 | 1379.3 | 1373.5 | 356.6 | 306.1 |
|  | ±13.8 | ±35.7 | ±220.1 | ±38.9 | ±12.9 |
| PUMA1304600 | 323.4 | 1297.6 | 1077.2 | 505.3 | 313.5 |
|  | ±13.8 | ±34.7 | ±184.6 | ±78.3 | ±13.5 |
| PUMA1703529 | 323.4 | 1346.2 | 1271.7 | 442.0 | 306.6 |
|  | ±13.8 | ±35.4 | ±178.3 | ±75.3 | ±13.0 |
| PUMA1703531 | 323.4 | 1747.3 | 1927.9 | 387.0 | 278.7 |
|  | ±13.8 | ±39.5 | ±223.2 | ±28.0 | ±12.2 |
| PUMA1901700 | 323.4 | 1242.3 | 1579.2 | 444.1 | 317.7 |
|  | ±13.8 | ±33.9 | ±208.5 | ±60.1 | ±13.6 |
| PUMA2401004 | 323.4 | 1138.8 | 1142.0 | 216.5 | 309.0 |
|  | ±13.8 | ±32.6 | ±146.4 | ±42.8 | ±13.1 |
| PUMA2602702 | 323.4 | 1457.8 | 1228.1 | 432.5 | 306.4 |
|  | ±13.8 | ±36.7 | ±189.6 | ±35.6 | ±12.8 |
| PUMA2801100 | 323.4 | 1651.1 | 2102.5 | 492.5 | 295.0 |
|  | ±13.8 | ±38.8 | ±283.4 | ±50.5 | ±12.7 |
| PUMA2901901 | 323.4 | 1493.7 | 1150.0 | 414.4 | 303.9 |
|  | ±13.8 | ±37.1 | ±172.6 | ±51.1 | ±12.9 |
| PUMA3200405 | 323.4 | 1058.1 | 965.0 | 422.6 | 321.7 |
|  | ±13.8 | ±31.4 | ±256.9 | ±39.9 | ±13.6 |
| PUMA3603710 | 323.4 | 952.9 | 967.8 | 379.0 | 331.0 |
|  | ±13.8 | ±29.7 | ±155.3 | ±25.2 | ±14.1 |
| PUMA3604010 | 323.4 | 1270.3 | 1090.5 | 470.3 | 327.4 |
|  | ±13.8 | ±34.3 | ±262.5 | ±58.1 | ±13.8 |
| PUMA5101301 | 323.4 | 1255.1 | 957.0 | 373.1 | 310.9 |
|  | ±13.8 | ±34.2 | ±162.1 | ±46.9 | ±13.1 |
| PUMA5151255 | 323.4 | 1047.3 | 838.4 | 378.7 | 328.4 |
|  | ±13.8 | ±31.2 | ±159.5 | ±30.1 | ±13.9 |

TABLE 60. Squared Error (with standard deviations). Sum Query. PUMS datasets. Gauss Mechanism ($\rho = 0.005$).