# OpenReview forum: "An Uncertainty Principle is a Price of Privacy-Preserving Microdata"
_NeurIPS.cc/2021/Conference — NeurIPS 2021 Poster_

### Official Review · Reviewer_bo89 · 2021-07-11

**Rating:** 8
**Confidence:** 4

**Summary:**


The authors study the problem of quantifying the statistical price of
releasing sum queries (for a population) versus point queries (for sub-populations).
Microdata (e.g., used extensively by the U.S. Census Bureau and associated governmental
agencies) is certainly very valuable. But naively releasing these microdata could lead to
privacy violations. So we can resort to differentially private (DP) methods for microdata
release.
For DP sum queries, one might incur an additional $O(d^2)$ multiplicative factor in additive
error loss while a $O(\log^2 d)$ factor might be incurred in the point queries setting.

The goal of the paper, it seems, is to create a collection of benchmark datasets and
mitigation strategies to alleviate ("inherent and unavoidable") issues with DP microdata release.

The uncertainty principle presented suggests that microdata-generating algorithms should be
designed as postprocessing algorithms that transform (unbiased) noisy measurements into microdata.

Through an extensive set of experiments (see Table 1, 2), the authors validate their principles.

**Ethics Review Area:**

["I don’t know"]

**Limitations And Societal Impact:**

Yes, the authors adequately consider the limitations and broader impact of their work. See Section 7 in their paper.

**Main Review:**

It is known publicly that the U.S. Census Bureau release of microdata had significant
anomalies. The system first produced measurements via DP noisy quer answers and then
attempted to match (via post-processing) the noisy measurements to certain microdata.

The authors introduce an uncertainty principle that trades off error between accuracy on
populations and accuracy on sub-populations.

The Uncertainty Principle
-------------------------

There are two criterion for squared error they consider.
The first is the per-query expected error (let's call PQ)
which is the maximum over all queries of the expected
error and the second is the sum-query (let's call SQ)
expected error which is the sum over all the
queries. The authors posit that to bound both PQ and SQ, one has to choose between a
$O(\log^2 d)$ penalty term for point queries and a $O(d^2)$ penalty term for sum queries.

The authors note that their bounds (upper and lower bounds) affect some, *but not all*,
datasets.

Algorithms
----------

The authors present two baselines and two proposed algorithms all based on first computing
noisy query answers and then postprocessing.

I'm wandering what happens with the baselines when
variance/std of $F_q$ is close to 0? It seems the optimization problem becomes infeasible. [response acknowledged]

Minor Comments
--------------
- Line 44: "... of Black of African Americans ..." -> "... of Black or African Americans ..."


**Time Spent Reviewing:**

4 hours

---

> ### Author Response · Authors · 2021-08-05
> **Response to question about when variance/std of F_q is close to 0**
>
> When the baseline variance is close to 0 (i.e., the queries have almost no noise) then the pseudocode of course needs some changes for numerical stability (these are more like implementation details rather than research details), but with such little noise essentially any postprocessing method would work well.

---

### Official Review · Reviewer_4cAR · 2021-07-12

**Rating:** 7
**Confidence:** 3

**Summary:**

The paper studies the problem of releasing count queries on a population along with counts from sub-populations. The first result of the paper is an “uncertainty principle“, a lower bound stating that if the mechanism is private (DP, approx, or zCDP) then either the error in at least one of on the subpopulations is large, or the error on the population is large. Then, in Theorem 5 the authors prove that there are private algorithms that nearly match the lower bounds. Finally, the authors provide new post processing algorithms for releasing microdata and evaluate them on 16 datasets with positive results.

**Limitations And Societal Impact:**

yes

**Main Review:**

The problem that the authors study is definitely of importance, the Census being a great example. The paper is easy to read and follow. I am unsure about how original the proof of Theorem 3 is and I think the authors should discuss a sketch of the proofs or discuss the novelty in the argument (if any). The proof of theorem 4 seems nontrivial and the postprocessing algorithms in section 5 seem original, specially Algorithm 2.
The experiments section shows that the proposed algorithms perform well in at least 2 of the proposed datasets, I think the section could be considerably improved by summarizing what were the results in the other 14 datasets.

**Time Spent Reviewing:**

3

---

> ### Author Response · Authors · 2021-08-05
> **response to questions**
>
> Thank you for the questions.
>
> - We view the novelty to be not in the proof of theorem 3, but in the conclusion/interpretation of the math, that there is an uncertainty principle explaining anomalies found in a CNSTAT workshop on the Census demonstration products.
>
> - The results for 16 datasets appear in Tables 1 and 2 (each row is a dataset), with additional experiments in the supplementary material. The results on the other datasets (the additional synthetic data) are comparable, with the 15 real datasets (rows 2-16 in the tables in the paper) appearing to be harder than the synthetic ones.

---

### Official Review · Reviewer_NcCk · 2021-07-14

**Rating:** 6
**Confidence:** 2

**Summary:**

The authors consider two different ways of answering counting queries in a differentially private manner; answering the queries directly through a querying system or giving synthetic microdata. They present an "uncertainty principle" stating that, for some data sets and for a particular accuracy metric, the accuracy of differentially private microdata
must suffer either on "sum queries" or "point queries" relative to the performance one could get from answering the queries directly. Finally, they propose algorithms to minimize this accuracy tradeoff and perform some experiments to show empirical performance of these algorithms.

**Limitations And Societal Impact:**

The authors point out that the uncertainty principle does not apply to all data sets and give appropriate caveats for their empirical work. I think they’ve done a pretty good job discussing limitations.

My one hesitation on this front is that I don't think I would go as far as to say that this work “suggests that microdata-generating algorithms should be designed as postprocessing algorithms that convert unbiased noisy measurements into microdata.” That may  be good advice, but the uncertainty principle is stated relative only to the max expected per-query squared error metric. I think the authors provide a reasonable justification for why that metric is practically important but I don't think it's obviously more important than other metrics on which synthetic microdata might perform better.

**Main Review:**

My review focuses primarily on practical and conceptual considerations, rather than the theoretical contribution.

I think the question of how to responsibly produce differentially private statistics for practitioners is important and timely. I find the uncertainty principle result to be interesting and a potentially useful consideration for data providers.

If I understand correctly, the uncertainty principle is an artefact of the requirement that synthetic data be non-negatively weighted. Nonnegative weights seems a natural requirement, but some survey data (ACS for example) do allow for negatively weighted records. So producing negatively weighted synthetic data would not (as far as I know) be completely out of line with current norms for survey data. Granted, there is at least some confusion over this and one potential solution people use is setting these weights to 0, which would reduce the setting back to your non-negative weights (see [here](https://forum.ipums.org/t/what-do-negative-replicate-weights-mean/2519)).

**Typos/Clarifications/Style**

Line 45: "Black **or** African Americans”

Line 823 of the full version should be $X_{100}$ rather than $X_{1}00$

The paper occasionally refers to "the result", "our results", etc. I understand this to be either lines 56-59 or Theorem 3, depending on the location, but I think this could be made more explicit.

**Time Spent Reviewing:**

8

---

> ### Author Response · Authors · 2021-08-05
> **Implications of negative weights**
>
> Yes, negatively weighted data would fix the uncertainty principle but that seems to have major policy implications:
>
> - For example, many end-user analyses, redistricting software, and federal funding allocation formulas do not work with negative numbers.
>
> - Thus, either these use-cases need to be updated, or a lot more research is needed on postprocessing to minimize the uncertainty principle. In both cases, we view this work as something that raises an important issue that needs significant attention from multidisciplinary communities.
>
> As an interesting side note, replicate weights that are used for variance estimation in the ACS and other surveys can have negative weights, but in general final design weights used for point estimation in surveys are positive.

---

> > ### Comment · Reviewer_NcCk · 2021-08-31
> > **Implications of negative weights - Response**
> >
> > Thanks for your response! I agree with the sentiment that negative weights would likely require changing those systems and presents a new set of challenges.

---

### Official Review · Reviewer_eGa2 · 2021-07-16

**Rating:** 6
**Confidence:** 5

**Summary:**

The paper studies the problem of answering disjoint count queries under differential privacy (DP), and presents lower bound of query errors under three variants of DP.

**Limitations And Societal Impact:**

Yes

**Main Review:**

My main concern of the paper is that it ignores two prior studies that present more general and significant results:

1. Irit Dinur, Kobbi Nissim: Revealing information while preserving privacy. PODS 2003: 202-210.

The query error bounds in this paper are applicable to any count queries and any reasonable notion of privacy.

2. Avrim Blum, Katrina Ligett, Aaron Roth: A learning theory approach to noninteractive database privacy. J. ACM 60(2): 12:1-12:25 (2013)

The paper shows that under differential privacy, the query error bounds are decided by the VC dimension of the query set.

Compared with the above studies, the current paper is much narrower in scope, as it only considers disjoint queries under differential privacy. In addition, some of the theoretical results presented in the paper (e.g., the case for \epsilon-differential privacy in Theorem 3) seem to be corollaries of the results in the J. ACM 2013 paper. Therefore, the contribution of the current paper is rather limited.

**Time Spent Reviewing:**

4

---

> ### Author Response · Authors · 2021-08-05
> **Clearing up distinctions from the work mentioned**
>
> Please allow us to correct some confusion, as the results in our paper do not follow from Blum et al. (and we study a different problem than what the review summary suggests; what we look at are tradeoffs between the accuracies of different queries, rather than just lower bounds on query accuracy). The quick version is the metrics we study (per-query error) have significantly different behavior than the metrics they study (simultaneous error), producing a  result that cannot happen under their metrics.
>
> Specifically, Blum et al. consider $P(\max_{q} error(q) > \alpha)$ - a simultaneous bound on error for all queries. We study something closer to $\max_q P(error(q) > \alpha)$ (or more accurately, $\max_q E[error(q)]$) which looks at each query individually. Moving the max to the outside has several important technical and practical implications as it exhibits different types of results and tradeoffs:
>
> 1. For the queries we consider (disjoint queries in addition to their sum) the Laplace mechanism achieves $O(1/\epsilon)$ per-query error. Under the metrics of Blum et al. it would be $O(log(d)/\epsilon)$, where $d$ is the domain size of a record (what Blum et al. call $|X|$). It is also important to note that the  lower bounds of Blum et al. (i.e., Theorem 3.12) do not model the dependence on $|X|$ at all, but this turns out to be an important quantity for our metrics.
>
> 2. Here is one of the differences that arise. If a differentially private algorithm is forced to produce synthetic data, the per query error can no longer be $O(1/\epsilon)$ and instead is roughly $\log(d)/\epsilon$ in the worst case. Meanwhile, nothing changes under simultaneous error - it is still has a logarithmic dependence on $\log(|X|)$, so their metrics behave differently.
>
> 3. However, we dig more deeply into the tradeoffs between the disjoint queries and their sum - tradeoffs that are caused precisely by the synthetic data requirement. Specifically, the per-query error of the disjoint queries can be reduced to $O(1/\epsilon)$
> at the expense of the error in the sum query becoming linearly dependent on the tuple domain size. Or, the sum query error can be made $O(1/\epsilon)$ at the expense of the per-query error in the other queries (logarithmic dependence on tuple domain size).
>
> This result has significant implications for the Census Bureau and other agencies that publish official statistics. It means that for algorithms that take noisy measurements (e.g., with the Laplace or Gaussian mechanisms) and then postprocess them into synthetic data, then in some cases, the synthetic data can be less accurate than some of the noisy measurements  (e.g., by a $\log$ factor in the tuple domain size). This means that:
>
> - Agencies may wish to reconsider producing outputs in the form of synthetic data (for example, by instead releasing data summaries that can produce negative counts).
>
> - Simultaneously releasing synthetic data and the noisy measurements (it was constructed from) will cause confusion among non-statistical users about why they can disagree so much (and about which one is more accurate).
>
> We appreciate the feedback on points we did not clarify well in the paper and will add such a discussion to related work.
>
> **If you still feel our results follow directly from Blum et al., please explain how. Thanks!**

---

> > ### Comment · Reviewer_eGa2 · 2021-09-05
> > **Comments**
> >
> > Thanks for the explanation. It clears my confusion, and I have changed my rating accordingly.
> >
> > One minor comment: If part of the Theorem 3 can be proved as corollaries of Theorem 7.2 in [3], then it is better to credit that part to [3].

---

### Decision · Program_Chairs · 2021-09-28

**Decision:**

Accept (Poster)

**Comment:**

This paper can be viewed as a response to the anomalies being found in the Census data release. The main result is that any private algorithm that is consistent on micro data must incur an error overhead on either "local" queries (think block populations) or a "global" query (think total population). Without a micro data constraint, one can answer these queries with constant error. With the micro data requirement, the authors show that the error must be asymptotically larger for one of these.
This is an interesting paper that throws light on an important phenomenon and explains the cost of micro data requirement. The authors are encouraged to carefully look at the reviews and address reviewer comments and incorporate relevant clarifications from the rebuttal. I recommend that the paper be accepted.

**Consistency Experiment:**

NeurIPS has a long history of experimentation. In 2014, NeurIPS ran an experiment in which 10% of submissions were reviewed by two independent committees to quantify the randomness in the review process. This year, we repeated a variant of this experiment to see how the quality of the review process has changed over time.  This paper was part of the experiment and was therefore assigned to two committees (consisting of reviewers, an Area Chair, and a Senior Area Chair) that reached independent decisions.  If both committees made the same recommendation, this recommendation was followed. If a single committee recommended acceptance, the paper was accepted (with the exception of a few cases in which the other committee identified what we considered a fatal flaw, e.g., an error in a key result).

This copy’s committee reached the following decision: **Accept (Spotlight)**

The other committee assigned to the paper recommended **Accept (Poster)**.  You can find the other set of reviews, along with any follow up discussion with the authors here:
https://openreview.net/forum?id=PYJnWEn-uMn